# Towards the Generation of Structured Scientific Vector Graphics with Large Language Models

## Abstract

We address the challenge of automatically visualizing scientific explanations. While prior work has explored large language model (LLM)-based vector graphic generation, existing approaches often overlook structural correctness, a key requirement for valid scientific diagrams. To achieve structurally correct generation, we make three key contributions. First, we introduce SSVG-Bench, a novel benchmark for evaluating the generation of Structured Scientific Vector Graphics. Unlike conventional visual similarity metrics, SSVG-Bench employs task-specific structural analysis for accurate evaluation, and it supports three vector formats: TikZ, SVG, and EPS. Second, we conduct an extensive benchmarking and analysis, revealing key findings such as the crucial role of LLM reasoning in ensuring structural validity. Third, we propose LLM-Oriented Orchestration Prompting (LOOP), a new prompting method that leverages LLMs' reasoning potential by combining familiar subtasks. Experiments demonstrate substantial improvements over existing prompting techniques, suggesting promising directions for scientific diagram generation. We will release our code and benchmark upon acceptance.

## 1 Introduction

In this paper, we address the challenge of automatically visualizing scientific explanations. Because scientific explanations are often abstract and complex, they can be difficult to grasp from text alone. To promote intuitive understanding, visualization is widely used in contexts such as science textbooks and research papers. These observations suggest that automated visualization could have a significant impact.

Recent research on scientific diagram generation has investigated vector graphics generation with large language models (LLMs) (Belouadi et al., 2024a;b; Zhang et al., 2025; Belouadi et al., 2025). As vector graphics encode visual content in structured text form, they can be directly produced by LLMs. Given that LLMs are capable of capturing scientific concepts in depth and encoding complex constraints, they are particularly promising for this task.

However, existing methods have overlooked a crucial aspect of scientific diagrams: structural correctness. To illustrate its importance, we present a physics scenario in Figure 1. Here, the visualization must strictly satisfy structural constraints: the object should remain in contact with the inclined plane, and three arrows must be shown, one vertical to the ground, one perpendicular to the plane, and one parallel to the plane. An existing method, namely a fine-tuned LLM for generating vector code (Belouadi et al., 2025), fails to meet these constraints. Although it prioritizes visual plausibility, the lack of structural enforcement ultimately leads to invalid scientific diagrams.

Towards the generation of scientific vector graphics with structural correctness, we make three main contributions. Our first contribution is a new benchmark for the generation of Structured Scientific Vector Graphics, named SSVG-Bench. It targets two foundational domains: plane geometry and molecular structure. The plane geometry task involves generating geometric figures from textual descriptions of theorems or construction methods, while the molecular structure task requires correctly generating a molecule's structure from its IUPAC name, which encodes structural information. These domains are representative of broader applications: the ability to generate plane geometry structures is fundamental for physics illustrations, engineering diagrams, and architec-

| Textual explanation | Expected output | Output by TikZero+ |
| --- | --- | --- |
| An object on an inclined plane experiences three forces: the gravitational force mg acting vertically downward, the normal force of magnitude mgcos(theta) acting perpendicular to the plane, and a static friction force of magnitude mgsin(theta) acting up the plane, parallel to its surface. | 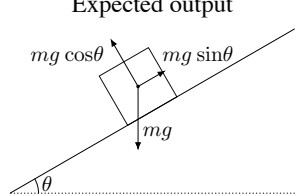 | 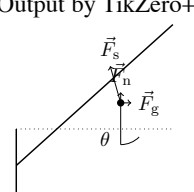 |

Figure 1: An example highlighting the importance of structural correctness: the object should remain in contact with the inclined plane, and three arrows must be shown, one vertical to the ground, one perpendicular to the plane, and one parallel to the plane. TikZero+ (Belouadi et al., 2025), a fine-tuned LLM for generating vector code, does not meet these structural constraints.

tural blueprints, while generating molecular structures requires correctly producing graph structures, which are important in various fields such as algorithm flowcharts, circuit designs, and biological pathways. The most significant feature of SSVG-Bench is its evaluation method. Conventional visual similarity metrics are insufficient for assessing structural correctness, as even minor visual changes can cause structural inconsistencies. To address this, we provide task-specific Python scripts that analyze the structure of the generated outputs, enabling accurate evaluation of structural correctness. In addition, SSVG-Bench supports three vector formats: TikZ, SVG, and EPS, which allows us to examine performance across formats.

Our second contribution is a comprehensive benchmarking and analysis of existing models using SSVG-Bench, which yields several key findings. First, LLLs fine-tuned on existing vector graphic generation datasets to produce vector code rarely generate structurally valid vector graphics. Second, we show that the reasoning capabilities of LLMs are essential for generating structurally correct vector graphics. Finally, although prior work has mainly focused on the TikZ format, our results demonstrate that the SVG format is better suited for LLM reasoning.

As our third contribution, we propose a new prompting technique, LLM-Oriented Orchestration Prompting (LOOP), to further enhance the reasoning capabilities of LLMs. Recent LLMs are explicitly trained to perform step-by-step reasoning and can solve complex tasks, but it has been reported that they struggle with tasks not encountered during training (Shojaee et al., 2025; Malek et al., 2025). Since LLMs are not explicitly trained to generate vector graphics from scientific explanations, they cannot fully utilize their reasoning potential when the task is presented in its original form. To fully leverage their reasoning abilities, we design a prompt that explicitly guides LLMs to perform LLM-friendly subtasks such as information extraction and relationship extraction, which enables them to generate vector graphics with correct structure. Our experiments demonstrate that LOOP achieves substantially better performance than existing prompting methods.

Our contributions can be summarized as follows.

- Dataset: We introduce SSVG-Bench, a new benchmark for structured scientific vector graphics generation, including scripts that verify structural correctness.
- Benchmarking and analysis: Our analysis shows that previous fine-tuned models cannot produce structurally correct graphics, that LLM reasoning capabilities are essential for ensuring structural correctness, and that the SVG format is well-suited for such reasoning.
- Method: We propose LOOP, a new prompting method that enhances LLM reasoning by solving LLM-friendly sub-problems step by step.

## 2 RELATED WORKS

**Scientific vector graphic generation benchmarks.** Vector graphics are gaining attention as an image format well-suited for LLMs, as they are represented in text and can be directly input or output by LLMs without requiring a vision adapter. Several benchmarks have been developed for general vector graphics generation, such as SVGEditBench (Nishina & Matsui, 2024), SVG Taxonomy (Xu & Wall, 2024), and VGBench (Zou et al., 2024). Additionally, several benchmarks for visualizing scientific data have been developed, such as MatPlotBench (Yang et al., 2024), Pandas-PlotBench (Galimzyanov et al., 2025), and ChartMimic (Yang et al., 2025).

Table 1: Comparison of previous benchmarks with our SSVG-Bench.

| Benchmark | Scientific? | # Evaluation data | Evaluation method | Vector format |
|---|---|---|---|---|
| SVGEditBench | No | 1,366 | Visual similarity | SVG |
| SVG Taxonomy | No | 400 | Exact match | SVG |
| VGBench | No | 5,845 | Visual similarity | TikZ, SVG, Graphviz |
| DaTikZ v1 | Yes | 1,000 | Visual/code similarity, Human evaluation | TikZ |
| DaTikZ v2 | Yes | 1,000 | Visual/code similarity, Human evaluation | TikZ |
| DaTikZ v3 | Yes | 1,000 | Visual/code similarity, Human evaluation | TikZ |
| ScImage | Yes | 404 | Human evaluation | TikZ |
| DiagramGenBench | Yes | 470 | Visual/code similarity, Human evaluation | TikZ, DOT |
| **SSVG-Bench (ours)** | **Yes** | **1,230** | **Structural analysis scripts** | **TikZ, SVG, EPS** |

Motivated by this trend, some benchmarks have also been created specifically for scientific vector graphics generation. DaTikZ v1 (Belouadi et al., 2024a) collects TikZ code and corresponding captions from sources such as arXiv papers. DaTikZ v2 (Belouadi et al., 2024b) collects pairs of hand-drawn sketches and TikZ code to evaluate the performance of sketch-to-TikZ conversion. DaTikZ v3 (Belouadi et al., 2025) further extends DaTikZ v1 and v2. ScImage (Zhang et al., 2025) employs synthetic data to analyze scientific vector generation in terms of attributes, numbers, and spatial dimensions. DiagramGenBenchmark (Wei et al., 2025) provides diagram structures in TikZ as well as graph structures in the DOT language.

However, these benchmarks generally do not focus on the structural correctness of the generated graphics. We present a comparison of these benchmarks with our SSVG-Bench in Table 15. The most significant feature of SSVG-Bench is its evaluation method. Visual similarity-based and code similarity-based metrics are insufficient for determining whether the structure of a generated graphic truly reflects the intended structure. Human evaluations, while informative, are not scalable and are subject to variability and inconsistency across evaluators. As a result, it is difficult to increase the number of models evaluated or to use diverse experimental conditions. To address these issues, we provide task-specific Python scripts that analyze the structure of the generated output and determine its correctness. Our evaluation framework offers precise, objective, and consistent assessments of performance. In addition, SSVG-Bench supports three vector formats: TikZ, SVG, and EPS, which allows us to examine performance across formats.

**Scientific vector graphic generation methods.** Based on the benchmarks, several scientific vector graphic generation methods have been proposed. AutomaTikZ (Belouadi et al., 2024a) is designed for TikZ generation by fine-tuning Llama (Touvron et al., 2023) to output TikZ code from captions. It leverages CLIP features (Radford et al., 2021), derived from captions, to further improve visual alignment. DeTikZify (Belouadi et al., 2024b) converts hand-drawn sketches into TikZ code by combining a vision encoder (SigLIP (Zhai et al., 2023)) with an LLM such as Llama. TikZero (Belouadi et al., 2025) addresses the scarcity of paired caption-TikZ data by leveraging readily available captioned raster images for training. DiagramAgent (Wei et al., 2025) enables the creation of complex diagrams by coordinating multiple agents. Despite their innovations, these models are primarily trained to predict output code and are not designed to guarantee structural correctness.

**LLM prompting methods.** The reasoning capabilities of LLMs can be elicited through effective prompting. Seminal work on Chain-of-Thought (CoT) demonstrated that allowing models to generate intermediate reasoning steps dramatically improves multi-step reasoning (Wei et al., 2022). Follow-ups revealed that simply appending "Let's think step by step" can unlock zero-shot reasoning (Kojima et al., 2022), and that sampling multiple reasoning paths and selecting the most consistent answer ("self-consistency") further boosts accuracy (Wang et al., 2023b). Beyond linear reasoning, researchers decomposed problems via least-to-most prompting (Zhou et al., 2023), introduced plan-first then execute strategies such as Plan-and-Solve (Wang et al., 2023a), and proposed Step-Back prompting, which encourages the model to abstract away from the immediate problem and reason at a higher conceptual level before providing a solution (Zheng et al., 2024). In this paper, we introduce a novel prompting method designed for scientific vector graphic generation.

| **Plane geometry** | | |
|---|---|---|
| Vector graphic |  | |
| Input text | The exterior angle bisector in A intersects the extended side BC in E, the exterior angle bisector in B intersects the extended side AC in D and the exterior angle bisector in C intersects the extended side AB in F.

The three points of intersection between the exterior angle bisectors and the extended triangle sides D, E, F are collinear, that is they lie on a common line. | To state the theorem, suppose that ABCD and AB'C'D' are two squares with common vertex A. Let E and G be the midpoints of B'D and D'B respectively, and let F and H be the centers of the two squares. Then the theorem states that the quadrilateral EFGH is a square as well. | The Apollonius point of a triangle is defined as follows. Let △ABC be any given triangle. Let the excircles of △ ABC opposite to the vertices A, B, C be EA, EB, EC respectively. Let E be the circle which touches the three excircles EA, EB, EC such that the three excircles are within E. Let A', B', C' be the points of contact of the circle E with the three excircles. The lines AA', BB', CC' are concurrent. The point of concurrence is the Apollonius point of △ABC. |

| **Molecular structure** | | |
|---|---|---|
| Vector graphic |  | |
| Input text | the molecular structure of the compound with the IUPAC name 4-butyl-2,6-dimethylmorpholine | the molecular structure of the compound with the IUPAC name 6-methoxy-4-methyl-5-phenyl methoxyquinolin-8-amine | the molecular structure of the compound with the IUPAC name 4-[[2,4-diamino-5-[(4-carboxyphenyl) diazenyl]phenyl]diazenyl]benzoic acid |

Figure 2: Examples in SSVG-Bench.[1]

## 3 SSVG-BENCH

To evaluate whether LLMs can generate vector graphics with structural correctness, we introduce SSVG-Bench. SSVG-Bench covers two key topics: 1) plane geometry and 2) molecular structure. For each topic, we also develop an automatic evaluation framework. Figure 2 presents some examples, and Table 2 summarizes respective statistics. We provide a detailed explanation of these components in the following sections.

### 3.1 PLANE GEOMETRY

This task involves generating visualizations of plane geometric figures from textual descriptions, translating explanations of theorems or constructions into precise visual representations. To succeed, LLMs must accurately interpret spatial relationships such as "intersection", "tangent", and "perpendicular", as well as uniquely determined constructions like "angle bisectors" and "excircles." This task can evaluate visualization capabilities that are important in a variety of applications, including physics illustrations, engineering diagrams, and architectural blueprints.

Table 2: Statistics of SSVG-Bench. The number of elements is counted based on the SVG.

| **Plane geometry** | |
|---|---|
| Total number of input texts | 110 |
| Average number of elements per vector graphic | |
| - line | 3.96 |
| - circle | 2.77 |
| - ellipse | 0.03 |
| - polygon | 1.65 |
| - polyline | 0.01 |

| **Molecular structure** | |
|---|---|
| Total number of input texts | 300 |
| Average number of elements per vector graphic | |
| - line | 32.45 |
| - circle | 30.38 |

---

[1]The input texts and vector graphics are from (Wikipedia contributors, 2025b;f; 2024; Kmhkmh, 2019a; 2015; Krishnachandranvn, 2012; National Center for Biotechnology Information, 2025f;l;g)

Input text The segment AB is bisected by drawing intersecting circles of equal radius r > 1/2|AB|, whose centers are the endpoints of the segment. The line determined by the points of intersection of the two circles is the perpendicular bisector of the segment.

Output vector graphic (TikZ)

Automatic evaluation results

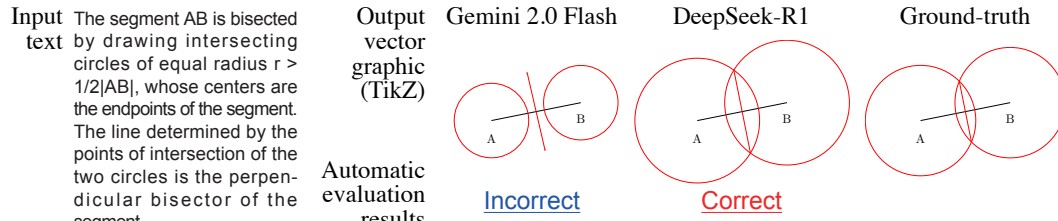

Figure 3: An example of the automatic evaluation results for Pattern 2 in the plane geometry visualization task, where the correct object is not uniquely determined. Our Python-based automatic evaluation framework checks whether the radius of each circle is greater than half the length of segment AB, enabling appropriate assessment.[2]

Input text the molecular structure of the compound with the IUPAC name tellurophene-2-carboxylic acid

Output vector graphic (TikZ)

Automatic evaluation results

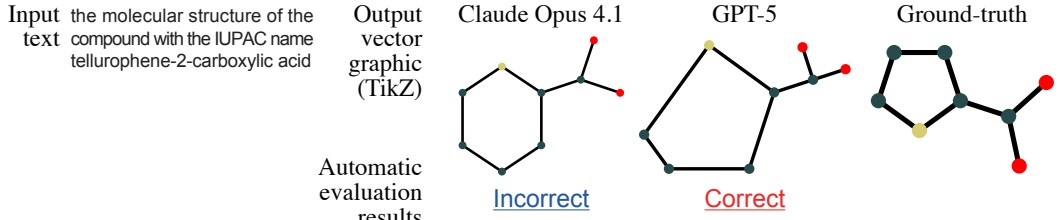

Figure 4: An example of the automatic evaluation results in the molecular structure visualization task. Correctness can be determined based on graph analysis, even when the images differ significantly in visual appearance.[3]

We curated a dataset by collecting paired textual descriptions and corresponding images related to plane geometry from Wikipedia. All images used are freely available for modification and redistribution. Most of them are in SVG, a vector graphic format. We cleaned the images using Adobe Illustrator by removing elements unrelated to the text. When only raster images were available, we manually recreated the visuals in vector format. In total, we compiled 110 text-vector graphic pairs.

The automatic evaluation falls into two distinct patterns. In Pattern 1, the correct output can be uniquely determined. We provide some parts of the vector graphics as input to the LLM. As shown in Figure 2, the elements depicted in black are given as input, and the LLM is expected to generate the red elements. These red elements are uniquely determined by the black elements. We developed a Python script to analyze the vector data and assess correctness by checking whether the red components are present in the LLM's output.

In Pattern 2, the correct output cannot be uniquely determined, even when the black elements are provided. For example, in the case illustrated in Figure 3, any circle with a radius greater than half the length of the given line segment is considered acceptable. For such cases, we implemented case-specific Python logic to evaluate correctness based on the textual input, allowing for variation in valid outputs. In Figure 3, the output of Gemini 2.0 Flash is judged as incorrect because the radius of the circle is shorter than half the length of segment AB. Although the output of DeepSeek-R1 differs from the predefined ground-truth radius, it is judged as correct since their circle radii exceed half the length of segment AB. We carefully read each text and manually categorized them into Pattern 1 and Pattern 2. There are 94 cases belonging to Pattern 1 and 16 cases belonging to Pattern 2.

For simplicity, we exclude text layout from the evaluation process.

## 3.2 MOLECULAR STRUCTURE

In this task, the input is an IUPAC name that describes the molecular structure, and the goal is to generate the corresponding molecular structure. The IUPAC name encodes the structural informa-

---

[2]The input text and the ground-truth are from (Wikipedia contributors, 2025c) and (Ag2gaeh, 2021).

[3]The molecular structure data is from (National Center for Biotechnology Information, 2025p).

tion of a molecule, and by interpreting it, the molecular structure can be reconstructed. For example, the IUPAC name shown on the left side of Figure 2 is "4-butyl-2,6-dimethylmorpholine," which indicates that a butyl group is attached to the nitrogen atom at the 4-position (shown in blue) of the morpholine ring (a six-membered ring), and that methyl groups are attached at the 2- and 6-positions. In this way, the molecular structure can be accurately restored from the IUPAC name. It is essential to correctly identify the types and numbers of atoms involved, with particular importance placed on the graph structure formed by atomic bonds. This task can evaluate the graph structure visualization capabilities of LLMs, which are important in various fields such as algorithm flowcharts, circuit design, and biological pathways.

We obtained pairs of IUPAC names and molecular structures from PubChem[4]. The structural data is stored in JSON format, and we developed a Python script to convert this information into vector graphics automatically. Using this script, we generated ground-truth data. We collected 300 molecules in total, with 50 examples each for molecules with fewer than 20 elements (atoms plus bonds), 21-40, 41-60, 61-80, 81-100, and more than 100 elements.

To enable automatic evaluation, we implemented a Python-based evaluation tool. The generated vector graphic is converted into a molecular graph, where nodes represent atoms and edges represent bonds. We then check for "graph isomorphism" between the generated structure and the ground-truth to automatically assess correctness. Graph isomorphism refers to the problem of determining whether two graphs are structurally identical, meaning their nodes and edges can be matched one-to-one while preserving connectivity. We use the NetworkX library to solve this problem. To simplify the task, we do not consider bond order. We present an example of automatic evaluation results in Figure 4. Although the output of GPT-5 appears visually different from the ground-truth, it is considered correct based on graph-theoretic equivalence.

### 3.3 MULTIPLE VECTOR FORMATS

There are various types of vector graphic formats. To analyze performance differences across formats, we use three types: TikZ, SVG, and EPS. For the plane geometry task, since the vector graphics collected from Wikipedia are in SVG format, we developed Python scripts to automatically convert SVG to TikZ and EPS, thereby generating ground-truth data. For the molecular structure task, we generate vector graphics in each format directly from molecular structure data stored in JSON files, using custom Python scripts to automate the process. When evaluating LLMs, we add instructions to the prompt to generate output in a specific format. This approach allows us to explore which vector format is most suitable for LLMs. If the syntax is incorrect, it will result in a compilation error (for TikZ and EPS) or a parsing error (for SVG). In such cases, the output is considered incorrect. Therefore, LLMs must strictly adhere to the syntax of each format. Our SSVG-Bench dataset consists of two tasks and three vector formats, comprising a total of 1,230 text-vector graphic pairs.

### 4 BENCHMARKING AND ANALYSIS

Using SSVG-Bench, we evaluate whether recent models can generate structurally correct vector graphics. Our experiments utilize two fine-tuned models: AutomaTikZ (Belouadi et al., 2024a) and TikZero+ (Belouadi et al., 2025). As these models are trained to generate TikZ, we evaluate them exclusively on TikZ. We also evaluate recent general-purpose LLMs, including DeepSeek-V3, R1, V3.1 (DeepSeek, 2025), Claude Opus 4.1 (Anthropic, 2025), Gemini 2.0 Flash, 2.5 Flash, 2.5 Pro (Google, 2025), o4-mini, GPT-4.1, and GPT-5 (OpenAI, 2025). The detailed prompts used for evaluation are provided in the Appendix. The overall performance is shown in Table 4, with a detailed analysis provided below.

**Limitations of fine-tuned models.** Our benchmarking with SSVG-Bench reveals that models fine-tuned to generate TikZ code (AutomaTikZ and TikZero+) seldom produce structurally valid outputs. This highlights a limitation: simply training to predict TikZ code from captions is insufficient to generate correct scientific figures.

---

[4]https://pubchem.ncbi.nlm.nih.gov/

Table 4: Accuracies on SSVG-Bench (%). The fill colors correspond to the values.

| Model | Plane geometry | | | Molecular structure | | | Average |
|---|---|---|---|---|---|---|---|
| | TikZ | SVG | EPS | TikZ | SVG | EPS | |
| **Fine-tuned models** | | | | | | | |
| AutomaTikZ | 0.0 | - | - | 0.0 | - | - | 0.0 |
| TikZero+ | 0.9 | - | - | 0.0 | - | - | 0.2 |
| **Non-reasoning models** | | | | | | | |
| DeepSeek-V3 | 10.0 | 5.5 | 7.3 | 5.3 | 3.3 | 3.0 | 4.9 |
| DeepSeek-V3.1 | 11.8 | 6.4 | 9.1 | 6.3 | 3.7 | 3.3 | 5.7 |
| Claude Opus 4.1 | 14.5 | 12.7 | 20.9 | 24.3 | 26.0 | 16.0 | 20.5 |
| Gemini 2.0 Flash | 7.3 | 5.5 | 1.8 | 6.0 | 3.7 | 0.7 | 3.8 |
| Gemini 2.5 Flash | 12.7 | 9.1 | 5.5 | 22.7 | 11.3 | 14.3 | 14.2 |
| GPT-4.1 | 10.9 | 10.0 | 14.5 | 19.0 | 15.0 | 13.7 | 14.8 |
| GPT-5 Chat | 12.7 | 10.0 | 7.3 | 16.0 | 14.3 | 11.0 | 12.8 |
| **Reasoning models** | | | | | | | |
| DeepSeek-R1 | 28.2 | 40.9 | 39.1 | 18.3 | 20.0 | 19.7 | 23.8 |
| DeepSeek-V3.1 reasoning | 23.6 | 39.1 | 27.3 | 31.0 | 7.3 | 20.7 | 22.4 |
| Claude Opus 4.1 thinking | 20.0 | 23.6 | 17.3 | 26.7 | 27.7 | 23.3 | 24.4 |
| Gemini 2.5 Flash reasoning | 30.0 | 55.5 | 41.8 | 32.0 | 39.3 | 34.7 | 37.2 |
| Gemini 2.5 Pro | 50.0 | 62.7 | 56.4 | 41.3 | **63.3** | **57.3** | 54.6 |
| o4-mini | 48.2 | 62.7 | 55.5 | 33.3 | 42.7 | 39.0 | 42.9 |
| GPT-5 | **54.5** | **75.5** | **66.4** | **52.3** | 55.7 | 49.7 | **56.0** |

Table 5: Comparison of averages for models with reasoning enabled vs. disabled. DeepSeek-V3.1, Claude Opus 4.1, Gemini 2.5 Flash, and GPT-5 are considered.

| Model | Plane geometry | | | Molecular structure | | | Average |
|---|---|---|---|---|---|---|---|
| | TikZ | SVG | EPS | TikZ | SVG | EPS | |
| Reasoning disabled | 13.0 | 9.5 | 10.7 | 17.3 | 13.8 | 11.2 | 13.3 |
| Reasoning enabled | **32.0** | **48.4** | **38.2** | **35.5** | **32.5** | **32.1** | **35.0** |

**Importance of reasoning.** To evaluate the effectiveness of reasoning, we consider models where reasoning can be toggled on and off, and we report results for both configurations in Table 4. The models compared are as follows: DeepSeek-V3.1 vs. DeepSeek-V3.1 reasoning, Claude Opus 4.1 vs. Claude Opus 4.1 thinking, Gemini 2.5 Flash vs. Gemini 2.5 Flash reasoning, and GPT-5 Chat vs. GPT-5. Table 5 compares the averages of models with reasoning disabled and enabled. Enabling reasoning significantly improves performance. These results demonstrate that enabling reasoning plays a crucial role in generating structured vector graphics.

**Impact of vector format.** Focusing on the two best-performing models, Gemini 2.5 Pro and GPT-5, we observe that their performance on SVG is the highest, surpassing their performance on

Table 3: Google search hits for format-specific keywords (November 2025).

| | TikZ | SVG | EPS |
|---|---|---|---|
| Keyword | "tikzpicture" | "</svg>" | "showpage" and "moveto" |
| # Hits | 272K | 524M | 20.7K |

TikZ and EPS. This represents a novel finding, as prior benchmarks have primarily focused on TikZ. One possible explanation is the scale of resources used for training. Existing research (Zhu et al., 2024) has demonstrated that the reasoning capabilities of LLMs tend to be weaker in low-resource languages (e.g., Bengali or Thai) compared to high-resource languages (e.g., English). Similarly, it is possible that LLMs are not well-suited for reasoning with TikZ and EPS, because they may be considered "low-resource languages." To test this hypothesis, it would be necessary to examine the training data, but the datasets used to train Gemini 2.5 Pro and GPT-5 remain unspecified. We instead query Google with format-specific keywords and record the number of hits, since much of the training data for LLMs is derived from internet sources. Table 3 presents the number of hits

Table 6: Comparison between our LOOP and other zero-shot prompting methods.

| Model | Plane geometry | | | Molecular structure | | | Average |
|---|---|---|---|---|---|---|---|
| | TikZ | SVG | EPS | TikZ | SVG | EPS | |
| Gemini 2.5 Pro | 50.0 | 62.7 | 56.4 | 41.3 | 63.3 | 57.3 | 54.6 |
| w/ Zero-shot CoT | 39.1 | 66.4 | 61.8 | **47.7** | 63.0 | 58.7 | 56.3 |
| w/ Plan-and-Solve | 39.1 | 69.1 | **66.4** | 41.3 | 58.7 | 55.0 | 53.4 |
| w/ Step-Back | 33.6 | 64.5 | 59.1 | 40.7 | 54.3 | 56.3 | 51.0 |
| w/ LOOP (ours) | **65.5** | **80.9** | 62.7 | **47.7** | **64.7** | **67.7** | **62.6** |
| GPT-5 | 54.5 | 75.5 | 66.4 | 52.3 | 55.7 | 49.7 | 56.0 |
| w/ Zero-shot CoT | 58.2 | **80.0** | 75.5 | 53.0 | 52.0 | 49.3 | 56.7 |
| w/ Plan-and-Solve | 61.8 | 77.3 | 70.9 | 52.3 | 50.3 | 50.7 | 56.2 |
| w/ Step-Back | 55.5 | 75.5 | 72.7 | 50.7 | 51.7 | 48.3 | 55.0 |
| w/ LOOP (ours) | **70.0** | **80.0** | **77.3** | **55.0** | **57.3** | **54.3** | **61.0** |

obtained from searches using format-specific keywords. Compared to TikZ and EPS, SVG yielded a much higher number of hits, suggesting that SVG constitutes a high-resource format.

## 5 LLM-ORIENTED ORCHESTRATION PROMPTING (LOOP)

Through our analysis, we find that LLM reasoning plays a crucial role. Building on this finding, we propose a method to enhance their reasoning capabilities. Previous research has shown that carefully crafted prompts can significantly improve LLM reasoning, even in zero-shot settings. For instance, zero-shot CoT prompting (Kojima et al., 2022), which simply instructs the model with "Let's think step by step," has been shown to improve performance. In this work, we introduce a novel zero-shot prompting method, termed LOOP. While recent LLMs are explicitly trained for step-by-step reasoning and can solve complex tasks, their performance often degrades on tasks outside their training distribution (Shojaee et al., 2025; Malek et al., 2025). Because LLMs are not inherently trained to generate vector graphics from scientific explanations, their reasoning potential remains underutilized. The core idea of LOOP is to instruct LLMs to generate vector graphics by orchestrating LLM-familiar tasks. Specifically, we use the following tasks: 1) information extraction, 2) relationship extraction, 3) mathematical reasoning, and 4) code generation. The first three tasks provide the information necessary for visualization, while the final task produces the vector graphics. Information and relationship extraction are long-standing tasks in the field of natural language processing, whereas mathematical reasoning and code generation are areas where recent LLMs have made significant progress. By orchestrating these familiar tasks, LOOP aims to accelerate and enhance LLM reasoning capabilities.

Specifically, for the plane geometry visual task, we use the following prompt:

*"Let's think step by step, following this workflow: 1. Information extraction: describe the necessary elements. 2. Relationship extraction: describe their relationships. 3. Mathematical reasoning: compute the attributes of each element so that they satisfy those relationships. 4. Code generation: generate the TikZ."*

For the molecular structure visualization task, we exclude mathematical reasoning, as it is not necessary. We use the following prompt:

*"Let's think step by step, following this workflow: 1. Information extraction: describe the functional groups and substituents present in the IUPAC name. 2. Relationship extraction: describe how these groups are connected. 3. Code generation: generate the TikZ."*

The word *"TikZ"* is replaced with the target vector format.

### 5.1 EVALUATION

We evaluate the performance of LOOP by applying it to the two best-performing models: Gemini 2.5 Pro and GPT-5. For comparison, we use the following three zero-shot prompting techniques:

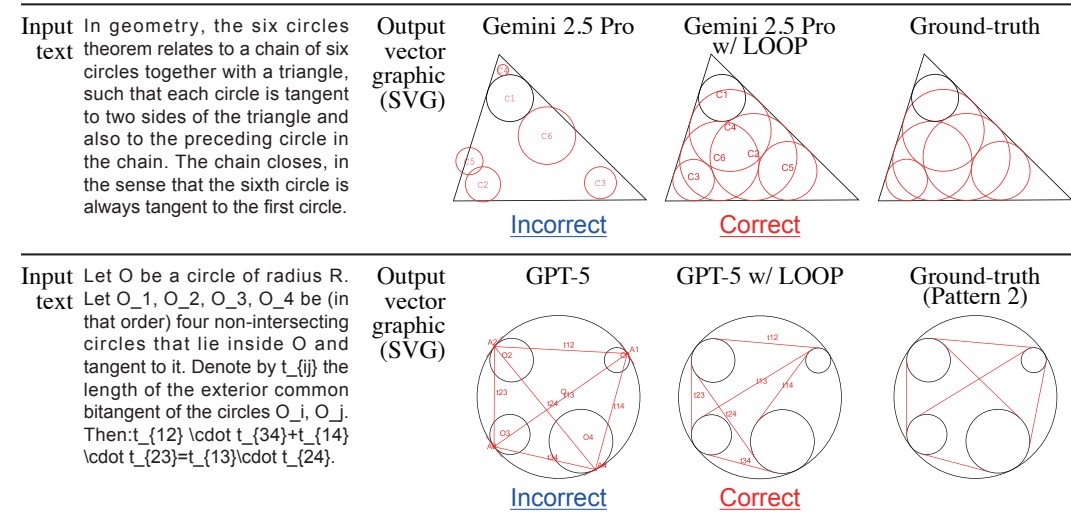

| Input text | In geometry, the six circles theorem relates to a chain of six circles together with a triangle, such that each circle is tangent to two sides of the triangle and also to the preceding circle in the chain. The chain closes, in the sense that the sixth circle is always tangent to the first circle. | Output vector graphic (SVG) | Gemini 2.5 Pro ... Incorrect | Gemini 2.5 Pro w/ LOOP ... Correct | Ground-truth |
| --- | --- | --- | --- | --- | --- |
| Input text | Let O be a circle of radius R. Let $O_1$, $O_2$, $O_3$, $O_4$ be (in that order) four non-intersecting circles that lie inside O and tangent to it. Denote by $t_{ij}$ the length of the exterior common bitangent of the circles $O_i$, $O_j$. Then: $t_{12} \cdot t_{34} + t_{14} \cdot t_{23} = t_{13} \cdot t_{24}$. | Output vector graphic (SVG) | GPT-5 ... Incorrect | GPT-5 w/ LOOP ... Correct | Ground-truth (Pattern 2) |

Figure 5: Examples demonstrating improvements from our prompting method on the plane geometry SVG generation task. In the bottom example, since there are two possible exterior common bitangents for each pair of circles, either line is considered correct.[5]

| Input text | the molecular structure of the compound with the IUPAC name 3-[[4-(4-methylphenyl)piperazin-1-yl]methyl]-5-[(2-methylquinazolin-4-yl)oxymethyl]-1,3,4-oxadiazole-2-thione | Output vector graphic (SVG) | Gemini 2.5 Pro ... Incorrect | Gemini 2.5 Pro w/ LOOP ... Correct | Ground-truth |
| --- | --- | --- | --- | --- | --- |
| Input text | the molecular structure of the compound with the IUPAC name (6-chloro-4-methyl-2-oxochromen-7-yl) (2R)-2-[(4-methylphenyl)sulfonylamino]pentanoate | Output vector graphic (SVG) | GPT-5 ... Incorrect | GPT-5 w/ LOOP ... Correct | Ground-truth |

Figure 6: Examples demonstrating improvements from our prompting method on the molecular structure SVG generation task.[6]

- Zero-shot CoT Prompting (Kojima et al., 2022): *"Let's think step by step."*

- Plan-and-Solve Prompting (Wang et al., 2023a): *"Let's first understand the problem and devise a plan to solve the problem. Then, let's carry out the plan and solve the problem step by step."*

- Step-Back Prompting (Zheng et al., 2024): *"Let's think step by step, following this workflow: 1. Step back and pose higher-level, abstract questions. 2. Answer those questions. 3. Generate the TikZ."*

The results are shown in Table 6. While some methods degrade performance, the proposed method provides the highest performance gain. This result clearly demonstrates that the proposed method can efficiently leverage the potential of LLMs.

We present examples in Figures 5 and 6 showing that LOOP enables the generation of structurally correct vector graphics. Without LOOP, LLMs fail to accurately produce complex structures. LOOP encourages deeper reasoning, which leads to the generation of structurally correct vector graphics.

---

[5]The input texts and the ground-truth vector graphics are from (Wikipedia contributors, 2022; 2025e; Rocchini, 2010; Kmhkmh, 2018).

[6]The molecular structure data is from (National Center for Biotechnology Information, 2025j;o).

## 6 CONCLUSION

In this paper, we tackled the problem of scientific vector graphics generation using LLMs. Specifically, aiming for structurally correct vector graphics generation, we made three contributions. First, we introduced a new benchmark that assesses the structural correctness of generated graphics using structural analysis scripts. Second, we conducted a comprehensive benchmarking study and provided detailed analyses based on this benchmark. Third, we proposed a novel prompting technique that accelerates LLM inference and significantly improves performance.

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

## A    THE USE OF LARGE LANGUAGE MODELS (LLMs)

We used ChatGPT when writing our paper, for translation purposes.

## B    DETAILED PROMPTS USED IN OUR EXPERIMENTS

Figures 7, 8, and 9 show examples of prompts used in the plane geometry task. The [Explanation] section represents the textual description, while the [TikZ], [SVG], and [EPS] sections represent the input elements (the black elements in Figure 2).

Figures 10, 11, and 12 show examples of prompts used in the molecular structure task. We first provide the IUPAC name, followed by a set of instructions. In particular, we include color specifications so that the types of atoms can be identified by their color. We also present an example using chlorobenzene, which is a relatively simple molecule.

[Explanation]The exterior angle bisector in A intersects the extended side BC in E, the exterior angle bisector in B intersects the extended side AC in D and the exterior angle bisector in C intersects the extended side AB in F.

The three points of intersection between the exterior angle bisectors and the extended triangle sides D, E, F are collinear, that is they lie on a common line.

```
[TikZ]\documentclass{standalone}
\usepackage{tikz}
\begin{document}
\begin{tikzpicture}[x=1pt,y=1pt]
\draw (38.23,-264.41) -- (65.98,-195.89) -- (32.16,-166.40) -- cycle;
\node at (23.54,-167.59) {A};
\node at (70.26,-197.89) {B};
\node at (28.79,-270.30) {C};
\end{tikzpicture}
\end{document}
```

Please visualize the explanation by adding elements to the TikZ diagram.
– Set the color of any added elements to red.
– Do not modify any elements that are originally present in the TikZ.
– Do not use any animations.

Figure 7: An example prompt from the TikZ generation task on plane geometry.

[Explanation]The exterior angle bisector in A intersects the extended side BC in E, the exterior angle bisector in B intersects the extended side AC in D and the exterior angle bisector in C intersects the extended side AB in F.

The three points of intersection between the exterior angle bisectors and the extended triangle sides D, E, F are collinear, that is they lie on a common line.

```
[SVG]<svg xmlns="http://www.w3.org/2000/svg"
xmlns:ev="http://www.w3.org/2001/xml-events" version="1.1"
viewBox="0 0 300 300">
<style>
.input_object {
 fill: none;
 stroke: black;
 stroke-width: 1;
}
.input_text {
 fill: black;
 stroke: none;
 font-size: 12px;
}
.output_object {
 fill: none;
 stroke: red;
 stroke-width: 1;
}
.output_text {
 fill: red;
 stroke: none;
 font-size: 12px;
}
</style>
<polygon class="input_object" points="38.23232323232324
264.4065656565657 65.9848484848485 195.8901515151515
32.15909090909092 166.40151515151516"/>
<text class="input_text" x="23.541666666666668"
y="167.58838383838383">A</text>
<text class="input_text" x="70.2588383838384"
y="197.89141414141415">B</text>
<text class="input_text" x="28.787878787878793"
y="270.30303030303037">C</text>
</svg>
```

Please visualize the explanation by adding elements to the SVG diagram.
- Assign class="output_text" to any added text elements, and class="output_object" to all other added elements.
- Do not modify any elements that are originally present in the SVG.
– Do not use any animations.

Figure 8: An example prompt from the SVG generation task on plane geometry.

[Explanation]The exterior angle bisector in A intersects the extended side BC in E, the exterior angle bisector in B intersects the extended side AC in D and the exterior angle bisector in C intersects the extended side AB in F.

The three points of intersection between the exterior angle bisectors and the extended triangle sides D, E, F are collinear, that is they lie on a common line.
[EPS]%!PS-Adobe-3.0 EPSF-3.0
%%BoundingBox: 0 0 300 300
1 setlinewidth
0 0 0 setrgbcolor
newpath
38.23232323232324 35.593434343434296 moveto
65.9848484848485 104.1098484848485 lineto
32.15909090909092 133.59848484848484 lineto
closepath
stroke
/Helvetica findfont 12 scalefont setfont
0 0 0 setrgbcolor
newpath
23.541666666666668 132.41161616161617 moveto
(A) show
/Helvetica findfont 12 scalefont setfont
0 0 0 setrgbcolor
newpath
70.2588383838384 102.10858585858585 moveto
(B) show
/Helvetica findfont 12 scalefont setfont
0 0 0 setrgbcolor
newpath
28.787878787878793 29.69696969696963 moveto
(C) show
showpage
Please visualize the explanation by adding elements to the EPS diagram.
– Set the color of any added elements to red.
– Do not modify any elements that are originally present in the EPS.
– Do not use any animations.

Figure 9: An example prompt from the EPS generation task on plane geometry.

Please create a TikZ file that visualizes the molecular structure of the compound with the IUPAC name 4-butyl-2,6-dimethylmorpholine. Represent each atom as a circle, using colors to indicate atom types. The color mapping for each atom type is provided below, although not all listed types may be present in the molecule. Omit hydrogen atoms from the visualization. Depict bonds between atoms as lines, using a single line for each bond regardless of bond order.

H: #638c8c, B: #2AD52A, C: #274A4A, N: #0000FF, O: #FF0000, F: #D52092, Si: #D59E13, P: #D58600, S: #D5D500, Cl: #2AD52A, Br: #D58639, Te: #D5CD72, I: #FF00FF, Eu: #00CCD5, Lu: #00CCD5, Os: #838C8C, U: #00CCD5

As a reference, an example TikZ visualization of the compound chlorobenzene is provided below.
```
\documentclass[tikz]{standalone}
\definecolor{274A4A}{HTML}{274A4A}
\definecolor{2AD52A}{HTML}{2AD52A}
\begin{document}
\begin{tikzpicture}[x=1pt,y=1pt]
 \draw[line width=1pt] (20.00,6.22) -- (28.66,1.22);
 \draw[line width=1pt] (45.98,1.22) -- (45.98,-8.78);
 \draw[line width=1pt] (45.98,1.22) -- (37.32,6.22);
 \draw[line width=1pt] (45.98,-8.78) -- (37.32,-13.78);
 \draw[line width=1pt] (37.32,6.22) -- (28.66,1.22);
 \draw[line width=1pt] (37.32,-13.78) -- (28.66,-8.78);
 \draw[line width=1pt] (28.66,1.22) -- (28.66,-8.78);
 \filldraw[fill=2AD52A, draw=none] (20.00,6.22) circle (1.5pt);
 \filldraw[fill=274A4A, draw=none] (45.98,1.22) circle (1.5pt);
 \filldraw[fill=274A4A, draw=none] (45.98,-8.78) circle (1.5pt);
 \filldraw[fill=274A4A, draw=none] (37.32,6.22) circle (1.5pt);
 \filldraw[fill=274A4A, draw=none] (37.32,-13.78) circle (1.5pt);
 \filldraw[fill=274A4A, draw=none] (28.66,1.22) circle (1.5pt);
 \filldraw[fill=274A4A, draw=none] (28.66,-8.78) circle (1.5pt);
\end{tikzpicture}
\end{document}
```

Figure 10: An example prompt from the TikZ generation task on molecular structure.

918
919
920
921
922
923
924
925
926
927
928
929
930

---

931 Please create an SVG file that visualizes the molecular structure of the
compound with the IUPAC name 4-butyl-2,6-dimethylmorpholine.
932 Represent each atom as a circle, using colors to indicate atom types.
The color mapping for each atom type is provided below, although not
933 all listed types may be present in the molecule. Omit hydrogen atoms
934 from the visualization. Depict bonds between atoms as lines, using a
single line for each bond regardless of bond order.
935

936 H: #638c8c, B: #2AD52A, C: #274A4A, N: #0000FF, O: #FF0000, F:
#D52092, Si: #D59E13, P: #D58600, S: #D5D500, Cl: #2AD52A, Br:
937 #D58639, Te: #D5CD72, I: #FF00FF, Eu: #00CCD5, Lu: #00CCD5,
938 Os: #838C8C, U: #00CCD5

939 As a reference, an example SVG visualization of the compound
940 chlorobenzene is provided below.
`<svg xmlns="http://www.w3.org/2000/svg" viewBox="10 -29.976`
941 `67.479 59.646">`
942 `<line x1="20" y1="6.223999999999999" x2="28.66" y2="1.224"`
`stroke="black" stroke-width="1" />`
943 `<line x1="45.980999999999995" y1="1.224"`
944 `x2="45.980999999999995" y2="-8.776" stroke="black"`
`stroke-width="1" />`
945 `<line x1="45.980999999999995" y1="1.224" x2="37.32"`
946 `y2="6.223999999999999" stroke="black" stroke-width="1" />`
`<line x1="45.980999999999995" y1="-8.776" x2="37.32" y2="-13.776"`
947 `stroke="black" stroke-width="1" />`
948 `<line x1="37.32" y1="6.223999999999999" x2="28.66" y2="1.224"`
`stroke="black" stroke-width="1" />`
949 `<line x1="37.32" y1="-13.776" x2="28.66" y2="-8.776" stroke="black"`
950 `stroke-width="1" />`
`<line x1="28.66" y1="1.224" x2="28.66" y2="-8.776" stroke="black"`
951 `stroke-width="1" />`
952 `<circle cx="20" cy="6.223999999999999" r="1.5" fill="#2AD52A" />`
`<circle cx="45.980999999999995" cy="1.224" r="1.5" fill="#274A4A"`
953 `/>`
954 `<circle cx="45.980999999999995" cy="-8.776" r="1.5" fill="#274A4A"`
`/>`
955 `<circle cx="37.32" cy="6.223999999999999" r="1.5" fill="#274A4A" />`
956 `<circle cx="37.32" cy="-13.776" r="1.5" fill="#274A4A" />`
`<circle cx="28.66" cy="1.224" r="1.5" fill="#274A4A" />`
957 `<circle cx="28.66" cy="-8.776" r="1.5" fill="#274A4A" />`
958 `</svg>`

959 Figure 11: An example prompt from the SVG generation task on molecular structure.
960
961
962
963
964
965
966
967
968
969
970
971

Please create an EPS file that visualizes the molecular structure of the compound with the IUPAC name 4-butyl-2,6-dimethylmorpholine. Represent each atom as a circle, using colors to indicate atom types. The color mapping for each atom type is provided below, although not all listed types may be present in the molecule. Omit hydrogen atoms from the visualization. Depict bonds between atoms as lines, using a single line for each bond regardless of bond order.

H: [0.39, 0.55, 0.55], B: [0.16, 0.84, 0.16], C: [0.15, 0.29, 0.29], N: [0.0, 0.0, 1.0], O: [1.0, 0.0, 0.0], F: [0.84, 0.13, 0.57], Si: [0.84, 0.62, 0.07], P: [0.84, 0.53, 0.0], S: [0.84, 0.84, 0.0], Cl: [0.16, 0.84, 0.16], Br: [0.84, 0.53, 0.22], Te: [0.84, 0.8, 0.45], I: [1.0, 0.0, 1.0], Eu: [0.0, 0.8, 0.84], Lu: [0.0, 0.8, 0.84], Os: [0.51, 0.55, 0.55], U: [0.0, 0.8, 0.84]

As a reference, an example EPS visualization of the compound chlorobenzene is provided below.

```
%!PS-Adobe-3.0 EPSF-3.0
%%BoundingBox: 10 -29 77 29
%%Creator: EPS Generator
%%EndComments

1 setlinewidth
0 0 0 setrgbcolor

newpath
20 6.223999999999999 moveto
28.66 1.224 lineto
stroke

newpath
45.980999999999995 1.224 moveto
37.32 6.223999999999999 lineto
stroke

newpath
37.32 6.223999999999999 moveto
28.66 1.224 lineto
stroke

newpath
28.66 1.224 moveto
28.66 -8.776 lineto
stroke

newpath
28.66 -8.776 moveto
37.32 -13.776 lineto
stroke

newpath
37.32 -13.776 moveto
45.980999999999995 -8.776 lineto
stroke

newpath
45.980999999999995 -8.776 moveto
45.980999999999995 1.224 lineto
stroke

0.16 0.84 0.16 setrgbcolor
newpath
20 6.223999999999999 1.5 0 360 arc
closepath
fill

0.15 0.29 0.29 setrgbcolor
newpath
45.980999999999995 1.224 1.5 0 360 arc
closepath
fill

0.15 0.29 0.29 setrgbcolor
newpath
45.980999999999995 -8.776 1.5 0 360 arc
closepath
fill

0.15 0.29 0.29 setrgbcolor
newpath
37.32 6.223999999999999 1.5 0 360 arc
closepath
fill

0.15 0.29 0.29 setrgbcolor
newpath
37.32 -13.776 1.5 0 360 arc
closepath
fill

0.15 0.29 0.29 setrgbcolor
newpath
28.66 1.224 1.5 0 360 arc
closepath
fill

0.15 0.29 0.29 setrgbcolor
newpath
28.66 -8.776 1.5 0 360 arc
closepath
fill

showpage
```

Figure 12: An example prompt from the EPS generation task on molecular structure.

## C    LIMITATIONS OF OUR AUTOMATIC EVALUATION FRAMEWORK

Our automatic evaluation framework for the plane geometry visualization task assesses whether the necessary elements are present in the output, but it does not penalize the inclusion of unnecessary elements. In the example shown in Figure 13, the output from Gemini 2.5 Flash reasoning includes an irrelevant straight line, yet it is still considered correct. We do not penalize unnecessary elements because it is often non-trivial to determine whether an additional element is truly unnecessary. For instance, the output from Gemini 2.5 Flash reasoning in Figure 13 includes circles not anticipated in the ground-truth, but these represent intersections and the circle center, and they do not hinder the explanation.

Because in practical scenarios it is usually easier for humans to remove unnecessary elements than to create necessary ones from scratch, we do not currently view this limitation as a major issue. However, enabling the framework to identify and evaluate such extraneous content remains an important direction for future work.

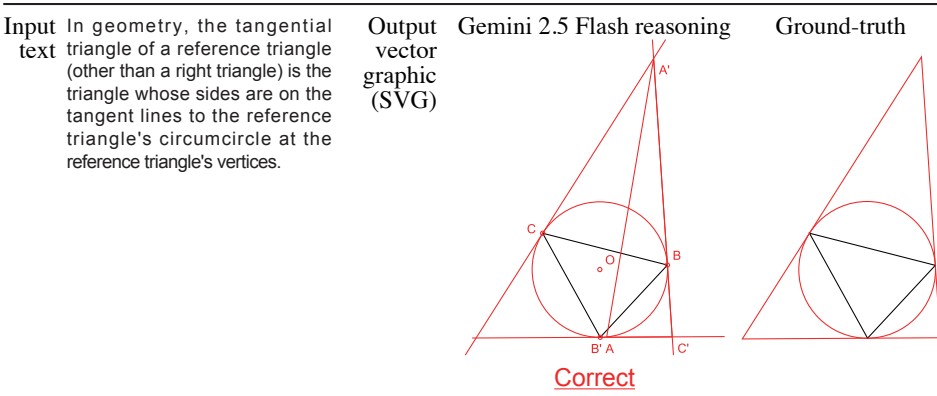

Figure 13: An example judged correct despite an unrelated line.[7]

---

[7]The input text and the ground-truth vector graphic are from (Wikipedia contributors, 2025m; Kmhkmh, 2019b).

## D  ADDITIONAL EXAMPLES OF GENERATED VECTOR GRAPHICS

### D.1  EXAMPLES GENERATED BY FINE-TUNED MODELS

Figure 14 shows examples generated by two fine-tuned models, AutomaTikZ (Belouadi et al., 2024a) and TikZero+ (Belouadi et al., 2025). The top example is the only case where TikZero+ produces a structurally correct vector graphic. However, in all other cases, the models fail to generate structurally correct vector graphics. In the plane geometry task, they cannot follow simple instructions such as "Set the color of any added elements to red" or "Do not modify any elements that are originally present in the TikZ." In the molecular structure task, the models generate entirely invalid vector graphics. These results reveal the limitations of fine-tuned models.

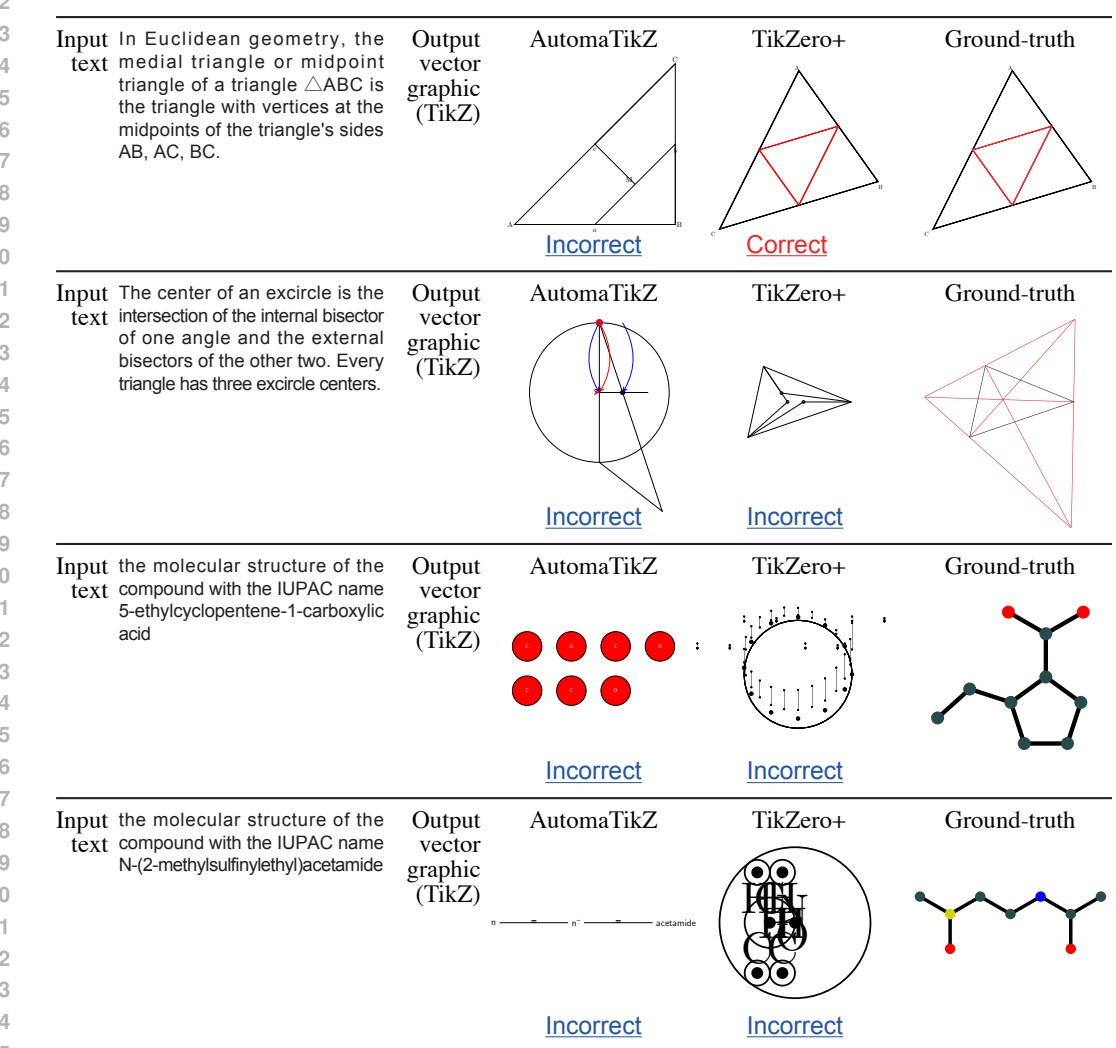

Figure 14: Examples generated by fine-tuned models.[8]

---

[8]The input text and the ground-truth vector graphic are from (Wikipedia contributors, 2025j;g; Braindrain0000, 2006; Inductiveload, 2007a). The molecular structure data is from (National Center for Biotechnology Information, 2025k;n).

## D.2 IMPACT OF REASONING

We show in Figures 15 and 16 that enabling reasoning allows LLMs to output the correct structure. Without reasoning, they struggle to generate even simple structures.

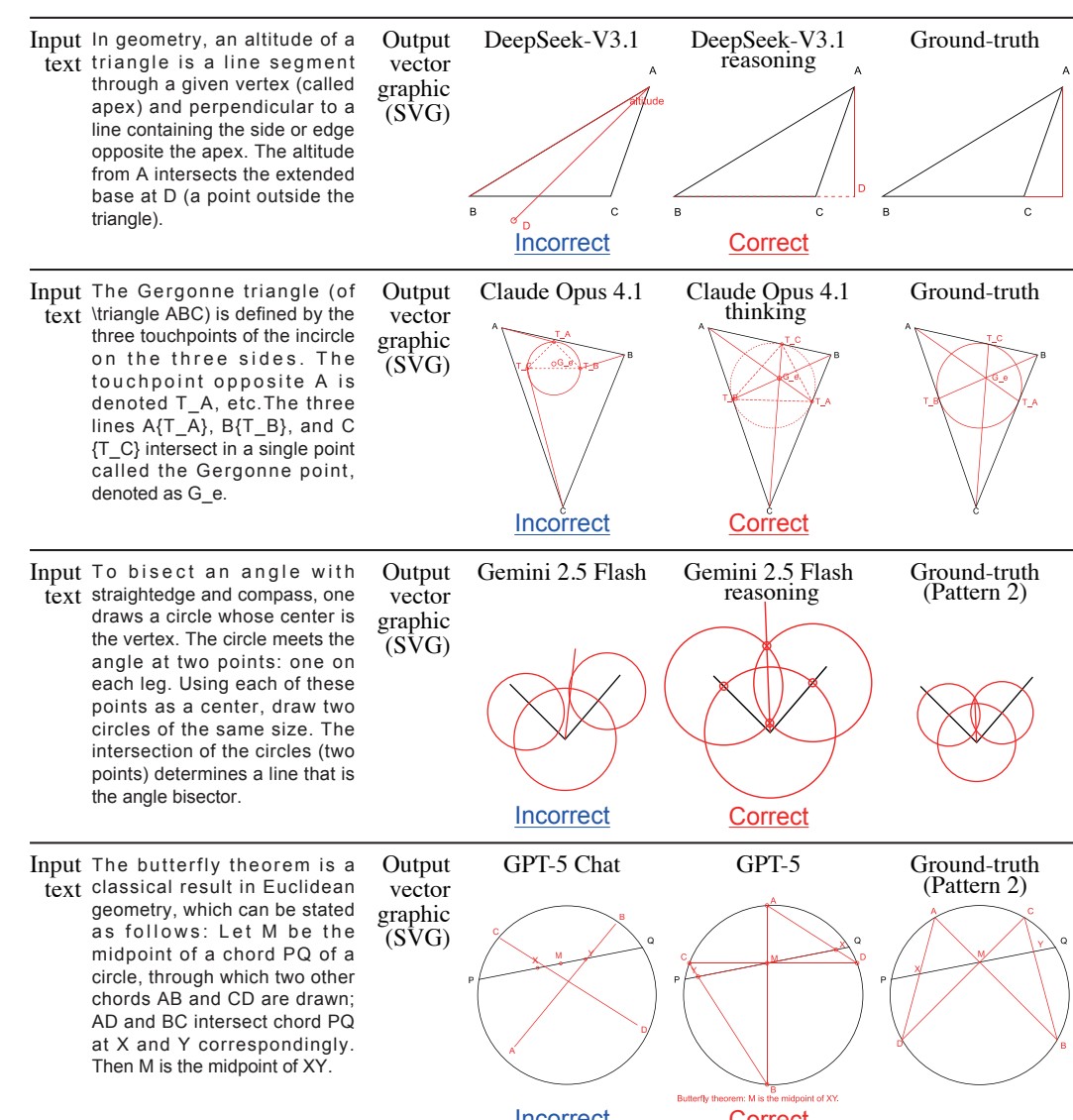

Figure 15: Examples where enabling reasoning allows LLMs to generate the correct structure in the plane geometry SVG geneartion task.[9]

---

[9]The input texts and the ground-truth vector graphics are from (Wikipedia contributors, 2025a;g;c;d; PegasusRoe, 2007; Inductiveload, 2007c; Ixnay, 2007; Gustavb, 2006).

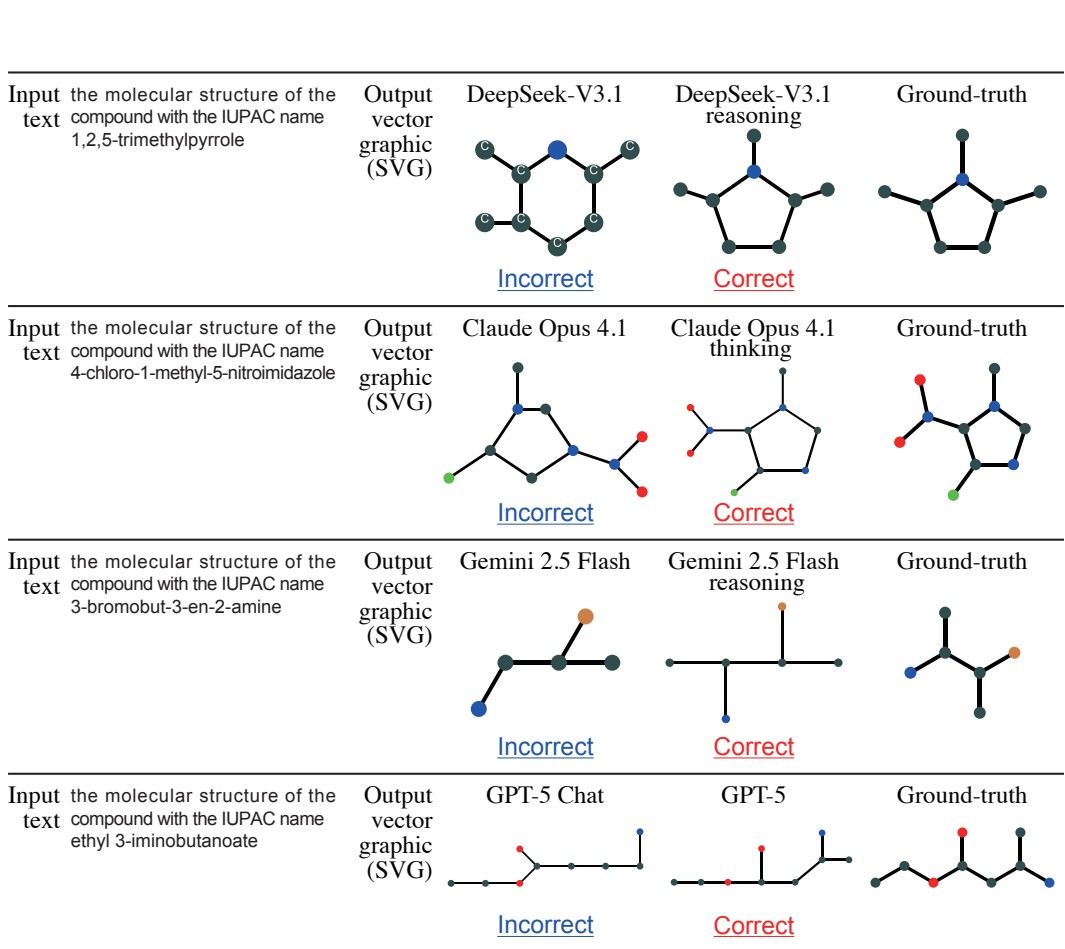

Figure 16: Examples where enabling reasoning allows LLMs to generate the correct structure in the molecular structure SVG geneartion task.[10]

---

[10]The molecular structure data is from (National Center for Biotechnology Information, 2025m;c;e;i).

## D.3 IMPACT OF FORMAT

Figures 17 and 18 present examples where LLMs produce correct structures in SVG format but fail in TikZ and EPS formats. Although the input text is identical, the results clearly vary depending on the output format.

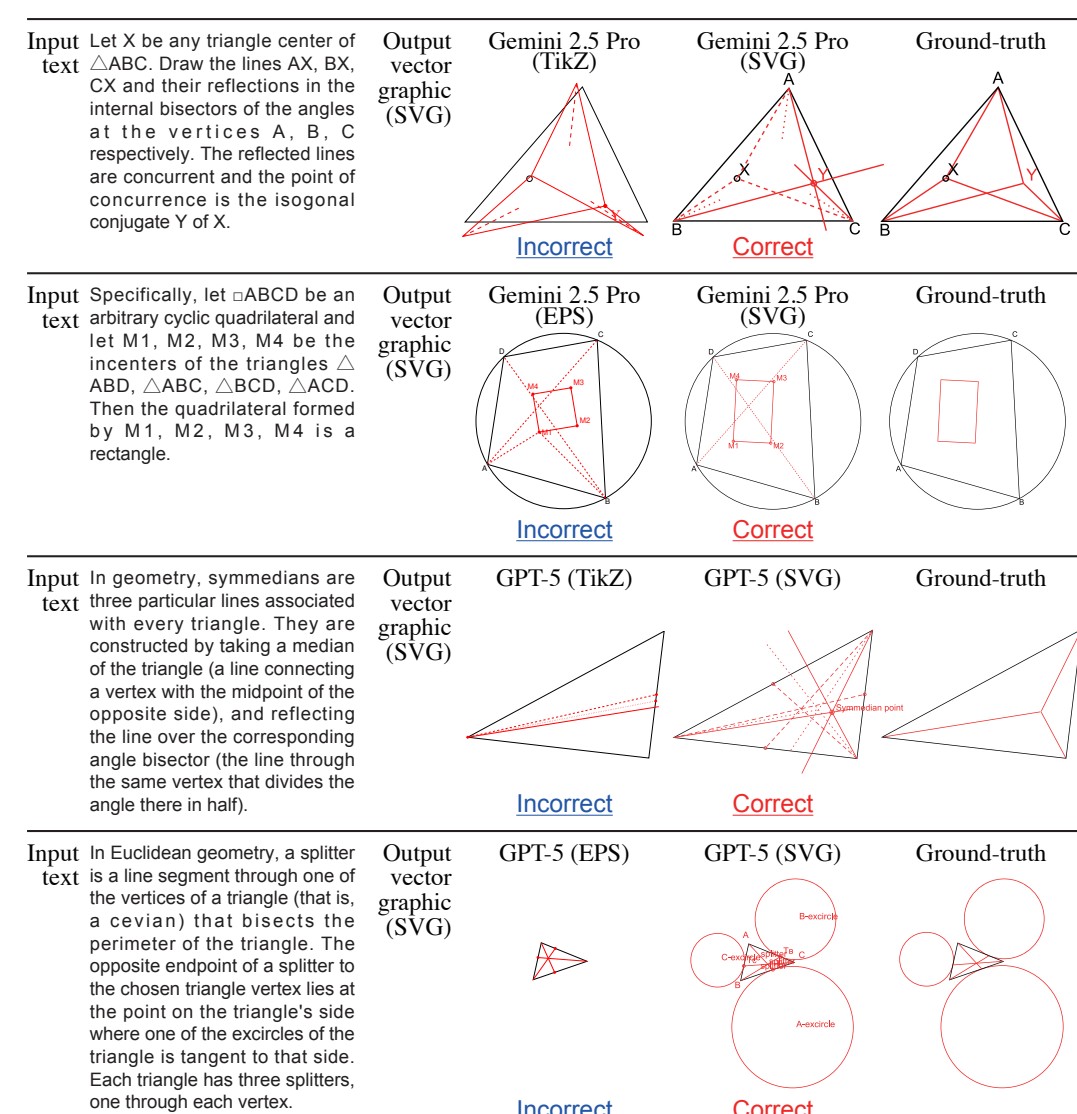

Figure 17: Examples where LLMs produce correct structures in SVG format but fail in TikZ and EPS formats in the plane geometry task.[11]

---

[11]The input texts and the ground-truth vector graphics are from (Wikipedia contributors, 2025h;i;l;k; Rocchini, 2008; Kmhkmh, 2024; 2016; Inductiveload, 2007b).

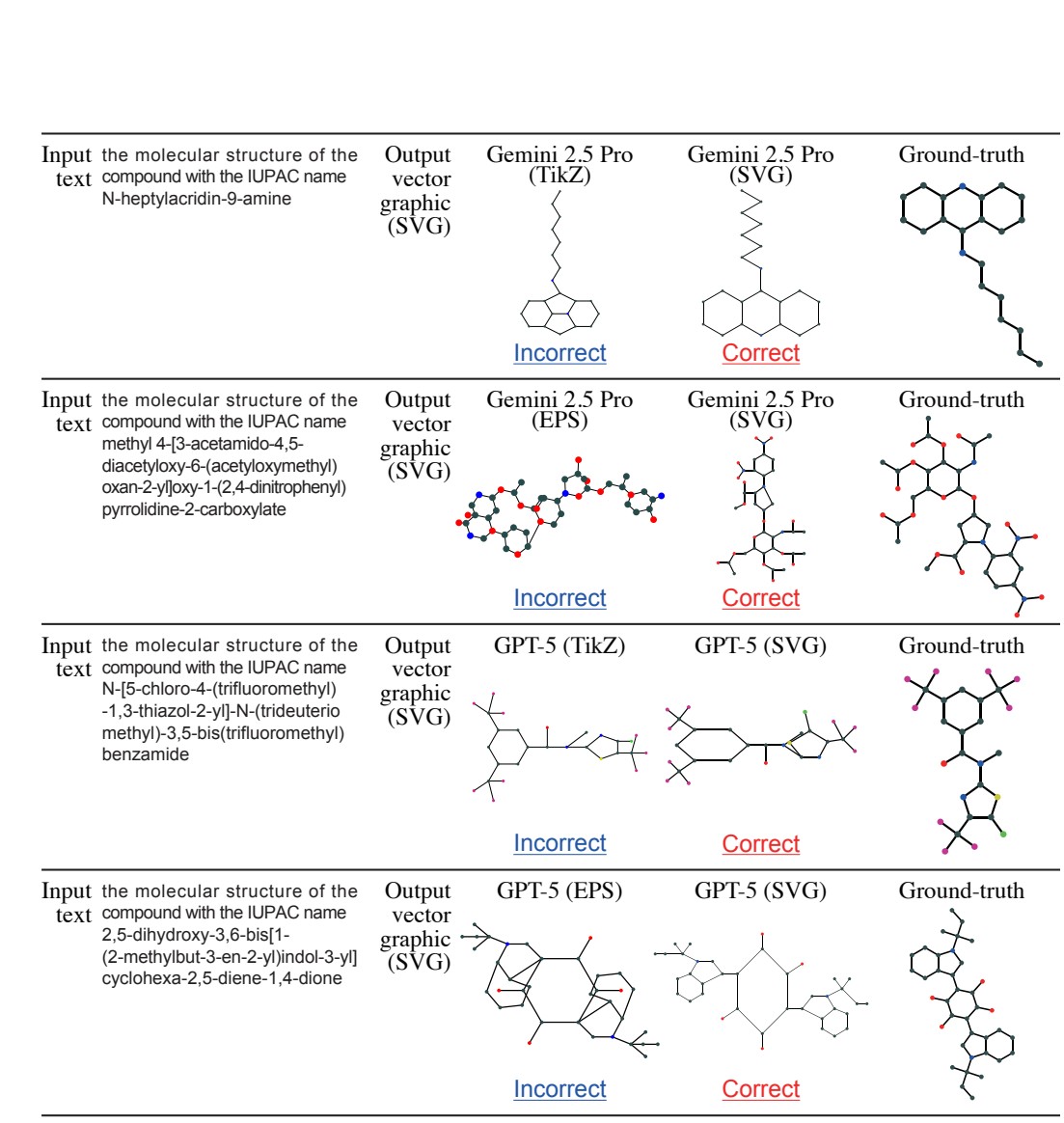

Figure 18: Examples where LLMs produce correct structures in SVG format but fail in TikZ and EPS formats in the molecular structure task.[12]

---

[12]The molecular structure data is from (National Center for Biotechnology Information, 2025d;h;b;a).

## D.4 COMPARISON OF ZERO-SHOT PROMPTING METHODS

We present results obtained by applying zero-shot prompting methods to Gemini 2.5 Pro in Figures 19, 20, 21, and 22. When using methods other than Zero-shot CoT, Gemini 2.5 Pro produces an explanation prior to generating vector graphics, and we include that explanation as well. In the case of Plan-and-Solve prompting, the output explanation "C3 will also be in corner B, tangent to C2" is incorrect, revealing that Gemini 2.5 Pro fails to capture the relationships. In the case of Step-Back prompting, the output explanation reveals that Gemini 2.5 Pro captures the necessary elements and relationships, but overlooks the importance of mathematical reasoning. In contrast, by applying our LOOP, Gemini 2.5 Pro captures the necessary elements and relationships and recognizes the importance of mathematical reasoning, enabling it to generate structurally correct vector graphics.

---

**Input text and ground-truth**

Input text — In geometry, the six circles theorem relates to a chain of six circles together with a triangle, such that each circle is tangent to two sides of the triangle and also to the preceding circle in the chain. The chain closes, in the sense that the sixth circle is always tangent to the first circle.

Ground-truth

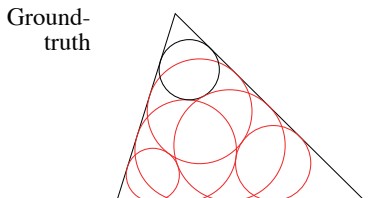

---

**Gemini 2.5 Pro w/ Zero-shot CoT**

Output explanation — (There is no output explanation.)

Output vector graphic (SVG)

Incorrect

Figure 19: Example output generated by applying Zero-shot CoT prompting to Gemini 2.5 Pro.[13]

---

[13]The input text and the ground-truth vector graphic are from (Wikipedia contributors, 2022; Rocchini, 2010).

**Gemini 2.5 Pro w/ Plan-and-Solve**

Output explanation
An explanation of the six circles theorem has been provided, and it needs to be visualized by adding elements to the given SVG.

### Plan:
1. **Deconstruct the Theorem**: The theorem describes a chain of six circles (C1 to C6) associated with a triangle.
    * There are three pairs of circles, with each pair nestled into one corner of the triangle.
    * Circles in the same corner are tangent to each other and to the two adjacent sides of the triangle.
    * The chain follows a sequence around the triangle's vertices: A -> B -> C -> A.
    * Let's say the given circle, C1, is in corner A.
    * The next circle, C2, will be in corner B and tangent to C1.
    * C3 will also be in corner B, tangent to C2.
    * C4 will be in corner C, tangent to C3.
    * C5 will also be in corner C, tangent to C4.
    * C6 will be in corner A, tangent to C5.
    * The theorem's conclusion is that C6 will also be tangent to the first circle, C1, thus closing the chain.

2. **Analyze the Input SVG**:
    * We have a triangle. Let's label its vertices A (top), B (bottom-left), and C (bottom-right).
    * We have one circle, which we will label C1. It is located in corner A.

3. **Add Visual Elements**:
    * Add the remaining five circles (C2, C3, C4, C5, C6) with the class `output_object`.
    * Place them according to the theorem's rules, ensuring they appear tangent to the correct triangle sides and preceding circles. The positions and radii will be estimated to create a visually plausible representation of the theorem.
    * C1 and C6 will be in corner A.
    * C2 and C3 will be in corner B.
    * C4 and C5 will be in corner C.
    * The crucial part is to draw the chain such that C6, which is tangent to C5, is also tangent to C1.
    * Add text labels "C1" through "C6" for each circle, assigning them the class `output_text`.

### Execution:
The following SVG code adds the five circles and six labels to illustrate the theorem. The positions and sizes of the new circles have been chosen to create a clear and accurate visual representation of the tangency relationships described.

Output vector graphic (SVG)

Incorrect

Figure 20: Example output generated by applying Plan-and-Solve prompting to Gemini 2.5 Pro.

1458
1459
1460
1461
1462
1463
1464
1465
1466
1467
1468
1469
1470
1471
1472
1473

---

**Gemini 2.5 Pro w/ Step-Back**

---

Output explanation: An explanation of the six circles theorem has been provided. This theorem describes a closed chain of six circles, where each circle is tangent to two sides of a triangle and the preceding circle in the chain. The provided SVG shows the triangle and the first circle of such a chain.

To visualize the theorem, we will add the remaining five circles to complete the chain.

1. **Circle C1:** The initial circle is provided, located in the top corner of the triangle.
2. **Circle C2:** We add the second circle, C2, in the bottom-right corner, making it tangent to the two sides of the triangle at that corner and also to circle C1.
3. **Circle C3:** The third circle, C3, is placed in the bottom-left corner, tangent to the two sides there and to circle C2.
4. **Circle C4:** The fourth circle, C4, is placed back in the top corner. It is tangent to the same two sides as C1, and also to circle C3.
5. **Circle C5:** The fifth circle, C5, is placed in the bottom-right corner, tangent to the sides and to circle C4.
6. **Circle C6:** The sixth and final circle, C6, is placed in the bottom-left corner, tangent to the sides and to circle C5.

The remarkable property of the theorem is that this chain always closes perfectly: circle C6 will not only be tangent to C5 but will also be tangent to the very first circle, C1, completing the loop. The visualization below illustrates this closed chain.

Output vector graphic (SVG)

Incorrect

---

Figure 21: Example output generated by applying Step-Back prompting to Gemini 2.5 Pro.

1498
1499
1500
1501
1502
1503
1504
1505
1506
1507
1508
1509
1510
1511

1512
1513
1514
1515
1516
1517
1518
1519
1520
1521
1522
1523
1524

**Gemini 2.5 Pro w/ LOOP (ours)**

Output explanation — An elegant theorem in geometry, the six circles theorem, describes a fascinating property of a chain of circles within a triangle. Starting with a circle tangent to two sides of the triangle, a sequence of five more circles is constructed. Each new circle is tangent to two sides of the triangle and to the previous circle in the chain. The remarkable result is that this chain always forms a closed loop, with the sixth circle being perfectly tangent to the first one.

This visualization completes the chain described by the theorem, starting from the single provided circle.

1. **Information extraction**: The core components are a triangle and a chain of six circles. The initial SVG provides the triangle and the first circle (let's call it C1). C1 is tangent to two sides of the triangle at one vertex. The goal is to construct the remaining five circles, C2 through C6, following the theorem's rules.
2. **Relationship extraction**: The chain proceeds sequentially around the vertices of the triangle.
    *   C1 is tangent to sides AB and AC (at vertex A).
    *   C2 must be tangent to sides AC and BC (at vertex C) and also to C1.
    *   C3 must be tangent to sides BC and AB (at vertex B) and also to C2.
    *   C4 must be tangent to sides AB and AC (at vertex A) and also to C3.
    *   C5 must be tangent to sides AC and BC (at vertex C) and also to C4.
    *   C6 must be tangent to sides BC and AB (at vertex B) and also to C5.
    The theorem guarantees that the resulting C6 will also be tangent to the starting circle, C1.
3. **Mathematical reasoning**: The center of any circle tangent to two sides of the triangle must lie on the angle bisector of the corresponding vertex. The condition that the new circle must also be tangent to the previous circle provides a second constraint. Combining these two conditions leads to a solvable quadratic equation for the position of the new circle's center along the angle bisector. This process is applied iteratively to find the center and radius of each subsequent circle in the chain (C2, C3, C4, C5, and C6). The final calculation confirms the theorem: the distance between the centers of C6 and C1 is indeed equal to the sum of their radii, proving their tangency.
4. **Code generation**: Based on the calculated centers and radii, five `<circle>` elements are added to the SVG to represent C2 through C6. Additionally, `<text>` elements are added to label each circle from C1 to C6, clarifying the sequence of the chain.

Output vector graphic (SVG)

Correct

Figure 22: Example output generated by applying our LOOP to Gemini 2.5 Pro.

# E    ADDITIONAL EXPERIMENTS

## E.1    COMPLEMENTARY AUTOMATIC METRICS

Our evaluation in the main paper relies solely on a single binary metric (Accuracy), which provides an overly coarse assessment because near-misses and completely incorrect outputs are both scored as 0. To address this limitation, we employ additional evaluation metrics. Following TikZero+ (Belouadi et al., 2025), we use five image- and code-similarity metrics: DreamSim (**DSim**), Kernel Inception Distance (**KID**), CLIPScore (**CLIP**), CrystalBLEU (**cBLEU**), and Extended Edit Distance (**EED**). We additionally use **Coverage** as an extra metric, which considers an output correct if it contains the required elements, regardless of their spatial arrangement. As a comprehensive evaluation, we compute an **Average** score, defined as the mean of all metrics after applying min-max normalization. When applying min-max normalization, we normalize indicators where higher values are better such that the minimum becomes 0 and the maximum becomes 100. For indicators where lower values are better, we normalize them such that the minimum becomes 100 and the maximum becomes 0. We present these metrics in Tables 7, 8, 9, 10, 11, and 12.

Focusing on the **Average** scores, we consistently observe that (1) reasoning models outperform non-reasoning models, and (2) our proposed LOOP improves the performance in most cases.

## E.2    ADDITIONAL MODELS

To demonstrate the versatility of our LOOP, we apply it to two additional reasoning models (DeepSeek-V3.2 Reasoning and Claude Opus 4.1 Thinking) and two additional non-reasoning models (Gemini 2.5 Flash and GPT-5 Chat). The results are shown in Tables 7, 8, 9, 10, 11, and 12. Our LOOP achieves improved performance in many cases, clearly demonstrating its robustness.

## E.3    IMPACT OF RESAMPLING ON FINE-TUNED MODELS

The impact of resampling is considered a possible reason for the low performance of the fine-tuned models. In our experiments, we evaluate each model using only a single generation attempt, and any outputs that resulted in compilation errors are considered incorrect. This evaluation procedure is applied uniformly across all models. However, the fine-tuned models are expected to be used under the assumption that resampling continues until no compilation errors occur. To investigate the impact of resampling, we allow up to 10 resampling attempts for each sample until a compilable output is obtained. As shown in Tables 7 and 10, resampling improves performance, and notably, TikZero+ achieves performance comparable to non-reasoning models.

Table 7: Complementary metrics for plane geometry TikZ generation.

| Model | Acc↑ | DSim↑ | KID↓ | CLIP↑ | cBLEU↑ | EED↓ | Coverage↑ | Average↑ |
|---|---|---|---|---|---|---|---|---|
| **Fine-tuned models** | | | | | | | | |
| AutomaTikZ | 0.0 | 36.0 | 109.4 | 3.2 | 1.1 | 63.5 | 3.6 | 17.3 |
| TikZero+ | 0.9 | 49.0 | 55.5 | 13.5 | 5.6 | 51.4 | 12.7 | 32.6 |
| AutomaTikZ resampling | 0.0 | 56.4 | 12.5 | 16.1 | 1.2 | 61.1 | 13.6 | 33.7 |
| DeTikZify resampling | 0.9 | 67.1 | 5.8 | 27.2 | 7.4 | 49.6 | 30.0 | 46.7 |
| **Non-reasoning models** | | | | | | | | |
| DeepSeek-V3 | 10.0 | 56.1 | 42.4 | 28.3 | 11.4 | 44.0 | 42.7 | 48.4 |
| DeepSeek-V3.1 | 11.8 | 53.8 | 49.6 | 23.8 | 10.1 | 45.2 | 41.8 | 45.9 |
| Claude Opus 4.1 | 14.5 | 60.8 | 31.4 | 32.6 | 9.5 | 46.1 | 51.8 | 51.8 |
| Gemini 2.0 Flash | 7.3 | 71.1 | 9.9 | 39.9 | 19.4 | 37.4 | 50.0 | 62.5 |
| Gemini 2.5 Flash non-reasoning | 12.7 | 48.4 | 70.7 | 17.7 | 6.9 | 50.7 | 30.0 | 37.5 |
| GPT-4.1 | 10.9 | 50.9 | 54.3 | 25.0 | 8.3 | 47.5 | 39.1 | 43.1 |
| GPT-5 Chat | 12.7 | 54.9 | 48.6 | 28.1 | 11.4 | 44.1 | 38.2 | 47.5 |
| **Rasoning models** | | | | | | | | |
| DeepSeek-R1 | 28.2 | 74.6 | 7.6 | 39.9 | 21.5 | 33.8 | 54.5 | 69.6 |
| DeepSeek-V3.1 reasoning | 23.6 | 63.5 | 29.5 | 31.6 | 11.3 | 40.8 | 40.9 | 54.4 |
| Claude Opus 4.1 thinking | 20.0 | 66.9 | 18.8 | 43.8 | 12.4 | 41.8 | 64.5 | 61.4 |
| Gemini 2.5 Flash reasoning | 30.0 | 55.7 | 49.6 | 25.2 | 8.5 | 47.8 | 44.5 | 48.7 |
| Gemini 2.5 Pro | 50.0 | 67.1 | 25.0 | 34.5 | 9.7 | 43.3 | 56.4 | 61.6 |
| o4-mini | 48.2 | 74.9 | 7.7 | 46.5 | 17.4 | 37.4 | 70.0 | 73.7 |
| GPT-5 | 54.5 | 69.7 | 13.3 | 44.4 | 9.3 | 43.4 | 66.4 | 66.9 |
| **Prompting** | | | | | | | | |
| Gemini 2.5 Pro | 50.0 | 67.1 | 25.0 | 34.5 | 9.7 | 43.3 | 56.4 | 61.6 |
|   w/ Zero-shot CoT | 39.1 | 64.3 | 28.5 | 37.4 | 9.8 | 43.7 | 52.7 | 58.8 |
|   w/ Plan-and-Solve | 39.1 | 60.7 | 37.3 | 28.2 | 9.1 | 45.9 | 48.2 | 54.1 |
|   w/ Step-Back | 33.6 | 57.2 | 45.3 | 28.6 | 8.1 | 47.0 | 42.7 | 50.3 |
|   w/ LOOP (ours) | 65.5 | 77.4 | 6.4 | 47.1 | 14.7 | 38.1 | 67.3 | 75.6 |
| GPT-5 | 54.5 | 69.7 | 13.3 | 44.4 | 9.3 | 43.4 | 66.4 | 66.9 |
|   w/ Zero-shot CoT | 58.2 | 72.5 | 12.4 | 44.0 | 10.1 | 43.3 | 66.4 | 68.6 |
|   w/ Plan-and-Solve | 61.8 | 74.0 | 8.1 | 49.5 | 11.6 | 41.4 | 71.8 | 72.9 |
|   w/ Step-Back | 55.5 | 70.0 | 13.8 | 41.7 | 9.6 | 43.5 | 68.2 | 67.0 |
|   w/ LOOP (ours) | 70.0 | 80.6 | 3.1 | 52.3 | 13.4 | 39.0 | 82.7 | 79.7 |
| Gemini 2.5 Flash non-reasoning | 12.7 | 48.4 | 70.7 | 17.7 | 6.9 | 50.7 | 30.0 | 37.5 |
|   w/ LOOP (ours) | 33.6 | 72.3 | 11.4 | 38.6 | 11.4 | 42.6 | 60.9 | 63.3 |
| GPT-5 Chat | 12.7 | 54.9 | 48.6 | 28.1 | 11.4 | 44.1 | 38.2 | 47.5 |
|   w/ LOOP (ours) | 15.5 | 66.8 | 15.0 | 47.2 | 13.5 | 40.8 | 58.2 | 61.3 |
| DeepSeek-V3.2 reasoning | 14.5 | 63.6 | 26.8 | 32.8 | 11.9 | 42.1 | 43.6 | 53.6 |
|   w/ LOOP (ours) | 22.7 | 70.9 | 12.1 | 37.6 | 16.0 | 36.7 | 61.8 | 64.7 |
| Claude Opus 4.1 thinking | 20.0 | 66.9 | 18.8 | 43.8 | 12.4 | 41.8 | 64.5 | 61.4 |
|   w/ LOOP (ours) | 32.7 | 84.5 | 0.2 | 58.3 | 19.8 | 34.5 | 87.3 | 80.2 |

Table 8: Complementary metrics for plane geometry SVG generation.

| Model | Acc↑ | DSim↑ | KID↓ | CLIP↑ | cBLEU↑ | EED↓ | Coverage↑ | Average↑ |
|---|---|---|---|---|---|---|---|---|
| **Non-reasoning models** | | | | | | | | |
| DeepSeek-V3 | 5.5 | 80.8 | 3.5 | 68.2 | 21.9 | 32.8 | 89.1 | 77.9 |
| DeepSeek-V3.1 | 6.4 | 81.1 | 2.4 | 68.5 | 21.0 | 33.6 | 86.4 | 77.2 |
| Claude Opus 4.1 | 12.7 | 81.8 | 3.4 | 74.8 | 15.0 | 37.1 | 96.4 | 77.2 |
| Gemini 2.0 Flash | 5.5 | 83.1 | 1.5 | 59.7 | 26.9 | 29.8 | 79.1 | 78.7 |
| Gemini 2.5 Flash non-reasoning | 9.1 | 86.5 | 1.0 | 61.4 | 22.7 | 31.4 | 84.5 | 78.7 |
| GPT-4.1 | 10.0 | 82.9 | 4.1 | 73.2 | 16.9 | 38.5 | 87.3 | 75.9 |
| GPT-5 Chat | 10.0 | 82.9 | 4.0 | 74.6 | 21.2 | 33.5 | 81.8 | 78.7 |
| **Rasoning models** | | | | | | | | |
| DeepSeek-R1 | 40.9 | 84.1 | 0.9 | 59.2 | 22.9 | 30.8 | 62.7 | 80.4 |
| DeepSeek-V3.1 reasoning | 39.1 | 83.1 | 1.1 | 65.2 | 20.8 | 32.4 | 63.6 | 79.7 |
| Claude Opus 4.1 thinking | 23.6 | 81.6 | 3.8 | 77.8 | 14.5 | 36.9 | 97.3 | 79.5 |
| Gemini 2.5 Flash reasoning | 55.5 | 87.8 | 0.7 | 65.2 | 19.7 | 36.1 | 84.5 | 85.4 |
| Gemini 2.5 Pro | 62.7 | 88.4 | 0.7 | 72.5 | 20.7 | 31.8 | 90.0 | 90.3 |
| o4-mini | 62.7 | 88.7 | 1.5 | 64.0 | 20.6 | 32.9 | 80.9 | 87.1 |
| GPT-5 | 75.5 | 83.5 | 4.4 | 76.3 | 17.4 | 35.5 | 94.5 | 90.1 |
| **Prompting** | | | | | | | | |
| Gemini 2.5 Pro | 62.7 | 88.4 | 0.7 | 72.5 | 20.7 | 31.8 | 90.0 | 90.3 |
|   w/ Zero-shot CoT | 66.4 | 81.4 | 2.3 | 64.6 | 17.7 | 33.2 | 90.0 | 86.1 |
|   w/ Plan-and-Solve | 69.1 | 83.9 | 1.3 | 69.0 | 17.8 | 32.5 | 91.8 | 88.4 |
|   w/ Step-Back | 64.5 | 81.7 | 2.4 | 66.3 | 17.0 | 33.4 | 91.8 | 86.0 |
|   w/ LOOP (ours) | 80.9 | 84.6 | 1.1 | 62.6 | 19.2 | 31.6 | 93.6 | 90.6 |
| GPT-5 | 75.5 | 83.5 | 4.4 | 76.3 | 17.4 | 35.5 | 94.5 | 90.1 |
|   w/ Zero-shot CoT | 80.0 | 84.6 | 3.9 | 74.4 | 16.0 | 36.3 | 96.4 | 90.2 |
|   w/ Plan-and-Solve | 77.3 | 82.1 | 5.6 | 76.7 | 15.0 | 37.4 | 95.5 | 88.6 |
|   w/ Step-Back | 75.5 | 82.9 | 4.7 | 75.3 | 16.6 | 35.8 | 94.5 | 89.3 |
|   w/ LOOP (ours) | 80.0 | 84.8 | 3.0 | 74.3 | 17.2 | 35.3 | 94.5 | 90.9 |
| Gemini 2.5 Flash non-reasoning | 9.1 | 86.5 | 1.0 | 61.4 | 22.7 | 31.4 | 84.5 | 78.7 |
|   w/ LOOP (ours) | 45.5 | 86.7 | 0.8 | 64.8 | 21.1 | 33.6 | 67.3 | 82.0 |
| GPT-5 Chat | 10.0 | 82.9 | 4.0 | 74.6 | 21.2 | 33.5 | 81.8 | 78.7 |
|   w/ LOOP (ours) | 21.8 | 83.7 | 3.5 | 73.4 | 18.9 | 34.5 | 86.4 | 80.0 |
| DeepSeek-V3.2 reasoning | 25.5 | 84.1 | 1.3 | 63.3 | 25.0 | 31.3 | 73.6 | 81.0 |
|   w/ LOOP (ours) | 48.2 | 87.1 | 0.8 | 55.0 | 27.5 | 28.8 | 68.2 | 85.2 |
| Claude Opus 4.1 thinking | 23.6 | 81.6 | 3.8 | 77.8 | 14.5 | 36.9 | 97.3 | 79.5 |
|   w/ LOOP (ours) | 30.9 | 83.3 | 2.1 | 71.5 | 15.0 | 36.1 | 96.4 | 80.4 |

Table 9: Complementary metrics for plane geometry EPS generation.

| Model | Acc↑ | DSim↑ | KID↓ | CLIP↑ | cBLEU↑ | EED↓ | Coverage↑ | Average↑ |
|---|---|---|---|---|---|---|---|---|
| **Non-reasoning models** | | | | | | | | |
| DeepSeek-V3 | 7.3 | 65.7 | 19.0 | 37.5 | 12.9 | 40.9 | 50.9 | 56.2 |
| DeepSeek-V3.1 | 9.1 | 66.9 | 16.9 | 35.5 | 8.6 | 43.8 | 55.5 | 54.4 |
| Claude Opus 4.1 | 20.9 | 80.8 | 2.1 | 54.1 | 10.1 | 43.4 | 84.5 | 69.1 |
| Gemini 2.0 Flash | 1.8 | 63.4 | 23.1 | 34.2 | 19.3 | 36.6 | 43.6 | 56.9 |
| Gemini 2.5 Flash non-reasoning | 5.5 | 60.4 | 29.3 | 29.9 | 11.3 | 44.0 | 39.1 | 49.3 |
| GPT-4.1 | 14.5 | 69.2 | 11.8 | 44.9 | 13.1 | 46.3 | 62.7 | 60.8 |
| GPT-5 Chat | 7.3 | 72.5 | 7.6 | 45.9 | 16.6 | 39.7 | 60.9 | 63.7 |
| **Rasoning models** | | | | | | | | |
| DeepSeek-R1 | 39.1 | 85.2 | 1.7 | 45.9 | 18.4 | 32.7 | 59.1 | 74.6 |
| DeepSeek-V3.1 reasoning | 27.3 | 77.5 | 4.4 | 38.4 | 14.0 | 36.0 | 51.8 | 65.3 |
| Claude Opus 4.1 thinking | 17.3 | 81.6 | 1.5 | 58.2 | 17.7 | 38.7 | 79.1 | 73.6 |
| Gemini 2.5 Flash reasoning | 41.8 | 79.9 | 2.8 | 48.3 | 14.7 | 45.8 | 65.5 | 70.5 |
| Gemini 2.5 Pro | 56.4 | 83.4 | 0.7 | 51.2 | 17.2 | 38.2 | 76.4 | 79.1 |
| o4-mini | 55.5 | 86.1 | 0.3 | 52.9 | 19.6 | 35.9 | 76.4 | 81.5 |
| GPT-5 | 66.4 | 87.0 | -0.6 | 57.6 | 19.3 | 37.9 | 78.2 | 84.2 |
| **Prompting** | | | | | | | | |
| Gemini 2.5 Pro | 56.4 | 83.4 | 0.7 | 51.2 | 17.2 | 38.2 | 76.4 | 79.1 |
| w/ Zero-shot CoT | 61.8 | 86.6 | -0.4 | 53.9 | 17.1 | 39.5 | 78.2 | 81.2 |
| w/ Plan-and-Solve | 66.4 | 84.3 | 0.8 | 52.9 | 17.3 | 38.4 | 75.5 | 81.2 |
| w/ Step-Back | 59.1 | 84.1 | 0.1 | 50.1 | 16.2 | 39.0 | 73.6 | 78.5 |
| w/ LOOP (ours) | 62.7 | 85.9 | -0.3 | 50.2 | 18.4 | 36.1 | 74.5 | 81.4 |
| GPT-5 | 66.4 | 87.0 | -0.6 | 57.6 | 19.3 | 37.9 | 78.2 | 84.2 |
| w/ Zero-shot CoT | 75.5 | 87.6 | -0.8 | 56.2 | 18.6 | 38.5 | 80.0 | 85.5 |
| w/ Plan-and-Solve | 70.9 | 86.2 | -0.3 | 56.9 | 15.2 | 42.1 | 85.5 | 82.8 |
| w/ Step-Back | 72.7 | 88.4 | -0.6 | 57.4 | 18.1 | 38.7 | 82.7 | 85.5 |
| w/ LOOP (ours) | 77.3 | 89.4 | 0.0 | 59.1 | 19.9 | 37.6 | 86.4 | 88.5 |
| Gemini 2.5 Flash non-reasoning | 5.5 | 60.4 | 29.3 | 29.9 | 11.3 | 44.0 | 39.1 | 49.3 |
| w/ LOOP (ours) | 34.5 | 67.8 | 15.8 | 35.4 | 14.3 | 40.2 | 51.8 | 62.3 |
| GPT-5 Chat | 7.3 | 72.5 | 7.6 | 45.9 | 16.6 | 39.7 | 60.9 | 63.7 |
| w/ LOOP (ours) | 24.5 | 81.7 | 1.9 | 51.7 | 16.3 | 37.3 | 73.6 | 72.4 |
| DeepSeek-V3.2 reasoning | 18.2 | 79.8 | 4.3 | 44.5 | 15.5 | 38.7 | 65.5 | 67.5 |
| w/ LOOP (ours) | 17.3 | 78.9 | 3.6 | 46.3 | 17.5 | 35.7 | 56.4 | 67.8 |
| Claude Opus 4.1 thinking | 17.3 | 81.6 | 1.5 | 58.2 | 17.7 | 38.7 | 79.1 | 73.6 |
| w/ LOOP (ours) | 22.7 | 86.8 | 1.4 | 60.3 | 19.8 | 36.2 | 88.2 | 78.9 |

Table 10: Complementary metrics for molecular structure TikZ generation.

| Model | Acc↑ | DSim↑ | KID↓ | CLIP↑ | cBLEU↑ | EED↓ | Coverage↑ | Average↑ |
|---|---|---|---|---|---|---|---|---|
| **Fine-tuned models** | | | | | | | | |
| AutomaTikZ | 0.0 | 24.0 | 177.3 | 0.00 | 0.00 | 96.8 | 0.0 | 0.7 |
| TikZero+ | 0.0 | 26.1 | 155.5 | 0.00 | 0.00 | 69.9 | 0.0 | 8.5 |
| AutomaTikZ resampling | 0.0 | 27.7 | 122.7 | 0.06 | 0.00 | 90.8 | 0.0 | 7.1 |
| DeTikZify resampling | 0.0 | 43.4 | 34.8 | 0.12 | 0.01 | 54.3 | 0.7 | 25.2 |
| **Non-reasoning models** | | | | | | | | |
| DeepSeek-V3 | 5.3 | 72.9 | 15.9 | 3.70 | 0.49 | 39.0 | 15.3 | 40.08 |
| DeepSeek-V3.1 | 6.3 | 75.4 | 12.5 | 4.91 | 0.61 | 38.2 | 19.0 | 42.04 |
| Claude Opus 4.1 | 24.3 | 80.9 | 6.0 | 8.26 | 0.69 | 38.7 | 52.0 | 52.27 |
| Gemini 2.0 Flash | 6.0 | 68.9 | 13.0 | 5.14 | 0.31 | 51.3 | 18.7 | 37.68 |
| Gemini 2.5 Flash non-reasoning | 22.7 | 64.4 | 21.5 | 3.25 | 0.25 | 53.9 | 34.3 | 40.38 |
| GPT-4.1 | 19.0 | 71.6 | 18.1 | 3.14 | 0.34 | 44.6 | 33.7 | 43.41 |
| GPT-5 Chat | 16.0 | 66.0 | 19.9 | 1.88 | 0.06 | 51.1 | 24.7 | 38.47 |
| **Rasoning models** | | | | | | | | |
| DeepSeek-R1 | 18.3 | 69.9 | 16.5 | 2.64 | 0.29 | 44.2 | 25.7 | 41.82 |
| DeepSeek-V3.1 reasoning | 31.0 | 71.0 | 12.8 | 3.14 | 0.54 | 39.5 | 46.3 | 48.85 |
| Claude Opus 4.1 thinking | 26.7 | 83.6 | 4.5 | 8.97 | 0.71 | 37.1 | 54.7 | 54.27 |
| Gemini 2.5 Flash reasoning | 32.0 | 60.0 | 31.3 | 3.06 | 0.15 | 53.1 | 38.7 | 41.03 |
| Gemini 2.5 Pro | 41.3 | 78.6 | 8.1 | 5.39 | 0.62 | 34.4 | 72.3 | 57.96 |
| o4-mini | 33.3 | 69.5 | 15.4 | 4.39 | 0.28 | 41.1 | 43.7 | 48.09 |
| GPT-5 | 52.3 | 71.3 | 13.6 | 3.01 | 0.81 | 42.7 | 74.7 | 56.20 |
| **Prompting** | | | | | | | | |
| Gemini 2.5 Pro | 41.3 | 78.6 | 8.1 | 5.39 | 0.62 | 34.4 | 72.3 | 57.96 |
| w/ Zero-shot CoT | 47.7 | 79.7 | 6.8 | 5.13 | 0.77 | 33.7 | 70.3 | 59.32 |
| w/ Plan-and-Solve | 41.3 | 77.6 | 8.2 | 4.63 | 0.63 | 33.8 | 70.3 | 57.44 |
| w/ Step-Back | 40.7 | 79.4 | 6.3 | 5.56 | 0.68 | 33.3 | 65.7 | 57.49 |
| w/ LOOP (ours) | 47.7 | 80.2 | 6.3 | 7.03 | 0.68 | 33.5 | 71.0 | 59.90 |
| GPT-5 | 52.3 | 71.3 | 13.6 | 3.01 | 0.81 | 42.7 | 74.7 | 56.20 |
| w/ Zero-shot CoT | 53.0 | 70.0 | 14.8 | 2.79 | 0.76 | 42.4 | 73.7 | 55.82 |
| w/ Plan-and-Solve | 52.3 | 71.5 | 13.4 | 3.03 | 0.70 | 39.6 | 73.3 | 56.66 |
| w/ Step-Back | 50.7 | 70.8 | 13.7 | 3.05 | 0.69 | 41.5 | 72.3 | 55.67 |
| w/ LOOP (ours) | 55.0 | 76.7 | 9.4 | 4.72 | 0.78 | 36.7 | 79.3 | 60.40 |
| Gemini 2.5 Flash non-reasoning | 22.7 | 64.4 | 21.5 | 3.25 | 0.25 | 53.9 | 34.3 | 40.38 |
| w/ LOOP (ours) | 21.3 | 60.5 | 25.4 | 4.19 | 0.16 | 56.2 | 26.0 | 37.44 |
| GPT-5 Chat | 16.0 | 66.0 | 19.9 | 1.88 | 0.06 | 51.1 | 24.7 | 38.47 |
| w/ LOOP (ours) | 19.0 | 69.7 | 17.0 | 2.61 | 0.13 | 48.5 | 22.3 | 40.38 |
| DeepSeek-V3.2 reasoning | 14.0 | 70.0 | 14.6 | 3.65 | 0.66 | 40.5 | 28.3 | 42.78 |
| w/ LOOP (ours) | 16.0 | 75.2 | 10.4 | 4.76 | 0.75 | 38.3 | 33.7 | 46.06 |
| Claude Opus 4.1 thinking | 26.7 | 83.6 | 4.5 | 8.97 | 0.71 | 37.1 | 54.7 | 54.27 |
| w/ LOOP (ours) | 30.7 | 83.7 | 4.9 | 9.43 | 0.70 | 37.0 | 54.7 | 55.05 |

Table 11: Complementary metrics for molecular structure SVG generation.

| Model | Acc↑ | DSim↑ | KID↓ | CLIP↑ | cBLEU↑ | EED↓ | Coverage↑ | Average↑ |
|---|---|---|---|---|---|---|---|---|
| **Non-reasoning models** | | | | | | | | |
| DeepSeek-V3 | 3.3 | 20.77 | 171.88 | 11.09 | 0.47 | 45.71 | 10.00 | 15.5 |
| DeepSeek-V3.1 | 3.7 | 22.79 | 161.72 | 11.75 | 0.33 | 49.49 | 24.84 | 18.3 |
| Claude Opus 4.1 | 26.0 | 30.04 | 152.34 | 7.81 | 0.58 | 48.64 | 66.00 | 30.1 |
| Gemini 2.0 Flash | 3.7 | 39.08 | 96.48 | 4.47 | 0.53 | 47.65 | 22.36 | 25.7 |
| Gemini 2.5 Flash non-reasoning | 11.3 | 39.68 | 103.13 | 3.70 | 0.44 | 56.45 | 27.16 | 25.3 |
| GPT-4.1 | 15.0 | 28.96 | 145.31 | 5.55 | 0.40 | 48.00 | 24.67 | 22.0 |
| GPT-5 Chat | 14.3 | 30.30 | 146.09 | 3.20 | 0.35 | 48.81 | 23.00 | 21.3 |
| **Rasoning models** | | | | | | | | |
| DeepSeek-R1 | 20.0 | 27.48 | 143.75 | 4.48 | 0.52 | 47.01 | 29.67 | 23.5 |
| DeepSeek-V3.1 reasoning | 7.3 | 22.73 | 164.84 | 9.79 | 0.49 | 48.47 | 31.89 | 19.6 |
| Claude Opus 4.1 thinking | 27.7 | 30.34 | 153.13 | 6.75 | 0.68 | 44.91 | 62.33 | 30.5 |
| Gemini 2.5 Flash reasoning | 39.3 | 51.54 | 66.80 | 2.81 | 0.37 | 52.59 | 45.63 | 38.9 |
| Gemini 2.5 Pro | 63.3 | 36.66 | 125.78 | 6.78 | 0.53 | 42.48 | 73.09 | 42.3 |
| o4-mini | 42.7 | 28.43 | 155.47 | 6.09 | 0.38 | 46.60 | 56.62 | 31.1 |
| GPT-5 | 55.7 | 37.97 | 121.09 | 4.07 | 0.60 | 51.98 | 76.00 | 39.6 |
| **Prompting** | | | | | | | | |
| Gemini 2.5 Pro | 63.3 | 36.66 | 125.78 | 6.78 | 0.53 | 42.48 | 73.09 | 42.3 |
| w/ Zero-shot CoT | 63.0 | 36.85 | 122.66 | 5.81 | 0.61 | 42.61 | 72.85 | 42.3 |
| w/ Plan-and-Solve | 58.7 | 35.59 | 125.00 | 5.61 | 0.57 | 42.44 | 70.53 | 40.8 |
| w/ Step-Back | 54.3 | 41.46 | 104.69 | 4.55 | 0.52 | 42.99 | 64.69 | 41.6 |
| w/ LOOP (ours) | 64.7 | 33.94 | 135.94 | 7.98 | 0.65 | 42.53 | 75.33 | 41.8 |
| GPT-5 | 55.7 | 37.97 | 121.09 | 4.07 | 0.60 | 51.98 | 76.00 | 39.6 |
| w/ Zero-shot CoT | 52.0 | 42.00 | 107.03 | 3.26 | 0.45 | 54.45 | 75.00 | 40.0 |
| w/ Plan-and-Solve | 50.3 | 41.66 | 107.81 | 3.75 | 0.53 | 52.63 | 72.00 | 39.6 |
| w/ Step-Back | 51.7 | 37.52 | 122.66 | 4.29 | 0.54 | 51.40 | 74.33 | 38.5 |
| w/ LOOP (ours) | 57.3 | 31.55 | 150.78 | 5.12 | 0.59 | 47.66 | 81.00 | 38.0 |
| Gemini 2.5 Flash non-reasoning | 11.3 | 39.68 | 103.13 | 3.70 | 0.44 | 56.45 | 27.16 | 25.3 |
| w/ LOOP (ours) | 30.0 | 46.39 | 82.81 | 3.21 | 0.34 | 57.12 | 39.68 | 33.1 |
| GPT-5 Chat | 14.3 | 30.30 | 146.09 | 3.20 | 0.35 | 48.81 | 23.00 | 21.3 |
| w/ LOOP (ours) | 19.7 | 27.16 | 159.38 | 5.27 | 0.59 | 45.16 | 28.00 | 22.5 |
| DeepSeek-V3.2 reasoning | 10.7 | 27.40 | 142.97 | 7.41 | 0.68 | 49.65 | 25.00 | 21.3 |
| w/ LOOP (ours) | 27.7 | 28.11 | 146.09 | 7.01 | 0.74 | 47.89 | 38.00 | 26.5 |
| Claude Opus 4.1 thinking | 27.7 | 30.34 | 153.13 | 6.75 | 0.68 | 44.91 | 62.33 | 30.5 |
| w/ LOOP (ours) | 27.0 | 27.25 | 166.41 | 7.64 | 0.82 | 45.15 | 59.00 | 28.3 |

Table 12: Complementary metrics for molecular structure EPS generation.

| Model | Acc↑ | DSim↑ | KID↓ | CLIP↑ | cBLEU↑ | EED↓ | Coverage↑ | Average↑ |
|---|---|---|---|---|---|---|---|---|
| **Non-reasoning models** | | | | | | | | |
| DeepSeek-V3 | 3.0 | 61.38 | 31.25 | 2.61 | 0.03 | 49.28 | 9.33 | 32.6 |
| DeepSeek-V3.1 | 3.3 | 64.77 | 24.90 | 1.77 | 0.03 | 49.10 | 13.33 | 34.3 |
| Claude Opus 4.1 | 16.0 | 77.47 | 11.13 | 5.14 | 0.07 | 49.79 | 58.33 | 47.4 |
| Gemini 2.0 Flash | 0.7 | 29.54 | 140.63 | 0.03 | 0.01 | 56.48 | 0.67 | 13.5 |
| Gemini 2.5 Flash non-reasoning | 14.3 | 58.86 | 34.57 | 2.23 | 0.04 | 57.81 | 31.33 | 35.1 |
| GPT-4.1 | 13.7 | 66.67 | 22.75 | 3.33 | 0.05 | 51.73 | 24.33 | 38.1 |
| GPT-5 Chat | 11.0 | 52.22 | 49.80 | 0.46 | 0.01 | 54.50 | 16.00 | 30.1 |
| **Rasoning models** | | | | | | | | |
| DeepSeek-R1 | 19.7 | 67.07 | 24.22 | 1.34 | 0.05 | 48.31 | 27.33 | 39.9 |
| DeepSeek-V3.1 reasoning | 20.7 | 65.49 | 22.66 | 1.48 | 0.04 | 49.62 | 33.00 | 40.4 |
| Claude Opus 4.1 thinking | 23.3 | 84.57 | 6.74 | 7.93 | 0.11 | 47.26 | 54.00 | 50.9 |
| Gemini 2.5 Flash reasoning | 34.7 | 57.88 | 38.67 | 2.63 | 0.03 | 54.74 | 43.33 | 40.7 |
| Gemini 2.5 Pro | 57.3 | 79.00 | 3.98 | 9.83 | 0.08 | 45.14 | 76.67 | 60.1 |
| o4-mini | 39.0 | 71.66 | 14.84 | 4.94 | 0.04 | 49.13 | 51.67 | 49.1 |
| GPT-5 | 49.7 | 69.79 | 18.26 | 3.13 | 0.05 | 57.29 | 73.33 | 51.4 |
| **Prompting** | | | | | | | | |
| Gemini 2.5 Pro | 57.3 | 79.00 | 3.98 | 9.83 | 0.08 | 45.14 | 76.67 | 60.1 |
| w/ Zero-shot CoT | 58.7 | 79.59 | 3.91 | 8.83 | 0.10 | 45.61 | 79.33 | 60.6 |
| w/ Plan-and-Solve | 55.0 | 79.22 | 3.88 | 9.25 | 0.07 | 45.44 | 76.00 | 59.5 |
| w/ Step-Back | 56.3 | 77.40 | 4.83 | 7.94 | 0.07 | 45.62 | 73.67 | 58.6 |
| w/ LOOP (ours) | 67.7 | 80.65 | 3.81 | 9.14 | 0.09 | 44.97 | 79.67 | 62.7 |
| GPT-5 | 49.7 | 69.79 | 18.26 | 3.13 | 0.05 | 57.29 | 73.33 | 51.4 |
| w/ Zero-shot CoT | 49.3 | 69.82 | 17.19 | 3.57 | 0.05 | 56.81 | 70.00 | 51.1 |
| w/ Plan-and-Solve | 50.7 | 70.57 | 15.92 | 4.05 | 0.04 | 55.41 | 70.33 | 52.1 |
| w/ Step-Back | 48.3 | 68.53 | 20.31 | 3.12 | 0.04 | 56.33 | 68.67 | 50.3 |
| w/ LOOP (ours) | 54.3 | 76.43 | 9.67 | 4.73 | 0.07 | 50.75 | 79.00 | 56.8 |
| Gemini 2.5 Flash non-reasoning | 14.3 | 58.86 | 34.57 | 2.23 | 0.04 | 57.81 | 31.33 | 35.1 |
| w/ LOOP (ours) | 22.3 | 54.08 | 46.48 | 2.50 | 0.04 | 59.45 | 34.33 | 34.8 |
| GPT-5 Chat | 11.0 | 52.22 | 49.80 | 0.46 | 0.01 | 54.50 | 16.00 | 30.1 |
| w/ LOOP (ours) | 18.7 | 70.96 | 20.70 | 2.19 | 0.06 | 48.52 | 22.33 | 40.2 |
| DeepSeek-V3.2 reasoning | 6.7 | 68.01 | 19.24 | 1.86 | 0.04 | 51.89 | 18.67 | 36.2 |
| w/ LOOP (ours) | 9.0 | 73.88 | 14.65 | 3.59 | 0.08 | 51.96 | 23.33 | 39.3 |
| Claude Opus 4.1 thinking | 23.3 | 84.57 | 6.74 | 7.93 | 0.11 | 47.26 | 54.00 | 50.9 |
| w/ LOOP (ours) | 22.3 | 83.46 | 6.59 | 7.45 | 0.10 | 47.23 | 51.67 | 50.1 |

## E.4 EVALUATION ON THE DATIKZ V3 DATASET

We evaluate the robustness of the proposed LOOP using the DaTikZ v3 dataset. The same prompt from the plane geometry task is employed. For comparison, we also include the fine-tuned models in the evaluation. Because DaTikZ v3 publishes only a subset of its test set, the fine-tuned models are re-evaluated. The results are shown in Figure 13. LOOP improves performance even on DaTikZ v3, demonstrating its robustness.

Table 13: Evaluation on the DaTikZ v3 Dataset.

| Model | DSim↑ | KID↓ | CLIP↑ | cBLEU↑ | EED↓ | Average↑ |
|---|---|---|---|---|---|---|
| AutomaTikZ resampling | 46.2 | 26.4 | 10.4 | 1.4 | 58.4 | 20.3 |
| TikZero+ resampling | 47.1 | 22.1 | 10.0 | 2.5 | 59.1 | 40.6 |
| Gemini 2.5 Pro | 46.6 | 14.2 | 28.4 | 1.7 | 60.7 | 30.7 |
|    w/ LOOP (ours) | 50.8 | 8.3 | 36.8 | 1.9 | 59.1 | 70.9 |
| GPT-5 | 53.1 | 6.1 | 42.3 | 2.2 | 59.5 | 84.7 |
|    w/ LOOP (ours | 53.5 | 6.1 | 39.5 | 2.5 | 59.6 | 87.5 |

## E.5 ABLATIONS ON LOOP

To identify the contribution of each component of LOOP to the overall performance, we perform an ablation study. The target task is plane geometry SVG generation using Gemini 2.5 Pro. The experiments are performed under the following three settings.

- w/o information extraction: *"Let's think step by step, following this workflow: 1. Relationship extraction: describe the relationships among the elements. 2. Mathematical reasoning: compute the attributes of each element so that they satisfy those relationships. 3. Code generation: generate the TikZ."*

- w/o relationship extraction: *"Let's think step by step, following this workflow: 1. Information extraction: describe the necessary elements. 2. Mathematical reasoning: compute the attributes of each element. 3. Code generation: generate the TikZ."*

- w/o mathematical reasoning: *"Let's think step by step, following this workflow: 1. Information extraction: describe the necessary elements. 2. Relationship extraction: describe their relationships. 3. Code generation: generate the TikZ."*

- w/o code generation: *"Let's think step by step, following this workflow: 1. Information extraction: describe the necessary elements. 2. Relationship extraction: describe their relationships. 3. Mathematical reasoning: compute the attributes of each element so that they satisfy those relationships."*

We show the results of applying these prompts in Figure 14. The original prompt achieves the best performance, clearly demonstrating that each component contributes to the improvement.

Table 14: The performance of Gemini 2.5 Pro with different prompts in the plane-geometry SVG generation task.

| Method | Accuracy |
|---|---|
| LOOP | 80.9 |
| w/o information extraction | 72.7 |
| w/o relationship extraction | 72.7 |
| w/o mathematical reasoning | 70.0 |
| w/o code generation | 72.7 |

# F    DETAILS OF THE EVALUATION CODE

We describe below the evaluation code used in our SSVG-Bench, specifically for the SVG format. For TikZ and EPS, since we first compile them to PDF and then convert them to SVG using the pdf2svg command before processing, the code is nearly identical to that used for SVG.

## F.1    PLANE GEOMETRY

### F.1.1    PARSING CODE

We first present the parsing code for extracting straight lines, circles, and ellipses from vector data (in this case, SVG), as shown in Figures 23, 24, 25, 26, and 27. This enables subsequent structural analysis.

**Overview.** The code uses Python's built-in XML parser (`xml.etree.ElementTree`) to traverse the SVG document tree. For selected shapes, it extracts the coordinates of geometric primitives in a uniform form:

- Line-like shapes are decomposed into straight line segments of the form $((x_1, y_1), (x_2, y_2))$.
- Circles are returned as $((cx, cy), r)$.
- Ellipses are represented as $((cx, cy), R_1, R_2, \theta)$, where $R_1, R_2$ are the principal semi-axes (radii) and $\theta$ is the rotation of the major axis from the $x$-axis.

All extracted data can optionally be filtered by CSS class.

**Extracting Line Segments.** The function `parse_segments()` scans the SVG tree and detects elements whose geometry can be expressed as connected straight lines:

- `<line>` elements directly provide two endpoints.
- `<polygon>` and `<polyline>` elements contain a sequence of vertex coordinates, where each consecutive pair defines a segment (polygons also include the closing edge).
- `<rect>` elements are decomposed into four boundary segments.
- `<path>` elements are partially supported for straight motions (M, L). Each L instruction yields a segment from the previous position.

For all these shapes, the function returns a list of straight-line segments.

**Extracting Circles.** The function `parse_circles()` searches for `<circle>` elements and extracts their center $(cx, cy)$ and radius $r$. When a filter is specified, only circles matching the given CSS class are processed.

**Extracting Ellipses with Affine Transforms.** The function `parse_ellipses()` identifies `<ellipse>` elements as well as transformed `<circle>` elements. It supports general SVG affine transformations including: `matrix`, `translate`, `rotate`, `scale`, `skewX`, `skewY`. The transformation is processed as follows:

1. Construct the affine matrix from the SVG `transform` attribute.
2. Apply the affine transformation to the ellipse center.
3. Decompose the linear part of the affine transform applied to the ellipse radii. The principal radii $R_1, R_2$ and orientation $\theta$ are obtained from eigenvalue analysis of the matrix $AA^T$.

**Result.** The function `parse_svg_file()` loads the SVG data, calls the three extraction routines, and returns: `segs, circs, ells`, where:

- `segs` : list of line segments  $(x_1, y_1), (x_2, y_2)$.
- `circs` : list of circles  $((cx, cy), r)$.
- `ells` : list of ellipses  $((cx, cy), R_1, R_2, \theta)$.

```python
#!/usr/bin/env python3
# -*- coding: utf-8 -*-

import re
import math
import xml.etree.ElementTree as ET

def tag(elem):
    """Return the tag name without namespace."""
    return elem.tag.split('}')[-1]

def parse_segments(root, filter_class=None):
    """
    Decompose line, polygon, polyline, path, and rect elements into line segments.
    If filter_class is specified, only elements that have that class attribute are used.
    Returns: [ ((x1, y1), (x2, y2)), ... ]
    """
    segs = []

    def ok_class(elem):
        if filter_class is None:
            return True
        return filter_class in elem.get('class', '').split()

    for elem in root.iter():
        t = tag(elem)

        def safe_float(value):
            try:
                return float(value)
            except (TypeError, ValueError):
                return 0.0

        if t == 'line' and ok_class(elem):
            x1 = safe_float(elem.get('x1', 0))
            y1 = safe_float(elem.get('y1', 0))
            x2 = safe_float(elem.get('x2', 0))
            y2 = safe_float(elem.get('y2', 0))
            segs.append(((x1, y1), (x2, y2)))

        elif t == 'polygon' and ok_class(elem):
            pts = re.split(r'[,\s]+', elem.get('points', '').strip())
            coords = [float(v) for v in pts if v]
            pts_list = list(zip(coords[0::2], coords[1::2]))
            for i in range(len(pts_list)):
                segs.append((pts_list[i], pts_list[(i + 1) % len(pts_list)]))

        elif t == 'polyline' and ok_class(elem):
            pts = re.split(r'[,\s]+', elem.get('points', '').strip())
            coords = [float(v) for v in pts if v]
            pts_list = list(zip(coords[0::2], coords[1::2]))
            for i in range(len(pts_list) - 1):
                segs.append((pts_list[i], pts_list[i + 1]))

        elif t == 'path' and ok_class(elem):
            d = elem.get('d', '')
            tokens = re.findall(r'[ML]|[-+]?\d*\.?\d+(?:\.\d+)?', d)
            cur = None
            i = 0
            while i < len(tokens):
                tok = tokens[i]
                if tok == 'M':
                    cur = (float(tokens[i + 1]), float(tokens[i + 2]))
```

Figure 23: SVG parsing code (1/5).

```
 66                        i += 3
 67                elif tok == 'L' and cur is not None:
 68                    nxt = (float(tokens[i + 1]), float(tokens[i + 2]))
 69                    segs.append((cur, nxt))
 70                    cur = nxt
 71                    i += 3
 72                else:
 73                    i += 1
 74
 75        elif t == 'rect' and ok_class(elem):
 76            x = float(elem.get('x', '0'))
 77            y = float(elem.get('y', '0'))
 78            w = float(elem.get('width', '0'))
 79            h = float(elem.get('height', '0'))
 80            p1 = (x, y)
 81            p2 = (x + w, y)
 82            p3 = (x + w, y + h)
 83            p4 = (x, y + h)
 84            segs.extend([(p1, p2), (p2, p3), (p3, p4), (p4, p1)])
 85
 86    return segs
 87
 88
 89 def parse_circles(root, filter_class=None):
 90     """
 91     Extract circle elements. If filter_class is specified, filter by that class.
 92     Returns: [ ((cx, cy), r), ... ]
 93     """
 94     circs = []
 95     for elem in root.iter():
 96         if tag(elem) == 'circle':
 97             cls = elem.get('class', '').split()
 98             if filter_class is None or filter_class in cls:
 99                 cx = float(elem.get('cx', 0))
100                 cy = float(elem.get('cy', 0))
101                 r = float(elem.get('r', 0))
102                 circs.append(((cx, cy), r))
103     return circs
104
105
106 def parse_ellipses(root, filter_class=None):
107     """
108     Extract ellipse elements. Supports translate / rotate / scale / matrix in the transform attribute.
109     Returns: [ ((cx, cy), rx, ry, angle_deg), ... ]
110     - angle_deg: angle in degrees, counter-clockwise from the x-axis
111     - Even with non-uniform scale or arbitrary matrices, approximates major/minor axis lengths and
       orientation.
112     """
113     ellipses = []
114     # Tokenizer for transform attribute
115     t_re = re.compile(r'(matrix|translate|rotate|scale|skewX|skewY)\s*\(([^)]+)\)', re.I)
116
117     def mult(A, B):
118         """3x3 matrix multiplication A @ B."""
119         return [
120             [
121                 A[0][0] * B[0][0] + A[0][1] * B[1][0] + A[0][2] * B[2][0],
122                 A[0][0] * B[0][1] + A[0][1] * B[1][1] + A[0][2] * B[2][1],
123                 A[0][0] * B[0][2] + A[0][1] * B[1][2] + A[0][2] * B[2][2],
124             ],
125             [
126                 A[1][0] * B[0][0] + A[1][1] * B[1][0] + A[1][2] * B[2][0],
127                 A[1][0] * B[0][1] + A[1][1] * B[1][1] + A[1][2] * B[2][1],
128                 A[1][0] * B[0][2] + A[1][1] * B[1][2] + A[1][2] * B[2][2],
129             ],
130             [0, 0, 1],
131         ]
```

Figure 24: SVG parsing code (2/5).

```python
132
133     def mat_translate(tx, ty=0.0):
134         return [[1, 0, tx], [0, 1, ty], [0, 0, 1]]
135
136     def mat_rotate(angle_deg, cx=0.0, cy=0.0):
137         a = math.radians(angle_deg)
138         c = math.cos(a)
139         s = math.sin(a)
140         # Affine transform including rotation around center (cx, cy)
141         return mult(
142             mult(mat_translate(cx, cy), [[c, -s, 0], [s, c, 0], [0, 0, 1]]),
143             mat_translate(-cx, -cy),
144         )
145
146     def mat_scale(sx, sy=None):
147         if sy is None:
148             sy = sx
149         return [[sx, 0, 0], [0, sy, 0], [0, 0, 1]]
150
151     def mat_skewx(a_deg):
152         t = math.tan(math.radians(a_deg))
153         # Note: SVG skewX corresponds to x' = x + tan(ax) * y
154         return [[1, math.tan(0), 0], [t, 1, 0], [0, 0, 1]]
155
156     def mat_skewy(a_deg):
157         t = math.tan(math.radians(a_deg))
158         # Note: SVG skewY corresponds to y' = y + tan(ay) * x
159         return [[1, t, 0], [math.tan(0), 1, 0], [0, 0, 1]]
160
161     def mat_matrix(a, b, c, d, e, f):
162         # SVG: [x', y'] = [a c e; b d f; 0 0 1] [x, y, 1]^T
163         return [[a, c, e], [b, d, f], [0, 0, 1]]
164
165     def parse_transform(txt):
166         """Compose transform string from left to right (SVG applies transforms in that order)."""
167         M = [[1, 0, 0], [0, 1, 0], [0, 0, 1]]
168         for m in t_re.finditer(txt or ''):
169             name = m.group(1).lower()
170             parts = [float(v) for v in re.split(r'[,\s]+', m.group(2).strip()) if v]
171             if name == 'matrix' and len(parts) == 6:
172                 Mi = mat_matrix(*parts)
173             elif name == 'translate':
174                 Mi = mat_translate(parts[0], parts[1] if len(parts) > 1 else 0.0)
175             elif name == 'rotate':
176                 if len(parts) >= 3:
177                     Mi = mat_rotate(parts[0], parts[1], parts[2])
178                 else:
179                     Mi = mat_rotate(parts[0])
180             elif name == 'scale':
181                 Mi = mat_scale(parts[0], parts[1] if len(parts) > 1 else None)
182             elif name == 'skewx':
183                 Mi = mat_skewx(parts[0])
184             elif name == 'skewy':
185                 Mi = mat_skewy(parts[0])
186             else:
187                 continue
188             M = mult(M, Mi)
189         return M
190
191     def apply_affine(M, x, y):
192         """Apply 3x3 affine matrix to point (x, y)."""
193         return (
194             M[0][0] * x + M[0][1] * y + M[0][2],
195             M[1][0] * x + M[1][1] * y + M[1][2],
196         )
197
198     def decompose_axes(Mlin, rx, ry):
```

Figure 25: SVG parsing code (3/5).

```
199          """
200          Linear part of ellipse transform: A = Mlin @ diag(rx, ry)
201          Major/minor axis lengths = sqrt(eigenvalues of A A^T),
202          orientation angle = angle of the principal eigenvector.
203          """
204          a, c = Mlin[0][0], Mlin[0][1]
205          b, d = Mlin[1][0], Mlin[1][1]
206          # A = [[a*rx, c*ry],
207          #      [b*rx, d*ry]]
208          arx, cry = a * rx, c * ry
209          brx, dry = b * rx, d * ry
210          # B = A A^T = [[p, r], [r, q]]
211          p = arx * arx + cry * cry
212          q = brx * brx + dry * dry
213          r = arx * brx + cry * dry
214          # Eigenvalues (>= 0)
215          trace = p + q
216          diff = p - q
217          disc = math.hypot(diff, 2 * r)  # sqrt(diff^2 + (2r)^2)
218          lam1 = 0.5 * (trace + disc)  # larger eigenvalue
219          lam2 = 0.5 * (trace - disc)  # smaller eigenvalue
220          # Radii (>= 0)
221          R1 = math.sqrt(max(lam1, 0.0))
222          R2 = math.sqrt(max(lam2, 0.0))
223          # Angle (orientation of major axis): 0.5 * atan2(2r, p - q)
224          angle = 0.5 * math.atan2(2 * r, diff) if (abs(r) + abs(diff)) > 0 else 0.0
225          return R1, R2, math.degrees(angle)
226
227      # --- Main loop ---
228      for elem in root.iter():
229          if tag(elem) == 'ellipse':
230              cls = elem.get('class', '').split()
231              if filter_class is not None and filter_class not in cls:
232                  continue
233              cx = float(elem.get('cx', 0.0))
234              cy = float(elem.get('cy', 0.0))
235              rx = float(elem.get('rx', 0.0))
236              ry = float(elem.get('ry', 0.0))
237
238              # Compose transform matrix (3x3)
239              M = parse_transform(elem.get('transform', ''))
240
241              # Transformed center
242              cx_t, cy_t = apply_affine(M, cx, cy)
243
244              # Linear part (2x2)
245              Mlin = [[M[0][0], M[0][1]], [M[1][0], M[1][1]]]
246
247              # Decompose radii and angle
248              if rx == 0.0 and ry == 0.0:
249                  R1 = R2 = 0.0
250                  angle_deg = 0.0
251              else:
252                  R1, R2, angle_deg = decompose_axes(Mlin, rx, ry)
253
254              ellipses.append(((cx_t, cy_t), R1, R2, angle_deg))
255
256          elif tag(elem) == 'circle' and elem.get('transform', '') != '':
257              cls = elem.get('class', '').split()
258              if filter_class is not None and filter_class not in cls:
259                  continue
260              cx = float(elem.get('cx', 0.0))
261              cy = float(elem.get('cy', 0.0))
262              rx = float(elem.get('r', 0.0))
263              ry = float(elem.get('r', 0.0))
264
265              # Compose transform matrix (3x3)
```

Figure 26: SVG parsing code (4/5).

```
266              M = parse_transform(elem.get('transform', ''))
267
268              # Transformed center
269              cx_t, cy_t = apply_affine(M, cx, cy)
270
271              # Linear part (2x2)
272              Mlin = [[M[0][0], M[0][1]], [M[1][0], M[1][1]]]
273
274              # Decompose radii and angle
275              if rx == 0.0 and ry == 0.0:
276                  R1 = R2 = 0.0
277                  angle_deg = 0.0
278              else:
279                  R1, R2, angle_deg = decompose_axes(Mlin, rx, ry)
280
281              ellipses.append(((cx_t, cy_t), R1, R2, angle_deg))
282      return ellipses
283
284
285  def parse_svg_file(path, filter_class=None):
286      root = ET.parse(path).getroot()
287      segs = parse_segments(root, filter_class)
288      circs = parse_circles(root, filter_class)
289      ells = parse_ellipses(root, filter_class)
290      return segs, circs, ells
```

Figure 27: SVG parsing code (5/5).

### F.1.2 EVALUATION CODE FOR PATTERN 1

We now present the evaluation code for Pattern 1, where the correct objects are not uniquely determined. This code checks whether the SVG produced by the LLM correctly reproduces the ground-truth primitives (straight segments, circles, and ellipses), as shown in Figures 28, 29, and 30. The core of this evaluation is implemented by the three functions `match_segments()`, `match_circles()`, and `match_ellipses()`.

**Overall Evaluation Procedure.** Given two SVG files,

- `ground_truth`: the reference SVG, and
- `llm_output`: the SVG generated by the model,

the script first calls `parse_svg_file()` to extract:

- line segments `gt_segs`, `out_segs`,
- circles `gt_circs`, `out_circs`,
- ellipses `gt_ells`, `out_ells`.

The output segments are then augmented by `merge_collinear_segments()`, which attempts to merge nearly collinear segments that share an endpoint, so that broken polylines can still match a single ground-truth segment.

The evaluation is decomposed into three independent stages:

$$\text{stage1} = \text{match\_segments}(\text{gt\_segs}, \text{out\_segs}),$$
$$\text{stage2} = \text{match\_circles}(\text{gt\_circs}, \text{out\_circs}),$$
$$\text{stage3} = \text{match\_ellipses}(\text{gt\_ells}, \text{out\_ells}).$$

If and only if all three stages succeed, the script prints `"1"`; otherwise it prints `"0"`.

**Step 1: Matching Line Segments.** The function `match_segments(gt_segs, out_segs, tol)` verifies that every ground-truth segment is represented in the LLM output within a geometric tolerance. Each segment is represented by its two endpoints:

$$\text{gt\_segs} = \{(p_1, p_2)\}, \quad \text{out\_segs} = \{(q_1, q_2)\},$$

where $p_1, p_2, q_1, q_2 \in \mathbb{R}^2$ denote 2D coordinates.

- For each ground-truth segment $(p_1, p_2)$, the function searches over all output segments $(q_1, q_2)$.
- The helper `dist_point_to_segment(pt, a, b)` computes the Euclidean distance from a point $\text{pt}$ to the finite segment $\overline{ab}$ by projecting $\text{pt}$ onto the segment and clamping the projection parameter to $[0, 1]$.
- A ground-truth segment is considered *matched* if there exists an output segment such that both endpoints $p_1$ and $p_2$ are within distance $\text{tol}$ of the segment $\overline{q_1 q_2}$:

$$\text{dist}(p_1, \overline{q_1 q_2}) \leq \text{tol}, \quad \text{dist}(p_2, \overline{q_1 q_2}) \leq \text{tol}.$$

If all ground-truth segments find such a matching segment in `out_segs`, the function returns `True`; otherwise it returns `False`. This ensures that every reference straight line is geometrically reproduced in the LLM output.

**Step 2: Matching Circles.** The function `match_circles(gt_circs, out_circs, tol_center, tol_r)` checks that all ground-truth circles are present in the output with similar centers and radii. Each circle is represented as $((c_x, c_y), r)$, where $c = (c_x, c_y)$ is the center and $r$ is the radius.

- The outer loop iterates over ground-truth circles $(c_{\text{gt}}, r_{\text{gt}})$. A working copy of output circles is stored in `unmatched` so that once an output circle is assigned to a ground-truth circle, it is removed and cannot be reused.

- For each ground-truth circle, the function selects the output circle with the maximum intersection-over-union (IoU), computed by `circle_iou(c1, r1, c2, r2)`. The IoU is defined between the two disks $D_1, D_2$ as:

$$\text{IoU}(D_1, D_2) = \frac{\text{area}(D_1 \cap D_2)}{\text{area}(D_1 \cup D_2)}.$$

  This IoU is used only to choose the *best* candidate in `unmatched`.

- After selecting the circle with maximum IoU, the function checks strict geometric tolerances on center and radius:

$$\|c_{\text{gt}} - c_{\text{out}}\| \leq \texttt{tol\_center}, \quad |r_{\text{gt}} - r_{\text{out}}| \leq \texttt{tol\_r}.$$

  If both conditions are satisfied, the output circle is removed from `unmatched` and the ground-truth circle is deemed matched. Otherwise, the entire matching fails and the function returns `False`.

If all ground-truth circles are successfully matched in this way, `match_circles()` returns `True`. Thus, every reference circle must appear in the LLM output with nearly the same center and radius.

**Step 3: Matching Ellipses.** The function `match_ellipses(gt_ells, out_ells, iou_thresh)` evaluates whether each ground-truth ellipse has a corresponding ellipse in the output that overlaps sufficiently in area. An ellipse is represented as:

$$((c_x, c_y), R_x, R_y, \theta),$$

where $(c_x, c_y)$ is the center, $R_x, R_y$ are the radii along the principal axes, and $\theta$ is the rotation angle.

- As in circle matching, a list `unmatched` stores the remaining output ellipses that have not yet been assigned.

- For each ground-truth ellipse $e_{\text{gt}}$, the function searches for the output ellipse $e_{\text{out}}$ that maximizes the IoU, computed by `ellipse_iou(e1, e2)`.

**Approximate IoU for Rotated Ellipses.** The helper `ellipse_iou(e1, e2, samples)` estimates IoU by Monte Carlo sampling:

1. For each ellipse, an axis-aligned bounding box is computed that tightly encloses the rotated ellipse.

2. A joint bounding box that covers both ellipses is obtained by taking the min/max of the two boxes.

3. A number of random points (given by `samples`) are uniformly sampled in this joint bounding box.

4. For each point, the function tests membership in each ellipse by transforming the point into the ellipse-aligned coordinate system and checking

$$\frac{x^2}{R_x^2} + \frac{y^2}{R_y^2} \leq 1.$$

5. From these samples, the areas of intersection and union are estimated, and the IoU is approximated as:

$$\text{IoU}(e_1, e_2) \approx \frac{\text{area}(e_1 \cap e_2)}{\text{area}(e_1 \cup e_2)}.$$

**IoU-based Acceptance Criterion.** Back in `match_ellipses()`, after computing IoU values between a ground-truth ellipse and all remaining output ellipses:

- The output ellipse with maximum IoU is selected as the candidate match.

- If this maximum IoU is greater than or equal to the threshold `iou_thresh` (default 0.95), the candidate is accepted and removed from `unmatched`.

- Otherwise, the function returns `False`, indicating that no sufficiently overlapping ellipse was found.

If every ground-truth ellipse is matched with IoU at least `iou_thresh`, the function returns `True`.

**Result.** Combining the three matching functions, the script provides a binary evaluation:

- The LLM output is considered *correct* if and only if all ground-truth line segments, circles, and ellipses are geometrically reproduced within the specified tolerances and IoU thresholds.
- In that case, the program prints `"1"`; otherwise it prints `"0"`.

This evaluation criterion enforces a strict structural fidelity of the generated SVG against the reference vector graphics.

```python
#!/usr/bin/env python3
# -*- coding: utf-8 -*-

import argparse
import xml.etree.ElementTree as ET
import re
import math
import sys
import random
from parse_svg_file import parse_svg_file

def dist_point_to_segment(pt, a, b):
    """Distance between point `pt` and segment `ab`"""
    x, y = pt; x1, y1 = a; x2, y2 = b
    dx, dy = x2-x1, y2-y1
    if dx == 0 and dy == 0:
        return math.hypot(x-x1, y-y1)
    t = ((x-x1)*dx + (y-y1)*dy) / (dx*dx + dy*dy)
    t = max(0.0, min(1.0, t))
    proj = (x1 + t*dx, y1 + t*dy)
    return math.hypot(x-proj[0], y-proj[1])

def match_segments(gt_segs, out_segs, tol=1e1):
    """Check whether each segment in ground truth matches any segment in output"""
    for p1, p2 in gt_segs:
        ok = False
        for q1, q2 in out_segs:
            if dist_point_to_segment(p1, q1, q2) <= tol and \
               dist_point_to_segment(p2, q1, q2) <= tol:
                ok = True
                break
        if not ok:
            return False
    return True

def circle_iou(c1, r1, c2, r2):
    """Calculate IoU (Intersection over Union) of two circles"""
    d = math.hypot(c1[0]-c2[0], c1[1]-c2[1])
    if d >= r1 + r2:
        return 0.0
    if d <= abs(r1 - r2):
        return (min(r1, r2)**2) / (max(r1, r2)**2)
    r1_sq, r2_sq = r1**2, r2**2
    alpha = math.acos((d*d + r1_sq - r2_sq) / (2*d*r1))
    beta  = math.acos((d*d + r2_sq - r1_sq) / (2*d*r2))
    inter = (r1_sq*alpha + r2_sq*beta -
                0.5*math.sqrt((-d+r1+r2)*(d+r1-r2)*(d-r1+r2)*(d+r1+r2)))
    union = math.pi*(r1_sq + r2_sq) - inter
    return inter / union

def match_circles(gt_circs, out_circs, tol_center=1e1, tol_r=1e1):
    """
    Match each ground-truth circle by maximizing IoU and
    filter by center distance and radius difference
    """
    unmatched = list(out_circs)
    for c_gt, r_gt in gt_circs:
        if not unmatched:
            return False
        # find best IoU
        best_i, best_iou = 0, circle_iou(c_gt, r_gt, unmatched[0][0], unmatched[0][1])
```

Figure 28: Evaluation code for Pattern 1 (1/3).

```
66          for i, (c_out, r_out) in enumerate(unmatched[1:], start=1):
67              iou = circle_iou(c_gt, r_gt, c_out, r_out)
68              if iou > best_iou:
69                  best_iou, best_i = iou, i
70          c_out, r_out = unmatched[best_i]
71          dc = math.hypot(c_gt[0]-c_out[0], c_gt[1]-c_out[1])
72          if dc <= tol_center and abs(r_gt-r_out) <= tol_r:
73              unmatched.pop(best_i)
74          else:
75              return False
76      return True

def ellipse_iou(e1, e2, samples=2000):
    """Compute approximate IoU for two rotated ellipses by Monte Carlo sampling"""
    def inside(pt, ell):
        (cx, cy), rx, ry, angle = ell
        rad = math.radians(-angle)
        cosA = math.cos(rad); sinA = math.sin(rad)
        dx, dy = pt[0]-cx, pt[1]-cy
        x = dx*cosA - dy*sinA
        y = dx*sinA + dy*cosA
        return x*x/(rx*rx) + y*y/(ry*ry) <= 1

    def bbox(ell):
        (cx, cy), rx, ry, angle = ell
        rad = math.radians(angle)
        cosA = math.cos(rad); sinA = math.sin(rad)
        w = abs(rx*cosA) + abs(ry*sinA)
        h = abs(rx*sinA) + abs(ry*cosA)
        return (cx-w, cy-h, cx+w, cy+h)

    b1 = bbox(e1); b2 = bbox(e2)
    minx = min(b1[0], b2[0]); miny = min(b1[1], b2[1])
    maxx = max(b1[2], b2[2]); maxy = max(b1[3], b2[3])
    area_box = (maxx-minx)*(maxy-miny)
    cnt1 = cnt2 = cnt_both = 0
    for _ in range(samples):
        x = random.uniform(minx, maxx)
        y = random.uniform(miny, maxy)
        in1 = inside((x, y), e1)
        in2 = inside((x, y), e2)
        if in1: cnt1 += 1
        if in2: cnt2 += 1
        if in1 and in2: cnt_both += 1
    if cnt1+cnt2-cnt_both == 0:
        return 0.0
    area_inter = cnt_both/samples * area_box
    area_union = (cnt1+cnt2-cnt_both)/samples * area_box
    return area_inter / area_union

def match_ellipses(gt_ells, out_ells, iou_thresh=0.95):
    """Match each ground-truth ellipse by maximizing IoU; pass if above threshold"""
    unmatched = list(out_ells)
    for gt in gt_ells:
        if not unmatched:
            return False
        best_i, best_iou = 0, ellipse_iou(gt, unmatched[0])
        for i, out in enumerate(unmatched[1:], start=1):
            iou = ellipse_iou(gt, out)
            if iou > best_iou:
                best_iou, best_i = iou, i
        if best_iou >= iou_thresh:
            unmatched.pop(best_i)
        else:
            return False
```

Figure 29: Evaluation code for Pattern 1 (2/3).

```
133        return True
134
135
136  def merge_collinear_segments(segments, angle_tol=math.radians(5)):
137        """Merge segments that share an endpoint and are almost collinear"""
138        merged = []
139        thresh = -math.cos(angle_tol)
140        n = len(segments)
141        for i in range(n):
142            u, v = segments[i]
143            for j in range(i+1, n):
144                w, x = segments[j]
145                shared = None
146                if u == w:
147                    shared, p1, p2 = u, v, x
148                elif u == x:
149                    shared, p1, p2 = u, v, w
150                elif v == w:
151                    shared, p1, p2 = v, u, x
152                elif v == x:
153                    shared, p1, p2 = v, u, w
154                else:
155                    continue
156                vec1 = (p1[0]-shared[0], p1[1]-shared[1])
157                vec2 = (p2[0]-shared[0], p2[1]-shared[1])
158                n1 = math.hypot(*vec1); n2 = math.hypot(*vec2)
159                if n1 == 0 or n2 == 0:
160                    continue
161                cos_ang = (vec1[0]*vec2[0] + vec1[1]*vec2[1]) / (n1*n2)
162                if cos_ang <= thresh:
163                    merged.append((p1, p2))
164        return merged
165
166
167  def main():
168      p = argparse.ArgumentParser(description="SVG Output Evaluation Script")
169      p.add_argument('ground_truth', help="File path of ground-truth SVG")
170      p.add_argument('llm_output',   help="File path of LLM-generated SVG")
171      args = p.parse_args()
172
173      try:
174          gt_segs, gt_circs, gt_ells = parse_svg_file(args.ground_truth, filter_class='output_object')
175          out_segs, out_circs, out_ells = parse_svg_file(args.llm_output)
176      except Exception:
177          print("0")
178          sys.exit(0)
179      out_segs.extend(merge_collinear_segments(out_segs))
180
181      # Matching
182      stage1 = match_segments(gt_segs, out_segs)
183      stage2 = match_circles(gt_circs, out_circs)
184      stage3 = match_ellipses(gt_ells, out_ells)
185
186      print("1" if (stage1 and stage2 and stage3) else "0")
187
188  if __name__ == '__main__':
189      main()
```

Figure 30: Evaluation code for Pattern 1 (3/3).

### F.1.3 EVALUATION CODE FOR PATTERN 2

In Pattern 2, it is not possible to uniquely determine the correct objects. To address this, we implemented case-specific Python logic that evaluates correctness based on textual input, allowing for variations in valid outputs. As examples, we present the evaluation code for the case in Figure 3 and for the bottom case in Figure 5.

The evaluation code corresponding to Figure 3 is shown in Figures 31 and 32.

**Overall Evaluation Procedure.** The script compares two SVG files:

- the ground-truth construction (annotated with the class `input_object`), and
- the LLM-generated output (annotated with the class `output_object`).

Both are parsed by `parse_svg_file()`, which returns:

$$(\texttt{gt\_segs}, \texttt{gt\_circs}, \texttt{gt\_ells}), \quad (\texttt{out\_segs}, \texttt{out\_circs}, \texttt{out\_ells})$$

for the input and output respectively. If parsing fails at any point, the script prints 0 and terminates, indicating an incorrect solution.

**Geometric Helper Functions.** Two helper routines implement basic Euclidean geometry needed for the evaluation:

- `dist_point_to_segment(pt, a, b)`: given a point $pt$ and a segment with endpoints $a$ and $b$, this function computes the shortest distance from $pt$ to the segment. It projects $pt$ onto the supporting line of $ab$, clamps the projection parameter to the interval $[0, 1]$, and returns the Euclidean distance from $pt$ to the resulting closest point.
- `circle_intersections(c1, r1, c2, r2)`: given two circles with centers $c_1, c_2$ and radii $r_1, r_2$, this function returns their intersection points. It first computes the center distance $d$; if the circles are too far apart, nested, or numerically degenerate, it returns an empty list. Otherwise, it computes the base point on the line between the centers and the perpendicular offset, yielding either one point (tangency) or two intersection points.

**Tolerance Parameters.** Because the LLM-generated SVG may differ slightly due to numerical precision or stylistic variation, the script uses small tolerances:

- `tol_center`: allowed deviation when matching circle centers to line endpoints.
- `tol_r`: allowed difference between radii.
- `tol_line`: allowed distance when checking whether intersection points lie on a candidate line segment.

These tolerances make the evaluation robust to minor floating-point noise.

**Step 1: Extracting the Reference Segment.** From the ground-truth data, the script takes the first input segment:

$$(p_1, p_2) = \texttt{gt\_segs[0]}.$$

This segment represents the original line in Figure 3 on which the construction is based. If no such segment exists, the evaluation immediately fails.

**Step 2: Matching Circles at the Endpoints.** The first condition requires that the output contain two circles whose centers coincide with the endpoints $p_1$ and $p_2$ of the input segment:

1. The script scans `out_circs` to find a circle with center within `tol_center` of $p_1$. This circle is stored as $(c_1, r_1)$.
2. It then scans again to find a *different* circle with center within `tol_center` of $p_2$, stored as $(c_2, r_2)$.

If either endpoint does not have a corresponding circle center, the script prints 0 (incorrect).

**Step 3: Checking the Radii.** The second condition enforces both equality and sufficient size of the radii:

- The radii must be equal up to tolerance: $|r_1 - r_2| \leq \texttt{tol\_r}$.
- Each radius must be strictly larger than half the length of the original segment:

$$r_1 > \frac{\|p_2 - p_1\|}{2} - \texttt{tol\_r}.$$

Intuitively, this ensures that the two circles intersect in two distinct points above and below the segment, as in the geometric construction. If either inequality is violated, the output is rejected.

**Step 4: Computing Circle Intersections.** The third condition uses the intersection points of the two circles:

1. The script calls `circle_intersections(c1, r1, c2, r2)` to compute the inter-section points.
2. If fewer than two intersection points are found (no intersection or tangency), the configu-ration cannot reproduce the intended construction, and the script outputs `0`.

When two intersections exist, denote them by $I_1$ and $I_2$.

**Step 5: Verifying a Line Through Both Intersections.** Finally, the script checks whether the LLM has drawn a line that passes through both intersection points:

- It iterates over each output segment $(q_1, q_2) \in \texttt{out\_segs}$.
- For each segment, it computes the distance from $I_1$ and $I_2$ to the segment using `dist_point_to_segment`.
- If both distances are within `tol_line`, then $(q_1, q_2)$ is considered to pass through both circle intersections.

If such a segment is found, the configuration is deemed correct and the script prints `1`. Otherwise, it prints `0`.

```python
#!/usr/bin/env python3
# -*- coding: utf-8 -*-

import argparse
import xml.etree.ElementTree as ET
import re
import math
import sys
from parse_svg_file import parse_svg_file

# ----- Tolerances (adjust if necessary) -----
tol_center = 1e-1   # Tolerance for matching a circle center with a line endpoint
tol_r      = 1e-1   # Tolerance for comparing radii
tol_line   = 1e-1   # Tolerance for checking if intersection points lie on a line segment

def dist_point_to_segment(pt, a, b):
    """
    Return the distance between a point pt and a line segment ab.
    """
    x,y = pt
    x1,y1 = a
    x2,y2 = b
    dx = x2 - x1
    dy = y2 - y1
    if dx==0 and dy==0:
        return math.hypot(x-x1, y-y1)
    t = ((x-x1)*dx + (y-y1)*dy) / (dx*dx + dy*dy)
    t = max(0.0, min(1.0, t))
    proj_x = x1 + t*dx
    proj_y = y1 + t*dy
    return math.hypot(x - proj_x, y - proj_y)

def circle_intersections(c1, r1, c2, r2, tol=1e-6):
    """
    Return the intersection points of two circles.
    If the circles intersect at two points, return a list of length 2.
    If they are tangent, return a list of length 1.
    If they do not intersect, return an empty list.
    """
    x0,y0 = c1
    x1,y1 = c2
    dx = x1 - x0
    dy = y1 - y0
    d = math.hypot(dx,dy)
    # Same center or too far apart
    if d < tol or d > r1 + r2 + tol or d < abs(r1 - r2) - tol:
        return []
    # Distance a and height h of intersection
    a = (r1*r1 - r2*r2 + d*d) / (2*d)
    h2 = r1*r1 - a*a
    if h2 < 0 and abs(h2) <= tol:
        h = 0.0
    elif h2 < 0:
        return []
    else:
        h = math.sqrt(h2)
    xm = x0 + a * dx / d
    ym = y0 + a * dy / d
    rx = -dy * (h / d)
    ry = dx * (h / d)
    p1 = (xm + rx, ym + ry)
    p2 = (xm - rx, ym - ry)
    return [p1] if h == 0 else [p1, p2]
```

Figure 31: Evaluation code for the case in Figure 3 (1/2).

```python
66  def main():
67      p = argparse.ArgumentParser(description="SVG Output Evaluation Script")
68      p.add_argument('ground_truth', help="File path of correct SVG")
69      p.add_argument('llm_output',   help="File path of SVG output from LLM")
70      args = p.parse_args()
71
72      try:
73          gt_segs, gt_circs, gt_ells = parse_svg_file(args.ground_truth, filter_class='input_object')
74          out_segs, out_circs, out_ells = parse_svg_file(args.llm_output, filter_class='output_object')
75      except Exception:
76          print("0")
77          sys.exit(0)
78
79      input_line = gt_segs[0]
80      if input_line is None:
81          print("0"); sys.exit(0)
82
83      p1, p2 = input_line
84
85      # Find circles that match p1 and p2
86      c1 = r1 = c2 = r2 = None
87
88      # ----- Condition 1: The output must contain exactly two circles whose centers match the endpoints
        of the input segment -----
89      # Search for the circle centered at p1
90      for center, r in out_circs:
91          if math.hypot(center[0]-p1[0], center[1]-p1[1]) <= tol_center:
92              c1, r1 = center, r
93              break
94      if c1 is None:
95          print("0"); sys.exit(0)
96
97      # Search for the circle centered at p2 (make sure it's not the same one as c1)
98      for center, r in out_circs:
99          # Avoid matching the same circle by excluding identical centers
100          if (abs(center[0]-c1[0]) > tol_center or abs(center[1]-c1[1]) > tol_center) \
101              and math.hypot(center[0]-p2[0], center[1]-p2[1]) <= tol_center:
102              c2, r2 = center, r
103              break
104      if c2 is None:
105          print("0"); sys.exit(0)
106
107      # --- Condition 2: Radii must match and must be greater than half the segment length ---
108      line_len = math.hypot(p2[0]-p1[0], p2[1]-p1[1])
109      if abs(r1 - r2) > tol_r or r1 <= line_len/2 - tol_r:
110          print("0"); sys.exit(0)
111
112      # --- Condition 3: There must exist a line segment passing through the intersection points of the
        circles ---
113      inters = circle_intersections(c1, r1, c2, r2, tol=tol_r)
114      if len(inters) < 2:
115          print("0"); sys.exit(0)
116
117      found_line = False
118      for q1, q2 in out_segs:
119          if dist_point_to_segment(inters[0], q1, q2) <= tol_line \
120              and dist_point_to_segment(inters[1], q1, q2) <= tol_line:
121              found_line = True
122              break
123
124      if not found_line:
125          print("0"); sys.exit(0)
126
127      # All conditions satisfied
128      print("1")
129
130  if __name__ == '__main__':
131      main()
```

Figure 32: Evaluation code for the case in Figure 3 (2/2).

The evaluation code corresponding to the bottom case in Figure 5 is shown in Figures 33 and 34.

**Overall Evaluation Procedure.** The script evaluates whether an LLM–generated SVG correctly draws the external tangents between four small circles found in a ground–truth SVG. It parses both SVG files using `parse_svg_file()`:

$$(\texttt{gt\_segs}, \texttt{gt\_circs}, \texttt{gt\_ells}), \quad (\texttt{out\_segs}, \texttt{out\_circs}, \texttt{out\_ells})$$

Only graphical items marked with the class names `input_object` (ground truth) and `output_object` (LLM output) are retrieved. If parsing fails, the script immediately prints 0.

**Geometric Helper Functions.** Two main routines support the evaluation:

- `dist_point_to_segment(pt, a, b)` computes the Euclidean distance from a point $pt$ to a segment with endpoints $a, b$. Using a projection onto line $ab$, the value is clamped to the segment and the closest-point distance is returned.
- `external_tangents(c1, r1, c2, r2)` returns pairs of tangent contact points between two circles with centers $c_1, c_2$ and radii $r_1, r_2$. If the distance between the centers is too small (one circle inside another or touching internally), no external tangents exist and an empty list is returned.

**Tolerance Handling.** A fixed distance tolerance

$$\texttt{tol} = 10$$

is used when determining whether a drawn segment approximates a target tangent. Small variations due to SVG scaling or floating precision are therefore accepted.

**Reference Circles in the Ground Truth.** The script extracts all circles from the ground truth drawing and sorts them by increasing radius. Only

the four circles with the smallest radii

are used as the construction targets. If fewer than five ground–truth circles exist, the problem is considered invalid and the output score is 0.

**Expected Geometry: External Tangents.** For every pair among the four selected circles (six pairs in total), the script computes the two possible external tangent segments:

$$\big((p_{11}, p_{12}), (p_{21}, p_{22})\big)$$

Each tangent is a pair of points, one on each circle. At least one tangent for each circle pair must appear in the LLM's output as a drawn segment.

**Matching Tangents in the LLM Output.** For each correct tangent candidate $(A, B)$, the script checks whether any output segment $(Q_1, Q_2)$ matches it using:

$$\texttt{match\_segments()} \iff \big(\text{dist}(A, Q_1 Q_2) \leq tol \,\wedge\, \text{dist}(B, Q_1 Q_2) \leq tol\big).$$

If no output segment matches at least one tangent for a given circle pair, evaluation halts and returns 0.

**Final Decision.** If all six circle pairs have at least one tangent segment successfully approximated by the LLM–generated SVG, the script prints: 1 otherwise it prints: 0.

```python
#!/usr/bin/env python3
# -*- coding: utf-8 -*-

import argparse
import xml.etree.ElementTree as ET
import re
import math
import sys
import itertools
from parse_svg_file import parse_svg_file

def dist_point_to_segment(pt, a, b):
    x,y = pt; x1,y1 = a; x2,y2 = b
    dx = x2 - x1; dy = y2 - y1
    if dx==0 and dy==0:
        return math.hypot(x-x1, y-y1)
    t = ((x-x1)*dx + (y-y1)*dy) / (dx*dx + dy*dy)
    t = max(0.0, min(1.0, t))
    proj_x = x1 + t*dx; proj_y = y1 + t*dy
    return math.hypot(x - proj_x, y - proj_y)

def match_segments(gt_segs, out_segs, tol=1e1):
    """Check whether each segment in gt is contained in any segment in out"""
    for (p1,p2) in gt_segs:
        ok = False
        for (q1,q2) in out_segs:
            if dist_point_to_segment(p1, q1, q2) <= tol and dist_point_to_segment(p2, q1, q2) <= tol:
                ok = True; break
        if not ok:
            return False
    return True

def external_tangents(c1, r1, c2, r2):
    """Return pairs of tangent points for the external common tangents of two circles"""
    (x1,y1), (x2,y2) = c1, c2
    dx = x2 - x1; dy = y2 - y1
    d2 = dx*dx + dy*dy
    if d2 <= (r1 - r2)**2:
        return []
    d = math.sqrt(d2)
    a = (r1 - r2) / d
    a = max(-1.0, min(1.0, a))
    b = math.sqrt(max(0.0, 1 - a*a))
    ux = dx / d; uy = dy / d
    vx = -dy / d; vy = dx / d
    n1 = (a*ux + b*vx, a*uy + b*vy)
    n2 = (a*ux - b*vx, a*uy - b*vy)
    p11 = (x1 + r1 * n1[0], y1 + r1 * n1[1])
    p12 = (x2 + r2 * n1[0], y2 + r2 * n1[1])
    p21 = (x1 + r1 * n2[0], y1 + r1 * n2[1])
    p22 = (x2 + r2 * n2[0], y2 + r2 * n2[1])
    return [(p11, p12), (p21, p22)]

def main():
    p = argparse.ArgumentParser(description="SVG output evaluation script")
    p.add_argument('ground_truth', help="Path to the ground truth SVG file")
    p.add_argument('llm_output',   help="Path to the LLM output SVG file")
    args = p.parse_args()

    try:
        gt_segs, gt_circs, gt_ells = parse_svg_file(args.ground_truth, filter_class='input_object')
```

Figure 33: Evaluation code for the bottom case in Figure 5 (1/2).

```
66          out_segs, out_circs, out_ells = parse_svg_file(args.llm_output, filter_class='output_object')
67      except Exception:
68          print("0")
69          sys.exit(0)
70
71      # Get circles from the ground truth and select the top 4 with the smallest radius
72      if len(gt_circs) < 5:
73          print("0"); sys.exit(0)
74      gt_circs_sorted = sorted(gt_circs, key=lambda x: x[1])[:4]
75
76      tol = 1e1
77      ok_all = True
78      for (c1, r1), (c2, r2) in itertools.combinations(gt_circs_sorted, 2):
79          tangents = external_tangents(c1, r1, c2, r2)
80          if not tangents:
81              ok_all = False; break
82          # It is OK if at least one of the two tangent pairs exists in the output
83          if not any(match_segments([seg], out_segs, tol) for seg in tangents):
84              ok_all = False; break
85
86      print("1" if ok_all else "0")
87
88  if __name__ == '__main__':
89      main()
```

Figure 34: Evaluation code for the bottom case in Figure 5 (2/2).

## F.2 Molecular Structure

### F.2.1 Parsing code

We present the parsing code for converting vector data into graphs representing molecular structures as shown in Figure 35.

The `parse_svg_file` function reads an SVG file and constructs a network graph using `networkx`. Nodes are extracted from `<circle>` elements, where each circle's coordinates and fill color are stored as node attributes. Line endpoints (`<line>` elements) are matched with the nearest circle positions to determine which nodes are connected. When both endpoints correspond to valid circles, an edge is created between the associated nodes. The function therefore produces an undirected graph whose topology reproduces the molecular connectivity encoded in the SVG drawing.

```python
1   #!/usr/bin/env python3
2   # -*- coding: utf-8 -*-
3
4   import math
5   import xml.etree.ElementTree as ET
6   import networkx as nx
7
8
9   def parse_svg_file(path, tol=1e-1):
10      """
11      A function that builds a graph structure from an SVG file.
12      Nodes are extracted from <circle> elements, and edges are extracted by associating
13      endpoints of <line> elements with those nodes.
14      """
15      root = ET.parse(path).getroot()
16
17      G = nx.Graph()
18      # Specify the SVG namespace
19      ns = {'svg': 'http://www.w3.org/2000/svg'}
20
21      circles = []
22      # Retrieve circle elements and register each node
23      for i, circle in enumerate(root.findall(".//svg:circle", ns)):
24          cx = float(circle.attrib['cx'])
25          cy = float(circle.attrib['cy'])
26          fill = circle.attrib.get('fill', '')
27          circles.append((cx, cy, fill, i))
28          G.add_node(i, fill=fill, pos=(cx, cy))
29
30      # Find the closest circle (node) to the given coordinates
31      def find_circle(x, y):
32          for cx, cy, fill, node_id in circles:
33              if math.hypot(cx - x, cy - y) < tol:
34                  return node_id
35          return None
36
37      # Add edges based on line elements
38      for line in root.findall(".//svg:line", ns):
39          x1 = float(line.attrib['x1'])
40          y1 = float(line.attrib['y1'])
41          x2 = float(line.attrib['x2'])
42          y2 = float(line.attrib['y2'])
43          n1 = find_circle(x1, y1)
44          n2 = find_circle(x2, y2)
45          # Add edge only if both endpoints correspond to circles (nodes)
46          if n1 is not None and n2 is not None:
47              G.add_edge(n1, n2)
48      return G
```

Figure 35: SVG parsing code.

### F.2.2 EVALUATION CODE

The script in Figure 36 evaluates whether two SVG files represent the same molecular graph. It takes two SVG file paths as input: a ground-truth structure and an SVG produced by a language model. Both files are parsed into graphs using the previously defined `parse_svg_file` function.

After parsing, the script compares the two graphs using `networkx.is_isomorphic`, where nodes are matched based on their `fill` attribute, ensuring that atoms of the same type correspond between graphs. If the two SVG-derived graphs are isomorphic under this constraint, the script outputs 1; otherwise, it outputs 0. This enables automated validation of molecular structure predictions generated from vector graphics.

```python
1  #!/usr/bin/env python3
2  # -*- coding: utf-8 -*-
3
4  import argparse
5  import xml.etree.ElementTree as ET
6  from parse_svg_file import parse_svg_file
7  import networkx as nx
8
9
10 def main():
11     p = argparse.ArgumentParser(description="SVG output evaluation script")
12     p.add_argument('ground_truth', help="File path of the ground truth SVG")
13     p.add_argument('llm_output',   help="File path of the LLM-generated SVG")
14     args = p.parse_args()
15
16     gt_G = parse_svg_file(args.ground_truth)
17     out_G = parse_svg_file(args.llm_output)
18
19     node_match = lambda n1, n2: n1['fill'] == n2['fill']
20
21     # Graph isomorphism check
22     iso = nx.is_isomorphic(gt_G, out_G, node_match=node_match)
23
24     print("1" if iso else "0")
25
26 if __name__ == '__main__':
27     main()
```

Figure 36: Evaluation code.

## G  VALIDATION OF HUMAN-SYSTEM AGREEMENT

We verify the validity of our automated evaluation code by assessing how closely human evaluations align with the code's judgments. Two human evaluators judge whether LLM outputs are correct or incorrect. One is a master's student and the other is an undergraduate student. They both major in engineering. They were not involved in our research and had no prior knowledge of the project. For each description, we present a ground-truth vector graphic along with an LLM-generated output, and ask them to evaluate its correctness. For the plane geometry task, we use all 110 instances. For the molecular structure task, highly complex structures can increase the likelihood of errors by human evaluators; therefore, we use a relatively simple set of 50 instances. We randomly select LLM outputs so that the data labeled as correct and incorrect by our code are evenly balanced. Therefore, the instances we used consist of 50% judged correct by our code and 50% judged incorrect. We include all output formats: TikZ, SVG, and EPS.

Table 15 shows the percentage of agreement and Cohen's Kappa scores between the human evaluators and our code. These results demonstrate very high agreement rates, indicating the reliability of our evaluation approach.

Table 15: The percentage of agreement and Cohen's Kappa scores between the human evaluators and our code.

| Annotator 1 | Plane geometry | | | Molecular structure | | |
|---|---|---|---|---|---|---|
| | TikZ | SVG | EPS | TikZ | SVG | EPS |
| Percentage of agreement | 97.3% | 95.5% | 95.5% | 98.0% | 98.0% | 98.0% |
| Cohen's Kappa | 0.946 | 0.909 | 0.909 | 0.960 | 0.960 | 0.960 |

| Annotator 2 | Plane geometry | | | Molecular structure | | |
|---|---|---|---|---|---|---|
| | TikZ | SVG | EPS | TikZ | SVG | EPS |
| Percentage of agreement | 96.4% | 99.1% | 95.5% | 98.0% | 100.0% | 96.0% |
| Cohen's Kappa | 0.927 | 0.982 | 0.909 | 0.960 | 1.000 | 0.920 |

