# OpenReview forum: "Towards the Generation of  Structured Scientific Vector Graphics with Large Language Models"
_ICLR.cc/2026/Conference — Submitted to ICLR 2026_

### Official Review · Reviewer_r4TZ · 2025-10-29

**Soundness:** 4
**Presentation:** 4
**Contribution:** 3
**Rating:** 4
**Confidence:** 4

**Summary:**

This paper fills the gap of evaluating the structural correctness of text-guided scientific figure generation, which previous work overlooks. Evaluating structural correctness is not easy to do with fully automated methods. Instead, the authors focus on a specific subset of scientific figures (plane geometry and molecular structure). They derive a benchmark of 410 text-figure pairs and implement rule-based programs that evaluate the correctness of generated outputs. The authors evaluate a range of existing fine-tuned and general-purpose models on this benchmark using three output formats (TikZ, SVG, and EPS). In addition, they introduce a new task-specific prompting method that can help improve performance.

**Strengths:**

* The paper is very well written and easy to understand.
* Structural correctness is an important property for the evaluation of scientific figure generation that is often omitted from evaluations due to the difficulty of assessing it. The provided benchmark fills this gap and will be useful for future work.
* The evaluation compares a wide range of models across three formats, which provides interesting insights.
* The provided prompting technique may be useful for future work and applications.

**Weaknesses:**

* Table 3 is an interesting approximation of potential occurrences in the training data, but a TikZ graphic doesn't necessarily have to start with `\documentclass[tikz]`.
* The rule-based programs seem central to the benchmark, yet no details or example snippets are provided in the paper, which would have been insightful for assessing how well they work or whether they are brittle with failure cases.
* Although the benchmark will be very useful when released, there is very little technical novelty in the paper, and while the introduced prompting technique seems to work well and leads to additional improvements, it is hardly exciting.
* The authors motivate their benchmark by stating that human evaluation doesn't scale well (l.129), but the creation of the benchmark still requires heavy manual curation (l.209ff), so it still doesn't scale well.
* The performance of fine-tuned models (AutomaTikZ, TikZero) seems surprisingly low. Have the authors ensured that the provided prompts are in the format the models expect? At least the example prompts provided in the appendix are not in the correct format.

**Questions:**

* In l.242, it says that parts of the output are provided to the models as input. How is this done exactly, as this is not clear from the prompt examples in the appendix? Furthermore, how is this provided for models that are fine-tuned and do not accept general-purpose prompts?
* Why are the scores of the TikZ models in Table 4 provided in the SVG column?

---

> ### Author Response · Authors · 2025-11-21
> **Official Comment by Authors**
>
> We sincerely appreciate your valuable feedback. You seem very knowledgeable about this topic, and your insightful comments have been extremely helpful in improving the paper. We have revised the manuscript according to your suggestions and look forward to receiving any further feedback you may have.
>
> **[W1. Table 3 is an interesting approximation of potential occurrences in the training data, but a TikZ graphic doesn't necessarily have to start with `\documentclass[tikz]`.]**
>
> Thank you for this insightful comment. We realized that TikZ code does not necessarily begin with `\documentclass[tikz]`. In response, we replaced this keyword with `tikzpicture` as a TikZ-specific token, which increased the number of Google search hits from 154K to 272K. We have updated **Table 3** accordingly.
>
> **[W2. The rule-based programs seem central to the benchmark, yet no details or example snippets are provided in the paper, which would have been insightful for assessing how well they work or whether they are brittle with failure cases.]**
>
> Thank you for this valuable suggestion. We have added “DETAILS OF THE EVALUATION CODE” to **Appendix F**, which we believe clarifies our evaluation framework and improves its trustworthiness and reproducibility
>
> **[W3.1. Although the benchmark will be very useful when released, there is very little technical novelty in the paper]**
>
> We believe that technical novelty is not a necessary criterion for evaluating the value of a benchmark. In NLP, several widely used benchmarks, such as VGBench (EMNLP 2024) and Scimage (ICLR2025), contain little technical novelty yet have been accepted to top-tier conferences. Their value lies in (1) introducing new evaluation criteria and (2) providing analysis based on those criteria. Our work offers a new metric for structural correctness in vector graphics generation, along with analyses such as the impact of reasoning and format differences. We would appreciate it if our work could be reassessed from these two perspectives.
>
> **[W3.2 while the introduced prompting technique seems to work well and leads to additional improvements, it is hardly exciting.]**
>
> We are sorry that you do not find our prompting technique particularly exciting, but we believe many researchers may consider it valuable. As summarized in **Section 2 (RELATED WORK)**, prompting strategies that improve LLM performance have recently been widely studied. From this broader community perspective, we believe our method could be of interest beyond an individual reviewer’s preference. We would greatly appreciate a reevaluation based on its potential value to the research community.
>
> **[W4. The authors motivate their benchmark by stating that human evaluation doesn't scale well (l.129), but the creation of the benchmark still requires heavy manual curation (l.209ff), so it still doesn't scale well.]**
>
> Our intention was to emphasize that our benchmark is scalable with respect to the number of models and experimental variations: once the benchmark is set up, new models or prompting strategies can be evaluated automatically, without incurring further manual effort, unlike human evaluation which must be repeated each time. To clarify this point, we have revised line 135 accordingly.
>
> **[W5.1. The performance of fine-tuned models (AutomaTikZ, TikZero) seems surprisingly low.]**
>
> There are several possible reasons for the low performance. First, AutomaTikZ and TikZero+ assume **resampling** until no compilation error occurs, whereas in our experiments only a single generation was allowed, and any result with a compilation error was counted as incorrect. Note that this rule is applied consistently across all models. Second, our **metric (accuracy)** is a strict indicator: even a small mistake is considered incorrect. In contrast, both AutomaTikZ and TikZero+ evaluate performance using looser indicators such as image or code similarity. To verify these points, we conducted experiments allowing up to 10 resamplings and additionally evaluated image/code similarity metrics. The results are shown in **Tables 7 and 10**, and they demonstrate that the performance of TikZero+ becomes comparable to other non-reasoning models.

---

> ### Author Response · Authors · 2025-11-21
> **Official Comment by Authors**
>
> **[W5.2. Have the authors ensured that the provided prompts are in the format the models expect? At least the example prompts provided in the appendix are not in the correct format.]**
>
> Our understanding is that both AutomaTikZ and TikZero are capable of handling diverse prompt formats. For example, the training data for DaTikZ v3 contains captions such as:
>
> > `Fit around nodes and edges`
>
> > `I know that I can use the fit library of tikz to get a fitted background behind some nodes.
> How can I get the fit to be around edges as well?
> My current code:
> \begin{tikzpicture}[prefix=fig/,
> state/.style={circle,draw,thick},
> hmm/.style={draw,rectangle}]`
>
> > `\node[state] (a1) at (0,0) {} edge [loop above,thick] () ;
> > \node[state] (a2) at (1,0) {} edge [loop above,thick] () edge[<-,thick] (a1);
> > \node[state] (a3) at (2,0) {} edge [loop above,thick] () edge[<-,thick] (a2);`
>
> > `\node[hmm,fit=(a1) (a2) (a3)] (a) {};
> > \end{tikzpicture}`
>
> > `This puts the top of the background box straight through the loops. How can I fit the box nicely around the nodes and edges (so the top of the loop must be a point somehow in the fit).`
>
> This example clearly shows that prompts may contain both partial TikZ code and instructions to modify it. Given that the models are trained on such data, it is reasonable to assume that they are capable of handling our prompts as well. If you believe there exists a specific “correct format,” we would greatly appreciate it if you could describe this format as concretely as possible.
>
> **[Q1.1. In l.242, it says that parts of the output are provided to the models as input. How is this done exactly, as this is not clear from the prompt examples in the appendix?]**
>
> This is already illustrated in Figures 7, 8, and 9 in the appendix. The input vector is explicitly included in the prompt, indicated by the tags [TikZ], [SVG], and [EPS]. To make this clearer, we have revised the explanation in **Appendix B Detailed Prompts Used in Our Experiments**.
>
> **[Q1.2. Furthermore, how is this provided for models that are fine-tuned and do not accept general-purpose prompts?]**
>
> We use the same prompts across all models. As explained above, we believe that AutomaTikZ and TikZero are capable of handling general-purpose prompts, given the diversity of prompts in their training data.
>
> **[Q2. Why are the scores of the TikZ models in Table 4 provided in the SVG column?]**
>
> We apologize for the confusion. This was entirely a mistake on our side. We have corrected **Table 4** and moved the scores of the fine-tuned models to the TikZ column.

---

> > ### Comment · Reviewer_r4TZ · 2025-11-25
> >
> > I thank the authors for their detailed rebuttal. They have provided comprehensive answers to my questions and addressed the raised weaknesses. I particularly appreciate their clarifications regarding the evaluation of fine-tuned models and the relevance of their work (responses to weaknesses 3-5). In light of this, I believe their work should be reassessed as requested in the following comment:
> >
> > > In NLP, several widely used benchmarks [..] contain little technical novelty yet have been accepted to top-tier conferences. Their value lies in (1) introducing new evaluation criteria and (2) providing analysis based on those criteria. Our work offers a new metric for structural correctness in vector graphics generation, along with analyses such as the impact of reasoning and format differences. We would appreciate it if our work could be reassessed from these two perspectives.
> >
> > I believe that the introduced metric for structural correctness in vector graphics generation has potential. However, I believe that proposing a new metric should be backed by evidence from human evaluation, a concern that appears to be shared by reviewer Q9nD. Do the authors have an estimate for when the promised human evaluation will be completed? If it supports their findings, I would be happy to raise my score.

---

> > > ### Author Response · Authors · 2025-11-26
> > > **Official Comment by Authors**
> > >
> > > Thank you for your prompt response! We appreciate your willingness to reassess the value of our benchmark.
> > >
> > > We are currently preparing for a human evaluation and expect to obtain the results within approximately three days. Once they become available, we will share them with you promptly and would greatly appreciate any further feedback.

---

> ### Author Response · Authors · 2025-11-28
> **Official Comment by Authors**
>
> Thank you for waiting for our response. The human evaluation is now complete, and I have updated the paper accordingly. Below is the summary. We hope it addresses your concerns.
>
> **[I believe that proposing a new metric should be backed by evidence from human evaluation]**
>
> We verify the validity of our automated evaluation code by assessing how closely human evaluations align with the code’s judgments. The following table shows the percentage of agreement and Cohen’s Kappa scores between the human evaluators and our code. These results demonstrate very high agreement rates, indicating the reliability of our evaluation approach. We have presented this result in **Table 15**.
>
> | **Annotator 1 (Plane geometry)** |  |  |  | **(Molecular structure)**||||
> | --- | --- | --- | --- | --- | --- | --- | --- |
> |  | TikZ | SVG | EPS | | TikZ | SVG | EPS |
> | Percentage of agreement | 97.3% |95.5% |95.5% ||98.0% |98.0% |98.0%|
> | Cohen's Kappa | 0.946 | 0.909 | 0.909 | |  0.960| 0.960| 0.960|
>
> | **Annotator 2 (Plane geometry)** |  |  |  | **(Molecular structure)**||||
> | --- | --- | --- | --- | --- | --- | --- | --- |
> |  | TikZ | SVG | EPS | | TikZ | SVG | EPS |
> | Percentage of agreement |  96.4% |99.1% |95.5% ||98.0% |100.0% |96.0%|
> | Cohen's Kappa | 0.927 | 0.982 | 0.909 | | 0.960 |1.000 |0.920|

---

### Official Review · Reviewer_vtgB · 2025-10-29

**Soundness:** 3
**Presentation:** 2
**Contribution:** 4
**Rating:** 8
**Confidence:** 2

**Summary:**

The paper introduces **SSVG-Bench**, a benchmark to evaluate **structural correctness** in scientific vector-graphics generation (TikZ/SVG/EPS), provides **automatic structure-aware evaluators** for plane geometry and molecular structures, benchmarks many LLMs, and proposes **LOOP**, a prompting workflow that improves accuracy, especially in SVG.

**Strengths:**

- **Clear problem motivation:** moves beyond visual/code similarity to **structure-aware** evaluation.
- **New benchmark & tooling:** SSVG-Bench (two tasks; three formats) with **Python scripts for structural checks**
- **Broad, timely evaluation:** compares fine-tuned TikZ models and recent LLMs; reveals the importance of **reasoning modes**.
- **Actionable insight on formats:** compelling evidence that **SVG > TikZ/EPS** for LLM reasoning; novel angle likely to influence future work.
- **LOOP** (information/relationship extraction → reasoning → code) yields **consistent gains** over popular CoT variants.
- Prompts, task setup, and evaluation logic are described in detail; many examples illustrate successes/failures.

**Weaknesses:**

- **Evaluation blind spots:** plane-geometry scorer does not penalize **extraneous elements**; results may overstate correctness in cluttered outputs.
- **Bond order ignored:** molecular task collapses bond multiplicity; risks awarding correctness to chemically different graphs.
- **Potential data leakage:** plane-geometry items sourced from Wikipedia/SVG Commons; large web-trained LLMs may have seen near-identical diagrams/captions.
- **Scope & scale:** overall size (1,230 items) is moderate; per-topic diversity (e.g., non-Euclidean, circuits, algorithmic flowcharts) could be broader.
- **Ablations on LOOP:** helpful but could be deeper (e.g., remove each stage, vary decomposition granularity, measure latency/cost).

**Questions:**

See weaknesses.

---

> ### Author Response · Authors · 2025-11-21
> **Official Comment by Authors**
>
> We sincerely appreciate your valuable feedback and are grateful for your positive evaluation of our paper. Our responses to your comments are provided below, and we hope they adequately address your concerns.
>
> **[W1. Evaluation blind spots: plane-geometry scorer does not penalize extraneous elements; results may overstate correctness in cluttered outputs.]**
>
> Thank you for your valuable feedback. An explanation of this issue is already provided in **Appendix C “LIMITATIONS OF OUR AUTOMATIC EVALUATION FRAMEWORK”**. We do not penalize unnecessary elements because it is often non-trivial to determine whether an additional element is truly unnecessary. For instance, the output from Gemini 2.5 Flash reasoning in Figure 13 includes circles not anticipated in the ground-truth, but these represent intersections and the circle center, and they do not hinder the explanation.  Because in practical scenarios it is usually easier for humans to remove unnecessary elements than to create necessary ones from scratch, we do not currently view this limitation as a major issue.
>
> **[W2. Bond order ignored: molecular task collapses bond multiplicity; risks awarding correctness to chemically different graphs.]**
>
> Thank you very much for this valuable comment. We recognize that ignoring bond order may lead to chemically distinct graphs being judged as equivalent, and we agree that this is a potential limitation. In our current setting, however, we intentionally relaxed bond multiplicity because certain molecular motifs, such as aromatic systems (e.g., benzene rings), can present ambiguity in how bond orders are defined or represented. This simplification allowed us to focus first on structural topology generation without being hindered by representational variability. We regard incorporating accurate bond order evaluation as an important next step, and improving this aspect remains a key direction for our future work.
>
> **[W3. Potential data leakage: plane-geometry items sourced from Wikipedia/SVG Commons; large web-trained LLMs may have seen near-identical diagrams/captions.]**
>
> Thank you for your valuable feedback. Because our dataset is sourced from Wikipedia, we cannot entirely exclude the possibility of data leakage. Nevertheless, even if an LLM already contains knowledge from Wikipedia, the knowledge encoded in a single model remains constant. For this reason, we believe that our prompting-based evaluation, particularly the within-model comparison, remains sufficiently meaningful.
>
> **[W4. Scope & scale: overall size (1,230 items) is moderate; per-topic diversity (e.g., non-Euclidean, circuits, algorithmic flowcharts) could be broader.]**
>
> Thank you for your valuable feedback. The diversity of our benchmark may be limited, but our goal is to provide a minimal yet representative benchmark that captures two fundamental forms of structural correctness. First, the ability to generate plane geometry structures is fundamental for many real-world applications, including physics illustrations, engineering diagrams, and architectural blueprints. The example shown in Figure 1 demonstrates precisely such an application. Second, generating molecular structures requires correctly producing graph-based structures. Many domains, including algorithm flowcharts, circuit designs, and biological pathways, require strict adherence to graph topology and structural constraints. Our intention is to capture both geometrical and graph-topological constraints, which we believe form fundamental building blocks for diverse structure-oriented generative applications.
>
> **[W5. Ablations on LOOP: helpful but could be deeper (e.g., remove each stage, vary decomposition granularity, measure latency/cost).]**
>
> Thank you for your valuable feedback. We conducted an ablation study to remove each step of LOOP and presented the result in **Table 14**. The original prompt achieves the best performance, clearly demonstrating that each component contributes to the improvement.

---

### Official Review · Reviewer_8knp · 2025-10-31

**Soundness:** 3
**Presentation:** 4
**Contribution:** 3
**Rating:** 4
**Confidence:** 4

**Summary:**

This paper addressed the structural correctness issue in the LLM's vector graphics generation. It provides a benchmark called SSVG-Bench consisting of two types of tasks, plane geometry task and molecular structure task, along with novel evaluating scripts to evaluate the structural correctness of the generated images. It performs a comprehensive benchmarking and analysis of existing models on the proposed benchmark, revealing the poor performance of LLM and key feature that might enhance a LLM's capability of generating correct vector graphics. Finally, it proposed LLM-Oriented Orchestration Prompting (LOOP), a method that enhances the accuracy of vector graphics generation in LLM. Experiments result shows that the proposed LOOP can improve the performance in terms of structural correctness on it's proposed benchmark.

**Strengths:**

1. The proposed evaluation method is novel. Instead of using common metrics in computer vision, the author proposed using scripts to evaluate the correctness of scientific vector graphics. This makes the evaluation process more explainable and accurate.
2. The evaluation in Table 4 is comprehensive, covering all recent SOTA models.
3. There's a significant improvement in performance with the proposed LOOP strategy.

**Weaknesses:**

1. The diversity of the proposed benchmark is greatly limited. It only contains two kinds of tasks, plane geometry task (with only 5 elements) and molecular structure task. Both tasks have a very clear and fixed path to solve, and therefore, might not be able to test the model's generalizability on all other tasks requiring structural correctness.
2. Table 1 missed lots of recent benchmark (even those benchmark are mentioned in the text from L105-L107, and lots of them are larger than the proposed benchmark). For example, the generation evaluation suite of VGBench contains 5845 instances in total, and SVGEditBench has 1366 instances in total. Not comparing them in Table 1 is unfair.
3. Molecular structure is an extremely specialized task. It not only evaluates the ability to generate graphics, but also evaluate the model's understanding of IUPAC name and the model's chemistry knowledge. The requirement for such specialized knowledge presents a bias so that the results will be better for models with chemistry knowledge than models without, while the chemistry knowledge is not usually related to the model's ability to generate vector graphics in general.
4. There is no evaluation on the proposed LOOP strategy's performance on other vector graphics generation benchmark. Evaluating it on other commonly used benchmark will ensure the method's generalizability.

**Questions:**

1. How to ensure the robustness of the Python script for Pattern 2? Are we using a different script for each case or using the same script for all cases?
2. In L241, it's mentioned that "In Pattern 1, the correct output can be uniquely determined", Why it can be uniquely determined? Even with the given element (in black), the remaining element (in red) can still have multiple ways of expression. For example, the circle can be represented as circle using `<circle>`, or represented as curve using `<path>` in SVG.

---

> ### Author Response · Authors · 2025-11-21
> **Official Comment by Authors**
>
> We sincerely appreciate your valuable feedback. Your comments precisely address the core ideas of our paper and have been extremely helpful in further refining and highlighting the value of our work. Below, we provide our responses to your concerns, and we hope they sufficiently address your questions.
>
> **[W1. The diversity of the proposed benchmark is greatly limited. It only contains two kinds of tasks, plane geometry task (with only 5 elements) and molecular structure task. Both tasks have a very clear and fixed path to solve, and therefore, might not be able to test the model's generalizability on all other tasks requiring structural correctness.]**
>
> Thank you for your valuable feedback. The diversity of our benchmark may be limited, but our goal is to provide a minimal yet representative benchmark that captures two fundamental forms of structural correctness. First, the ability to generate plane geometry structures is fundamental for many real-world applications, including physics illustrations, engineering diagrams, and architectural blueprints. The example shown in Figure 1 demonstrates precisely such an application. Second, generating molecular structures requires correctly producing graph-based structures. Many domains, including algorithm flowcharts, circuit designs, and biological pathways, require strict adherence to graph topology and structural constraints. Our intention is to capture both geometrical and graph-topological constraints, which we believe form fundamental building blocks for diverse structure-oriented generative applications.
>
> **[W2. Table 1 missed lots of recent benchmark (even those benchmark are mentioned in the text from L105-L107, and lots of them are larger than the proposed benchmark). For example, the generation evaluation suite of VGBench contains 5845 instances in total, and SVGEditBench has 1366 instances in total. Not comparing them in Table 1 is unfair.]**
>
> Thank you for pointing this out. Following your suggestion, we have expanded **Table 1** to include additional benchmarks. We would like to clarify that our benchmark is designed specifically for scientific vector generation, whereas the added benchmarks are not primarily designed for scientific vector graphics. Nonetheless, we agree that presenting them provides a more comprehensive and fair comparison.
>
> **[W3. Molecular structure is an extremely specialized task. It not only evaluates the ability to generate graphics, but also evaluate the model's understanding of IUPAC name and the model's chemistry knowledge. The requirement for such specialized knowledge presents a bias so that the results will be better for models with chemistry knowledge than models without, while the chemistry knowledge is not usually related to the model's ability to generate vector graphics in general.]**
>
> We agree that the molecular structure task inherently involves chemistry-specific knowledge, and therefore cross-model comparison may introduce bias toward models with greater domain expertise. However, the chemistry knowledge of a single model remains constant across our experiments, meaning the task is still valuable for evaluating within-model performance comparisons, especially regarding the effectiveness of our prompting strategies. Thus, we believe the task remains meaningful for assessing relative model improvements.
>
> **[W4.There is no evaluation on the proposed LOOP strategy's performance on other vector graphics generation benchmark. Evaluating it on other commonly used benchmark will ensure the method's generalizability.]**
>
> Thank you for your suggestion. Following your recommendation, we evaluated the robustness of the proposed LOOP strategy using the DaTikZ v3 dataset. The results are presented in **Figure 13**, and they show that LOOP consistently improves performance even on DaTikZ v3, further demonstrating its robustness and generalizability.
>
> **[Q1. How to ensure the robustness of the Python script for Pattern 2? Are we using a different script for each case or using the same script for all cases?]**
>
> For Pattern 2, we designed case-specific evaluation code for each instance to ensure precise and rigorous structural correctness assessment. We have added two example evaluation scripts in **Figures 31, 32, 33, and 34**, which clarify our evaluation framework further.

---

> > ### Author Response · Authors · 2025-11-21
> > **Official Comment by Authors**
> >
> > **[Q2. In L241, it's mentioned that "In Pattern 1, the correct output can be uniquely determined", Why it can be uniquely determined? Even with the given element (in black), the remaining element (in red) can still have multiple ways of expression. For example, the circle can be represented as circle using `<circle>`, or represented as curve using `<path>` in SVG.]**
> >
> > We appreciate this detailed observation. In our inspection, we did not identify instances where circles were expressed using `<path>`. In contrast, we observed many variations for lines, which may be represented using `<line>`, `<polygon>`, `<polyline>`, `<path>`, or even `<rect>`. To account for such variability, we carefully design our SVG parsing codes to support a wide range of possible representations. To improve transparency, we now include our parsing code in **Figures 23, 24, 25, 26, and 27**. While we cannot guarantee coverage of every possible edge case, we have designed the parsing framework to handle as many realistic variations as possible.

---

> > > ### Comment · Reviewer_8knp · 2025-11-23
> > >
> > > Thank you for the response. I have a follow-up question based on your response to Q1 and Q2. How the SVG parsing codes are generated? Which measure in the benchmark constructing process is taken to ensure they have considered the variations for geometry elements?

---

> > > > ### Author Response · Authors · 2025-11-24
> > > > **Official Comment by Authors**
> > > >
> > > > Thank you for your prompt response! Below is our reply to your additional question. We hope it addresses your concerns.
> > > >
> > > > **[How the SVG parsing codes are generated? Which measure in the benchmark constructing process is taken to ensure they have considered the variations for geometry elements?]**
> > > >
> > > > **(Parsing code for the plane geometry task.)** The SVG parsing code was not automatically generated, but carefully engineered based on empirical observations of both ground-truth and LLM-generated SVG.
> > > >
> > > > **(1) Ground-truth Analysis**
> > > >
> > > > We first manually inspected all ground-truth SVG files in our dataset to enumerate the geometric element types they use. Based on this inspection, we implemented a parser that supports every element type actually appearing in the annotated data (i.e., `<line>`, `<circle>`, `<ellipse>`, `<polygon>`, `<polyline>`).
> > > >
> > > > **(2) Exploratory Evaluation of LLM-Generated SVG**
> > > >
> > > > To investigate how LLMs express geometry differently from ground truth, we then generated a large variety of SVG samples using multiple LLMs. Through this analysis, we observed additional representation patterns absent from the dataset, such as:
> > > >
> > > > - shapes represented by `<rect>` or `<path>` instead of `<polygon>`
> > > > - ellipses implicitly expressed using `<circle>` with `transform` attributes
> > > >
> > > > **(3) Iterative Extension of the Parser**
> > > >
> > > > Based on these empirical findings, we iteratively extended the parser to handle such alternative representations robustly. This procedure ensured that our benchmark does not rely solely on canonical SVG syntax, but instead reflects realistic variations seen in practice, particularly those produced by LLMs.
> > > >
> > > > Although it is impossible to cover all hypothetical edge cases in SVG specification, our parsing framework covers all observed cases from both ground truth and LLM outputs during the benchmark construction process, making it suitable for fair evaluation.
> > > >
> > > > **(Parsing code for the molecular structure task.)** For the molecular structure generation task, we explicitly provide an example SVG format in the prompt, specifying that atoms should be represented by `<circle>` and bonds by `<line>`. Therefore, we implemented the parser to handle only `<circle>` and `<line>` elements and ignore all others.

---

> > > > > ### Comment · Reviewer_8knp · 2025-11-25
> > > > >
> > > > > Thank you very much! I think this is a novel way to evaluate the correctness of vector graphics. Therefore, I increased my ratings to 6.

---

> > > > > > ### Author Response · Authors · 2025-11-26
> > > > > > **Official Comment by Authors**
> > > > > >
> > > > > > Thank you for raising our rating! We truly appreciate your valuable and insightful comments. We believe that our paper has been greatly improved as a result of your feedback.

---

### Official Review · Reviewer_Q9nD · 2025-10-31

**Soundness:** 2
**Presentation:** 3
**Contribution:** 2
**Rating:** 2
**Confidence:** 4

**Summary:**

The paper introduced SSVG-Bench, a benchmark for generating structured scientific vector graphics from text, covering plane geometry and molecular structures across three vector formats (TikZ, SVG, and EPS). It applied Python-based automatic accuracy evaluation and reports results for multiple LLMs. The paper also proposed a prompting method, LOOP, which yields measurable gains on the benchmark.

**Strengths:**

1. Clear, well-structured writing with concrete, illustrative examples that make the task and setup easy to follow.
2. The automated molecular evaluation via graph isomorphism is sensible.
3. Broad coverage of vector formats (TikZ, SVG, EPS) and a diverse model suite, comparing both reasoning and non-reasoning LLMs.
4. The proposed prompting method (LOOP) is simple to implement and yields consistent gains.

**Weaknesses:**

W1. Insufficient technical detail for automatic plane-geometry evaluation (Pattern 1/2): missing normalization pipeline, tolerance settings, red-element extraction, alignment strategy, and failure modes. This undermines the trustworthiness and reproducibility of the reported Accuracy.


W2. Table 4 inconsistency: AutomaTikZ and TikZero+ are trained on TikZ, yet the TikZ column is empty while SVG numbers are reported. Is this a mistake or a data-availability issue? Please clarify/correct.


W3. Over-reliance on a single binary metric (Accuracy): no complementary automatic metrics or human evaluation, leading to an overly coarse assessment; near-misses and completely wrong outputs are both scored 0. Consider adding metrics used in related work TikZero+ (e.g., DreamSim, KID, CLIPScore, code-level CrystalBLEU and TEX Edit Distance, Mean Token Efficiency) and geometry-specific measures (e.g., how many elements are correctly covered?)


W4. Narrow evaluation of the prompting strategy: results are shown primarily on GPT/Gemini and mostly as Accuracy; broader model coverage and multi-metric reporting are needed.
Lack of human evaluation: no user study to validate automatic metrics, resolve borderline cases, or assess readability/usability of the generated diagrams.

**Questions:**

Q1. Pattern taxonomy: How exactly categorize plane-geometry items into Pattern 1 (unique solution) vs Pattern 2 (multiple valid solutions), and what is the split (%) across the 110 examples?
Q2. Pattern 2 evaluation cost: Does Pattern 2 require case-specific code per instance?
Q3. Prompting generality: LOOP appears to be a fixed “think” scheme—do its gains differ between reasoning and non-reasoning models? Q4. Please provide results on more models and with more metrics (as above)

---

> ### Author Response · Authors · 2025-11-21
> **Official Comment by Authors**
>
> We sincerely appreciate your valuable feedback. You have clearly indicated the direction for improvement, and your suggestions are very helpful for improving our manuscript. We have revised the paper accordingly and look forward to hearing your further comments.
>
> **[W1. Insufficient technical detail for automatic plane-geometry evaluation (Pattern 1/2): missing normalization pipeline, tolerance settings, red-element extraction, alignment strategy, and failure modes. This undermines the trustworthiness and reproducibility of the reported Accuracy.]**
>
> Thank you for your insightful comment. Following this suggestion, we have added “DETAILS OF THE EVALUATION CODE” section in **Appendix F**. We believe this addition clarifies our evaluation framework and enhances its trustworthiness and reproducibility.
>
> **[W2. Table 4 inconsistency: AutomaTikZ and TikZero+ are trained on TikZ, yet the TikZ column is empty while SVG numbers are reported. Is this a mistake or a data-availability issue? Please clarify/correct.]**
>
> We apologize for the confusion. This was entirely our mistake. We have corrected **Table 4** and moved the fine-tuned models’ scores to the TikZ column.
>
> **[W3. Over-reliance on a single binary metric (Accuracy): no complementary automatic metrics or human evaluation, leading to an overly coarse assessment; near-misses and completely wrong outputs are both scored 0. Consider adding metrics used in related work TikZero+ (e.g., DreamSim, KID, CLIPScore, code-level CrystalBLEU and TEX Edit Distance, Mean Token Efficiency) and geometry-specific measures (e.g., how many elements are correctly covered?)]**
>
> Thank you for the valuable suggestion. In response, we incorporated multiple complementary automatic metrics and evaluated all models accordingly. The results are presented in **Tables 7, 8, 9, 10, 11, and 12**. When focusing on the **Average** scores across all metrics, we consistently observe that (1) reasoning models outperform non-reasoning models, and (2) our proposed LOOP improves performance in most cases. We believe these additions significantly enhance the reliability of our findings.
>
> **[W4.1. Narrow evaluation of the prompting strategy: results are shown primarily on GPT/Gemini and mostly as Accuracy; broader model coverage and multi-metric reporting are needed.]**
>
> Thank you for your insightful suggestion. Following it, we extended our study by applying LOOP to two additional reasoning models (DeepSeek-V3.2 reasoning and Claude Opus 4.1 thinking) and two additional non-reasoning models (Gemini 2.5 Flash non-reasoning and GPT-5 Chat). Their results are also reported in **Tables 7, 8, 9, 10, 11, and 12**. Our findings show that LOOP improves performance in many cases, demonstrating its robustness.
>
> **[W4.2 Lack of human evaluation: no user study to validate automatic metrics, resolve borderline cases, or assess readability/usability of the generated diagrams.]**
>
> Thank you very much for your valuable feedback. Although we did not conduct a user study, we manually verified the evaluation results to validate the automatic metrics and confirmed that the outcomes were reliable. To properly assess borderline cases, we also introduced additional evaluation indicators as you suggested. We agree that readability and usability are important aspects; however, the primary goal of this study is to improve the structural correctness of the generated diagrams. Therefore, we do not address readability and usability evaluation in this work, and we leave it as an important direction for future research.
>
> **[Q1. Pattern taxonomy: How exactly categorize plane-geometry items into Pattern 1 (unique solution) vs Pattern 2 (multiple valid solutions), and what is the split (%) across the 110 examples?]**
>
> We manually inspected each text description and categorized them into Pattern 1 or Pattern 2 accordingly. Among 110 cases, 94 belong to Pattern 1 and 16 belong to Pattern 2. We have added this explanation to line 259.
>
> **[Q2. Pattern 2 evaluation cost: Does Pattern 2 require case-specific code per instance?]**
>
> Yes. For Pattern 2, we designed case-specific evaluation code for each instance to ensure precise and rigorous structural correctness assessment. We have added two example evaluation scripts in **Figures 31, 32, 33, and 34**, which clarify our evaluation framework further.
>
> **[Q3. Prompting generality: LOOP appears to be a fixed “think” scheme—do its gains differ between reasoning and non-reasoning models?]**
>
> Thank you for the thoughtful question. As mentioned in our response to W4.1, we evaluated LOOP on two additional non-reasoning models (Gemini 2.5 Flash non-reasoning and GPT-5 Chat), and their results are provided in **Tables 7, 8, 9, 10, 11, and 12**. LOOP yields improvements in many cases, demonstrating its applicability to non-reasoning models as well.

---

> > ### Author Response · Authors · 2025-11-21
> > **Official Comment by Authors**
> >
> > **[Q4. Please provide results on more models and with more metrics (as above])**
> >
> > Thank you for your valuable comments. As explained above, we added evaluations using more models and additional metrics. The corresponding results are provided in **Tables 7, 8, 9, 10, 11, and 12**, which we believe greatly enhance the reliability of our experiments.

---

> ### Comment · Reviewer_Q9nD · 2025-11-23
>
> Thanks, I appreciate the authors' additions and increase to 4. Still the lack of human evaluation is concerning, as it is known that automatic metrics do not always match well with automatic ones. The paper might benefit from another revision in the next conference.

---

> > ### Author Response · Authors · 2025-11-23
> > **Official Comment by Authors**
> >
> > Thank you very much for your prompt response, and we truly appreciate you raising our rating!
> >
> > We realized that we had misunderstood your comments regarding human evaluation. As you pointed out, automated evaluation metrics should be assessed based on how well they align with human judgment. We will add a human evaluation as soon as possible.

---

> > > ### Author Response · Authors · 2025-11-28
> > > **Official Comment by Authors**
> > >
> > > Thank you for waiting for our response. The human evaluation is now complete, and I have updated the paper accordingly. Below is the summary. We hope it addresses your concerns.
> > >
> > > **[Still the lack of human evaluation is concerning, as it is known that automatic metrics do not always match well with automatic ones.]**
> > >
> > > We verify the validity of our automated evaluation code by assessing how closely human evaluations align with the code’s judgments. The following table shows the percentage of agreement between the human evaluators and our code. These results demonstrate very high agreement rates, indicating the reliability of our evaluation approach. We have presented this result in **Table 15**.
> > >
> > > |  | TikZ | SVG | EPS |
> > > | --- | --- | --- | --- |
> > > | Percentage of agreement | 96.8 % | 97.3 % | 95.5 % |

---

> > > > ### Comment · Reviewer_Q9nD · 2025-11-28
> > > >
> > > > Thanks for the updates. I just screened your text before Table 15. I have more questions:
> > > >
> > > > (i) demographics of annotators: who are they? what is their expertise? (honestly, I fear you have taken two co-authors, with all their biases about how the automatic evaluation works...)
> > > >
> > > > (ii) cost and kappa agreement of human annotators; how many instances were annotated?
> > > >
> > > > (iii) why do you report percentage agreement instead of kappa agreement? Do your annotators and your code both make a binary choice?
> > > >
> > > > (iv) do you have 50% instances labeled as 0 and 50% as 1 according to your code?
> > > >
> > > > And a final question which I only noticed now: In Tables 7, etc., in which you report averages, do you take into account that for some metrics lower is better, while for others, higher is better?

---

> ### Author Response · Authors · 2025-11-28
> **Official Comment by Authors**
>
> Thank you for your prompt response! Below are our replies to your additional questions. We hope these replies address your concerns.
>
> **[(i) demographics of annotators: who are they? what is their expertise? (honestly, I fear you have taken two co-authors, with all their biases about how the automatic evaluation works...)]**
>
> The human evaluators consisted of one master's student and one undergraduate student, both majoring in engineering. They were not involved in our research and had no prior knowledge of the project, ensuring that they did not share any potential biases with the authors. To be clear, neither evaluator is a co-author.
>
> **[(ii) cost and kappa agreement of human annotators; how many instances were annotated?]**
>
> We use 110 instances and 50 instances for the plane geometry task and for the molecular structure task, respectively.
>
> **[(iii) why do you report percentage agreement instead of kappa agreement?]**
>
> Thank you for your valuable feedback. Since kappa agreement is an important metric, we calculated the Cohen’s Kappa between the judgments of the two annotators and our code. The results are shown below. The high Cohen’s Kappa scores demonstrate the strong reliability of our evaluation system. We have added these results to **Table 15**.
>
> | **Annotator 1 (Plane geometry)** |  |  |  | **(Molecular structure)**||||
> | --- | --- | --- | --- | --- | --- | --- | --- |
> |  | TikZ | SVG | EPS | | TikZ | SVG | EPS |
> | Cohen's Kappa | 0.946 | 0.909 | 0.909 | |  0.960| 0.960| 0.960|
>
> | **Annotator 2 (Plane geometry)** |  |  |  | **(Molecular structure)**||||
> | --- | --- | --- | --- | --- | --- | --- | --- |
> |  | TikZ | SVG | EPS | | TikZ | SVG | EPS |
> | Cohen's Kappa | 0.927 | 0.982 | 0.909 | | 0.960 |1.000 |0.920|
>
> [**Do your annotators and your code both make a binary choice?**]
>
> As clearly shown in Figures 3 and 4, our code performs a binary judgment of “correct” or “incorrect.” We requested the annotators to provide binary judgments as well.
>
> **[(iv) do you have 50% instances labeled as 0 and 50% as 1 according to your code?]**
>
> Yes. The instances we used consist of 50% judged “correct” by our code and 50% judged “incorrect.” To clarify this point, including the above responses, we have updated the section **“G. Validation of Human-System Agreement.”**
>
> **[In Tables 7, etc., in which you report averages, do you take into account that for some metrics lower is better, while for others, higher is better?]**
>
> Yes. When applying min-max normalization, we normalized indicators where higher values are better such that the minimum becomes 0 and the maximum becomes 100. For indicators where lower values are better, we normalized them such that the minimum becomes 100 and the maximum becomes 0. To clarify this point, we updated **line 1578**.

---

### Comment · Area_Chair_x385 · 2025-11-22

Dear Authors and Reviewers,

I would like to thank the authors for providing detailed rebuttal messages on time.

To reviewers: I would like to encourage you to carefully read all other reviews and the author responses and engage in an open exchange with the authors. Please post your first response as soon as possible within the discussion time window. Ideally, all reviewers will respond to the authors, so that the authors know their rebuttal has been read.

Best regards,
AC

---

### Author Response · Authors · 2025-12-01
**Discussion Summary**

# Dear Area Chair,

We deeply appreciate your tremendous effort in handling this complex situation. Below, we summarize the key points of our discussion, and we hope this summary will support your decision-making.

---
# Our Contributions

First, we outline our main contributions:

- **Dataset:** We introduced SSVG-Bench, a new benchmark for structured scientific vector graphics generation.
- **Evaluation code:** We presented new evaluation scripts to assess the structural correctness of scientific vector graphics generated by LLMs.
- **Analysis:** We analyzed the performance of state-of-the-art LLMs using our dataset and evaluation code.
- **Method:** We proposed LOOP, a novel prompting technique that improves LLM performance.

Based on these contributions, we have prepared the following summary.

---
# Reviewer Q9nD (original rating: 2, final rating: 4)
Reviewer Q9nD initially raised the following main concerns:

- The details of our **evaluation code** are unclear.
- Our **analysis** relies on a single metric and lacks reliability.
- The number of models used to evaluate our **method** is insufficient.

We addressed these concerns as follows:

- To clarify the **evaluation code**, we included it directly in the paper.
- To improve the reliability of our **analysis**, we added multiple complementary metrics.
- To strengthen the evaluation of our **method**, we incorporated several additional models.

In response, Reviewer Q9nD **raised the rating from 2 to 4**. At the same time, the reviewer raised an additional concern:

- It should be demonstrated that the **evaluation code**’s judgments align with human judgments.

We addressed this concern as follows:

- We conducted additional experiments showing that our **evaluation code**’s judgments align with human judgments.

We provided further details in response to the reviewer's request; then, the discussion phase ended. Unfortunately, we did not receive further comments, but we believe that we have adequately addressed all of the concerns.

---
# Reviewer 8knp (original rating: 4, final rating: 6)

Reviewer 8knp initially raised the following main concerns:

- The diversity of data included in our **dataset** is limited.
- Only one dataset was used to evaluate our **method**, which is insufficient.
- The details of our **evaluation code** are unclear.

We addressed these concerns as follows:

- We clarified that our **dataset** contains representative data.
- We evaluated our **method** using an additional dataset.
- To clarify the **evaluation code**, we included it directly in the paper.

The reviewer raised an additional concern:

- It should be explained how the **evaluation code** was created.

We addressed this concern as follows:

- We provided further details on the **evaluation code**.

In response, Reviewer 8knp **raised the rating from 4 to 6**.

---
# Reviewer vtgB (original rating: 8, final rating: 8)

Reviewer vtgB initially raised the following main concerns:

- Our **evaluation code** has several limitations: extraneous elements are not penalized, and bond orders are ignored.
- Our **dataset** has several limitations: potential data leakage and moderate data size.
- No ablation study was conducted regarding our **method**.

We addressed these concerns as follows:

- We explained that the limitations of the **evaluation code** are not significant issues in practice.
- We clarified the substantial value of the **dataset** despite its limitations.
- We conducted an ablation study of our **method**.

After this response, the discussion phase ended. Unfortunately, we did not receive further comments, but we believe that we have adequately addressed all of the concerns.

---
# Reviewer r4TZ (original rating: 4, final rating: 4)

Reviewer r4TZ initially raised the following main concerns:

- The details of our **evaluation code** are unclear.
- The novelty of our **dataset** is limited.
- Our **method** is not exciting.
- In our **analysis**, some existing methods may be evaluated incorrectly.

We addressed these concerns as follows:

- To clarify the **evaluation code**, we included it directly in the paper.
- We explained that our **dataset** holds significant value.
- We clarified the substantial potential impact of our **method**.
- In our **analysis**, we clarified that we use a correct method to evaluate the existing methods.

The reviewer raised an additional concern:

- It should be demonstrated that the **evaluation code**’s judgments align with human evaluations.

We addressed this concern as follows:

- We conducted additional experiments showing that our **evaluation code**’s judgments align with human evaluations.

After this response, the discussion phase ended. Unfortunately, we did not receive further comments, but the reviewer left the comment:

> **If it (human evaluation) supports their findings, I would be happy to raise my score.**

We believe that, by adding the human evaluation results, we have fully addressed this concern.

---

### Meta-Review · Area_Chair_Yfe4 · 2026-01-06

**Summary:**

This work introduces a new benchmark, i.e., SSVG-Bench, to evaluate the generation of structured scientific vector graphics. The dataset consists of 110 and 300 unique examples from plane geometry and molecular structure, respectively. Each example contains three vector formats: TikZ, SVG, and EPS. The evaluation is conducted by task-specific Python scripts to capture details in the generated results. Finally, prompting is also investigated to demonstrate the potential.

**Reviewer Concerns:**

The concerns before rebuttal are mainly from insufficient technical details (Reviewer Q9nD, 8knp, r4TZ), limited diversity/examples in the benchmark (Reviewer 8knp, vtgB), ablation on prompt, i.e., LOOP (Reviewer Q9nD, 8knp, vtgB), cost/robustness of task-specific Python scripts for evaluation (Reviewer Q9nD, 8knp, r4TZ) and human evaluation (Reviewer Q9nD). While rebuttal shows the sufficient details of the benchmark and the evaluation pipeline, the size of the dataset is still quite limited, e.g., a total of 410 unique examples. Since the main contribution of this work is the benchmark as emphasized in rebuttal, the technique novelty of proposed prompting strategy is not the major concern. During discussion, Reviewer r4TZ also find that human evaluation is important to confirms the effectiveness of the proposed automatic evaluation metric. In revision, Table 15 demonstrates a preliminary result of human evaluation. Although it shows that the agreement between human and Python evaluation is highly consistent, the experiment is insufficient: only 2 persons are employed in the study and 50 out of 300 examples from molecular structure task are evaluated.

**Reviewer Scores:**

The submission had mixed scores before rebuttal. The rebuttal addressed partial concerns by providing more evaluation details and making the contribution of a dataset work clear. However, the key limitation is still unsolved. Therefore, reviewers may increase scores but the current version can be further improved and does not meet the bar of the conference.

---

### Decision · Program_Chairs · 2026-01-26

Reject